



# A first investigation of hydrogeology and hydrogeophysics of the Maqu catchment in the Yellow River source region.

Mengna Li[1,3], Yijian Zeng[1], Maciek W. Lubczynski[1], Jean Roy[2], Lianyu Yu[1], Hui Qian[3], Zhenyu Li[4], Jie Chen[3], Lei Han[5], Tom Veldkamp[1], Jeroen M. Schoorl[6], Harrie-Jan Hendricks Franssen[7], Kai Hou[3], Qiying Zhang[3], Panpan Xu[3], Fan Li[4], Kai Lu[4], Yulin Li[4], Zhongbo Su[1]

[1]Faculty of Geo-Information Science and Earth Observation (ITC), University of Twente, Enschede, 7500 AE, The Netherlands
[2]IGP, Outremont, QC H2V 4T9, Canada
[3]School of Water and Environment, Chang'an University, Xi'an 710054, China
[4]Institute of Geophysics and Geomatics, China University of Geosciences, Wuhan, 430074, China
[5]School of Land Engineering, Chang'an University, Xi'an 710054, China
[6]Soil Geography and Landscape Group, Wageningen University, P.O. Box 47, NL-6700 AA Wageningen The Netherlands
[7]Forschungszentrum Jülich GmbH, Agrosphere (IBG-3), Jülich, 52425, Germany

*Correspondence to*: Zhongbo Su (z.su@utwente.nl), Yijian Zeng (y.zeng@utwente.nl) and Hui Qian (qianhui@chd.edu.cn)

**Abstract.** The Tibetan Plateau is the source of most of Asia's major rivers and has been called the Asian Water Tower. Detailed knowledge of its hydrogeology is paramount to enable the understanding of groundwater dynamics, which plays a vital role in headwater areas like the Tibetan Plateau. Nevertheless, due to its remoteness and the harsh environment, there is a lack of field survey data to investigate its hydrogeology. In this study, borehole core lithology analysis, altitude survey, soil thickness measurement, hydrogeological survey, and hydrogeophysical surveys (e.g., Magnetic Resonance Sounding – MRS, Electrical Resistivity Tomography – ERT, and Transient Electromagnetic – TEM) were conducted in the Maqu catchment within the Yellow River Source Region (YRSR). The soil thickness measurements were done in the western mountainous area of the catchment, where hydrogeophysical surveys were difficult to be carried out. The results indicate soil thicknesses are within 1.2 m in most cases, and the soil thickness decreases as the slope increases. The hydrogeological survey reveals that groundwater flows from the west to the east, recharging the Yellow River. The hydraulic conductivity ranges from 0.2 m/d to 12.4 m/d. The MRS soundings results, i.e., water content and hydraulic conductivity, confirmed the presence of unconfined aquifer in the flat eastern area. The depth of the Yellow River deposits was derived at several places in the flat eastern area based on TEM results. These survey data and results can be used to develop integrated hydrological modeling and water cycle analysis to improve a full–picture understanding of the water cycle at the Maqu catchment in the YRSR. The raw data set is freely available at https://doi.org/10.17026/dans-z6t-zpn7 (Li et al., 2020).





## 1 Introduction

With a huge amount of water storage,  the Tibetan Plateau (TP) acts as the "Water Tower of Asia" (Qu et al., 2019; Wang et al., 2017), recharging many major Asian rivers including the Salween, Mekong, Brahmaputra, Irrawaddy, Indus, Ganges, Yellow, and Yangtze rivers (Immerzeel et al., 2009), feeding more than 1.4 billion people (Immerzeel et al., 2010), and promoting regional social and economic development (Xiang et al., 2016). Due to climate change, the TP has experienced

accelerated temperature rise over the past decades (Huang et al., 2017). Since the 1950s, the warming rate over the TP ranges between 0.16 °C – 0.36 °C per decade, and rises to 0.50 °C – 0.67 °C per decade from the 1980s (Kuang and Jiao, 2016). The retreating glaciers and snow cover, decreasing wetland area, and rising snow lines indicate that the hydrological system on the TP is undergoing profound changes (Kang et al., 2010; Xu et al., 2016; Yao et al., 2013; Zhao et al., 2004).

So far, the groundwater-related studies on the TP are mainly satellite-based, focusing on using GRACE to estimate terrestrial

water storage, which consists of surface water and subsurface water (Haile, 2011; Jiao et al., 2015; Zhong et al., 2009). Among those studies, Xiang et al. (2016) separated the groundwater storage from terrestrial water storage observed by GRACE using hydrological models and a glacial isostatic adjustment model.

An Integrated Hydrological Model (IHM), integrating groundwater with surface and above surface water fluxes, is essential for improving the understanding of different processes quantitatively (Graham and Butts, 2005). To set up an IHM, different

kinds of data are needed for parameterization of land surface and subsurface, for atmospheric forcing,  and state variables are required for model calibration and validation. Land surface data such as topography, land cover, and soil parameters can be obtained from Digital Elevation Models (DEMs) and regional or global soil databases (Su et al., 2011; Zhao et al., 2018). Atmospheric forcing data, including precipitation, air temperature, wind velocity, and other variables, are available from regional or global meteorological datasets (Su et al., 2013; Yang, 2017). However, subsurface data, like hydrogeological

information (i.e., lithology, water table depth, hydrogeological parameters) and state variables (i.e., hydraulic heads and soil moisture content), usually require in situ measurements. These hydrogeology–related data are usually the most difficult ones to acquire, particularly considering the remoteness and harsh environment of TP (Yao et al., 2019).

The conventional way to acquire hydrogeological information in an unknown area is by drilling boreholes and carrying out hydraulic tests, for example, pumping tests (Vouillamoz et al., 2012). However, due to the harsh environment of the TP, and

the high costs and time–consuming of the traditional hydrogeological survey methods, little work has been done on the TP.

The hydrogeophysical methods are up-and-coming in hydrogeological studies (Chirindja et al., 2016). They have been applied in various conditions, for example in: wetlands (Chambers et al., 2014), rivers (Steelman et al., 2015), proglacial moraine (McClymont et al., 2011), karst regions (McCormack et al., 2017), and volcanic systems (Di Napoli et al., 2016; Fikos et al., 2012). Compared to other hydrogeophysical methods, such as seismics, gravity and resistivity method, Magnetic Resonance

Sounding (MRS) is the only method that is able to detect the free water in the subsurface directly (Lubczynski and Roy, 2003; Lubczynski and Roy, 2004), and quantify hydrogeological parameters and water storage (Lachassagne et al., 2005; Legchenko et al., 2002; Legchenko et al., 2018; Lubczynski and Roy, 2007). The MRS excitation is done at the earth's magnetic field.



Earth System
Science
Data

Therefore it depends on the subsurface resistivity. The electrical resistivity measurement is suggested to be jointly used with MRS (Braun and Yaramanci, 2008; Descloitres et al., 2007; Vouillamoz et al., 2002). Electrical Resistivity Tomography (ERT)

is one of the predominantly employed hydrogeophysical methods to estimate the subsurface electrical resistivity (Herckenrath et al., 2012; Jiang et al., 2018). It has been widely applied together with MRS to explore regional hydrogeology (Vouillamoz et al. (2003), Descloitres et al. (2008), Pérez-Bielsa et al. (2012)). The Transient Electro-Magnetic survey (TEM), also referred to as the Time-Domain Electromagnetic Method (TDEM) in the literature, provides subsurface resistivity, but is able to achieve deeper penetration than ERT. On the TP, Gao et al. (2019) and You et al. (2013) used ERT to investigate permafrost.

Nevertheless, there has not been any work done on the TP in terms of joint use of MRS, TEM, and ERT for hydrogeological surveys.

Some investigations have been done on the TP based on existing DEMs. Zhang et al. (2006) analyzed the geomorphic characteristics of the Minjiang drainage basin with SRTM (Shuttle Radar Topography Mission) data. Wei and Fang (2013) assessed the trends of climate change and temporal-spatial differences over the TP from 1961–2010, with a generalized

temperature zone–elevation model and SRTM. Niu et al. (2018) mapped permafrost distribution throughout the Qinghai–Tibet Engineering Corridor based on ASTER Global DEM. However, before applying DEMs, it is essential to evaluate the accuracy of DEMs with a Real-time Kinematic-Global Positioning System (GPS-RTK), which has not been given attention in many studies over the TP.

This study jointly uses hydrogeological and hydrogeophysical methods, including aquifer tests, MRS, ERT, TEM, and other

necessary approaches at Maqu catchment in the Yellow River Source Region (YRSR) on TP. The paper is focusing on the data part. Setting up a hydrogeological conceptual model will be presented in another paper. In what follows, the study area is introduced in Sect. 2. Borehole core lithology analysis, altitude survey, soil thickness measurement, hydrogeological survey, and hydrogeophysical survey are presented in Sect. 3. The results are documented and discussed in Sect. 4. Data availability is given in Sect.5. Conclusions are made in Sect. 6.

## 2 Study area

The study area is a catchment (33°43′ N – 33°58′ N, 101°51′ E – 102°16′ E) in Maqu county, China. It is located at the northeastern edge of the TP, the first major bend of the Yellow River. Maqu is regarded as the "reservoir" of the YRSR. The length of the Yellow River passing through Maqu is 433.3 km. When the Yellow River flows through Maqu county, the annual runoff increases by 10.8 billion $m^3$, accounting for 58.7% of the total runoff of 18.4 billion $m^3$ of the Yellow River in the

YRSR (Wang, 2008). The Maqu catchment is characterized by a cold climate with dry winter and warm summer (Dwb) in the updated Köppen–Geiger climate classification (Peel et al., 2007). The annual mean temperature is about 1.8 °C, and the precipitation is around 620mm annually. The catchment is covered by short grasses used for grazing by yaks and sheep. The elevation ranges between 3367 to 4017 m.a.s.l. according to ALOS PALSAR RT1.

In terms of geomorphology and geology, the catchment can be divided into two parts, the flat eastern area and the western

mountainous area. The western mountains are feldspathic quartzose sandstone and sandy slate with soil covered at the top. While in the east part, the sediments are mainly alluvial deposits with intercalated eolian units. It is a high energy environment in which water is moving fast and able to carry particles of large grain sizes. The eastern part, together with its extension outside of the study area, is called the Ruoergai Basin. Surface processes cause erosion, mixing, unmixing, and redistribution of alluvial materials within the thick alluvia accumulation on the Eastern part. Geomorphological characterization was carried

out in the Maqu catchment in 2018, and three terraces were identified (Fig. 1).

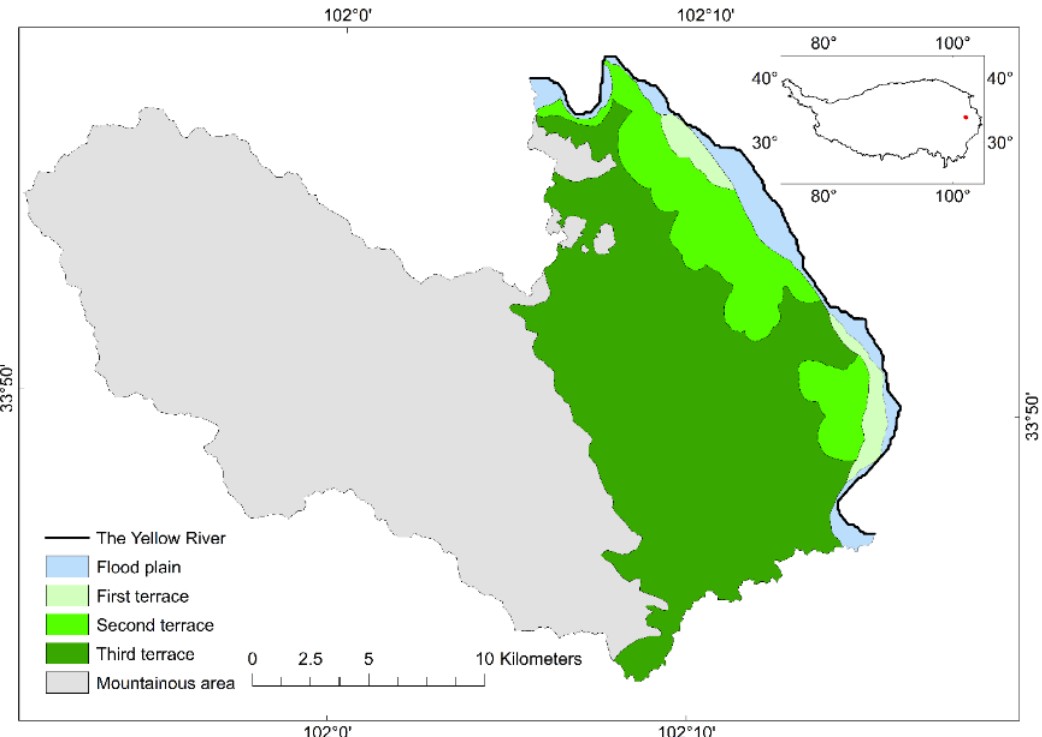

**Figure 1. The geographical location of Maqu catchment in the TP and geomorphologic map.**

Some previous works have been done in or around the catchment. Su et al. (2011) monitored the soil moisture and soil temperature from 5 to 80 cm below the ground surface. Dente et al. (2012) assessed the reliability of AMSR-E and ASCAT

soil moisture products. Zheng et al. (2016) investigated the impacts of Noah model physics on catchment-scale runoff simulations. Zeng et al. (2016) combined the in situ soil moisture networks with the classification of climate zones to produce the in situ measured soil moisture climatology at the plateau scale. Zhao et al. (2018) studied the soil hydraulic and thermal properties of the 0.8 m top soil column. Zhuang et al. (2020) blended the surface soil moisture data from satellites, land data assimilation, and in–situ measurements with the constraint of in–situ data climatology, and estimated the root zone soil

moisture by scaling the blended surface soil moisture product. The present research focuses on the hydrogeological and hydrogeophysical aspects, complementing previous studies.



# 3 Materials and methods

Figure 2 shows the fieldwork workflow towards establishing a hydrogeological conceptual model, which includes the borehole core lithology analysis, altitude survey, soil thickness measurement, hydrogeological survey, and hydrogeophysical survey

(Table 1 and Fig. 2). Borehole core lithology was analyzed in 2017. Altitudes were surveyed in 2019. Soil thicknesses were measured in both 2018 and 2019. The hydrogeological survey was carried out during 2017, 2018, and 2019, including water table depth measurements and aquifer tests. The hydrogeophysical survey was conducted in 2018 and 2019, deploying magnetic susceptibility measurements with magnetic susceptibility meter, resistivity measurement with ERT and TEM, and water content and transmissivity measurement with MRS. The locations of the surveys and measurements are shown in Fig. 3

and Fig. 4.

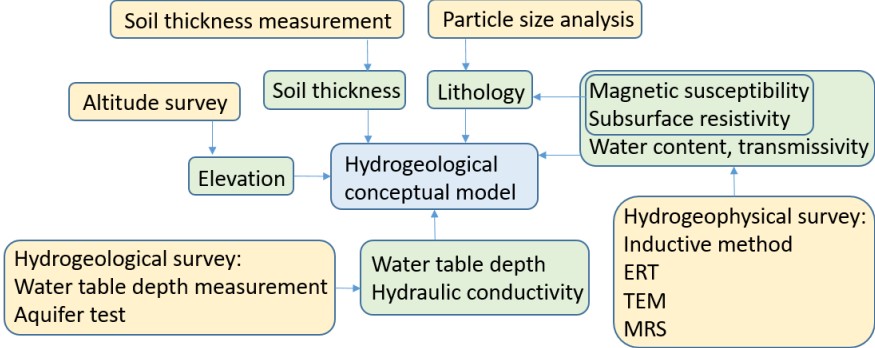

**Figure 2. Fieldwork workflow for setting up a hydrogeological conceptual model at Maqu catchment.**

**Table 1. Methods, equipment, and timing for carrying out relevant measurements as in Figure 2.**

| Item | | Method | Equipment | Time | Number of measurements | Source |
|---|---|---|---|---|---|---|
| Borehole core lithology | | Particle size analysis | Sieve | 2017 | 1 | Well Report |
| Altitude | | GPS-RTK | CHCNAV T4 | 2019 | 46 | fieldwork |
| Soil thickness | | Sampling | Auger, clinometer | 2018,2019 | 77 | fieldwork |
| Hydrogeological survey | Water table depth | Manual | Dipper | 2018,2019 | 40* | fieldwork |
| | Hydraulic conductivity | Aquifer tests | Logger (3001–M10 Levelogger Edge and TD–Diver), pump, slug | 2017,2019 | 11 | fieldwork |
| Hydrogeophysical survey | Magnetic susceptibility | Inductive method | SM–20 | 2019 | 11 | fieldwork |
| | Subsurface resistivity | ERT | WGMD–9 | 2018 | 7 | fieldwork |
| | | TEM | TEM–FAST–48 | 2019 | 10 | fieldwork |
| | Water content, Transmissivity | MRS | Numis Poly | 2018 | 18* | fieldwork |

\* sporadic measurements, not time series.


(a)

(b)

(c)

(d)

(e)

(f)

(g)

**Figure 3. (a) Locations of the hydrogeological surveys, elevation measurements, and soil thickness measurements. (b), (c), (d), (e),**
**(f), and (g) are the exact locations of soil thickness measurements at sites b, c, d, e, f, g, respectively shown in (a), in the *.KML**

**formatted image from © Google Earth. The numbers from 1 to 46 (due to limited space, several numbers are not shown in the figure) indicate the measurement sequence of GPS-RTK, and the sequence from b to f indicates the measurement sequence of soil thickness.**

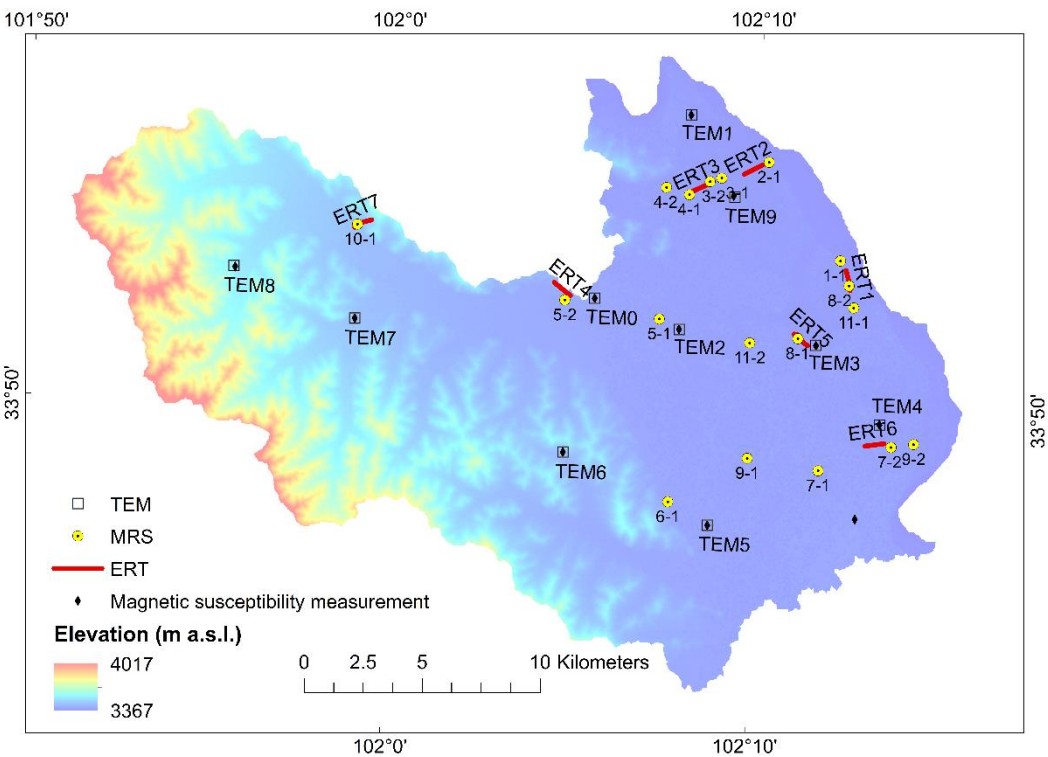

**Figure 4. Location of hydrogeophysical surveys.**

## 3.1 Borehole core lithology

The borehole core lithology is helpful in terms of understanding the formation of the area and estimating hydrogeological parameters. Some boreholes are available for water table depth measurement in the study area, but information of borehole core lithology is only available in one borehole ITC_Maqu_1 (Fig. 3a) drilled in 2017 down to the depth of 32 m from the ground surface. According to the borehole report, the lithology of the core was determined based on particle size analysis using the sieving method. Samples were analyzed using sieves with mesh sizes of 60, 40, 20, 10, 5, 2, 1, 0.5, 0.25, and 0.075 mm.

## 3.2 Altitude survey

The accuracy of ground surface elevation is crucial for groundwater modeling because it influences hydraulic heads, hydraulic gradient, and also groundwater flow and its direction. As a dynamic type of GPS positioning technique, GPS-RTK is able to achieve point position and elevation with centimeter-level accuracy in real-time. GPS-RTK instrument CHCNAV T4 from Shanghai Huace Navigation Technology Limited (https://www.chcnav.com), with a vertical accuracy of 3 cm and a horizontal accuracy of 2 cm, was employed to measure elevations in 2019. Before obtaining the first results, we spent a few minutes to





initialize the system. Among the 46 elevation measurements made in total, 33 were located in the flat eastern area, and 13 in the mountainous area (see Fig. 3a). The data was intended to be used to evaluate seven DEM datasets (Table 2). The most accurate DEM will be applied as the top model boundary in groundwater modeling and also for calculation of hydraulic heads

where the ground-based altitude survey is not available. Seven DEMs are all open access and were downloaded from websites of the United States Geological Survey (USGS), Japan Aerospace Exploration Agency (JAXA), and Alaska Satellite Facility (ASF).

**Table 2. Seven different DEM datasets.**

| Number | Name | DEM | Resolution | Source |
|---|---|---|---|---|
| 1 | SRTM | Shuttle Radar Topography Mission | 1 Arc–Second | USGS |
| 2 | ASTER V1 | ASTER GDEM Version 1 | 1 Arc–Second | USGS |
| 3 | ASTER V2 | ASTER GDEM Version 2 | 1 Arc–Second | USGS |
| 4 | ASTER V3 | ASTER GDEM Version 3 | 1 Arc–Second | USGS |
| 5 | AW3D30 | ALOS World 3D – 30 m Version 2.2 | 30 m | JAXA |
| 6 | ALOS RT2 | ALOS PALSAR RT2 | 30 m | ASF |
| 7 | ALOS RT1 | ALOS PALSAR RT1 | 12.5 m | ASF |

### 3.3 Soil thickness measurement

Due to limited conditions for hydrogeophysical surveys in the mountainous west, we sampled the thickness of the overlying soils in the west to build the hydrogeological conceptual model and to validate simulations of spatially distributed soil thickness by landscape evolution models like LEM LAPSUS (Schoorl et al., 2006; Schoorl et al., 2002) (will be presented in another paper). In the mountainous west, feldspathic quartzose sandstone and sandy slate parent materials show variable soil depths related to landscape position. The fieldwork was carried out at six sites (see Fig. 3b–3g). Measurements in sites 1 and 2 were

conducted in 2018, while the rest in 2019. Soil thickness and slope of the ground surface were measured using an auger and a clinometer from Eijkelkamp Soil & Water Company (https://en.eijkelkamp.com). The exact measurement positions at each site were decided based on slope forms and surface pathways.

### 3.4 Hydrogeological surveys

### 3.4.1 Water table depth measurement

Water table depth information is important for hydrology and hydrogeology. By subtracting the water table depth from ground surface elevation, the hydraulic head and further the regional groundwater piezometric map can be obtained to enable a general understanding of the groundwater flow system in the study area. We measured 40 water table depths in 36 boreholes during 05-08 August 2018 and 20 August – 05 September 2019 using a dipper (Fig. 3a). In six boreholes, water table depths were measured both in 2018 and 2019. Eight level-loggers were installed to monitor the long–term groundwater level fluctuation,

but the data are not available yet.



### 3.4.2 Aquifer tests

Aquifer tests, including pumping tests and slug tests, were conducted to obtain hydraulic conductivity. The first pumping test was done in 2017, in the borehole ITC_Maqu_1, where core lithology information is available (Fig. 3a). The pumping rate was 55.6 m³/d measured with a flowmeter, and the pumping duration was about 30 minutes. The pumping rate was limited
because the borehole ITC_Maqu_1 could easily collapse if the pumping rate were too high. The water level became stable soon after the start of pumping and was recorded every minute using a data logger (TD–Diver manufactured by Van Essen Instruments, with a range of 10 m). Other tests were carried out in 2019, including two pumping tests and eight slug tests (Fig. 3a). For the two pumping tests with the pumping rate of 31.6 m³/d and 101.52 m³/d, due to practical reasons, only water level recovery data were analyzed. In the eight slug tests, the groundwater level was abruptly lowered by extracting 11.75 L water
from the well. The water levels were recorded every second or two seconds in slug tests and every five seconds or 20 seconds in pumping tests using a data logger (3001 Levelogger Edge manufactured by Solinst, with a range of 10 m). The methods used for analyzing the data of pumping tests and slug tests were chosen based on the aquifer information from hydrogeophysical survey and well–related information from local borehole owners.

### 3.5 Hydrogeophysical surveys

### 3.5.1 Magnetic susceptibility

The magnetic susceptibility of rocks changes the local geomagnetic field. The magnetic rocks, which lead to different gradient and intensity of the geomagnetic field, result in different Larmor frequency and further can make the MRS signal undetectable (Lubczynski and Roy, 2007; Plata and Rubio, 2007). The MRS sounding is usually not possible when the magnetic susceptibility is larger than $10^{-2}$ SI units, but possible when it is lower than $10^{-3}$ SI units, and may be or may not be possible
within the interval probably depending on the remanent magnetization of the material (Bernard, 2007). Therefore, it is always recommended to measure the magnetic susceptibility before embarking on a large scale MRS survey (Roy et al., 2008). In this study, portable magnetic susceptibility meter SM–20 was used to measure the magnetic susceptibility at 11 sites in the field (Fig. 4). At each site, an average magnetic susceptibility was obtained from 3–5 repeated measurements.

### 3.5.2 ERT

Subsurface resistivity depends on many different parameters, e.g., lithology, water content, and water conductivity. Its distribution in the subsurface can be visualized by 2D ERT. ERT was employed in this study because it provides subsurface resistivity, which not only supports the analysis of MRS measurements but also can give us a general understanding of the aquifer.

We performed seven ERT surveys with ERT instrument WGMD–9 manufactured by Chongqing Benteng Digital Control
Technical Institute (http://www.cqbtsk.com.cn), China using two configurations, Wenner and dipole-dipole. Wenner and dipole-dipole are standard and commonly used configurations. Wenner usually has a good signal–to–noise ratio (S/N) and is



good at detecting vertical changes in resistivity, i.e., suitable to image horizontal structures. Dipole-dipole is sensitive to horizontal changes in resistivity, so it is ideal for vertical structure delineation. Multicore cables with a fixed electrode spacing of 10 m were used in the field. The length of cable was 890 m for ERT1 – ERT4, and 810 m for ERT5 – ERT7 (see Fig. 4).

Electrode positions were measured with a hand-held GPS instrument Unistrong MG858s (http://www.unistrong.com), with a horizontal and vertical accuracy of 30 cm. The industry–standard RES2DINV V3.54 (Loke, 1999) was employed for ERT inversion.

### 3.5.3 MRS

MRS was conducted to define aquifer geometry, estimate hydraulic conductivity or transmissivity and water content with

depth. In total, 18 soundings (Fig. 4) were performed using MRS instrument Numis Poly, the latest version of MRS equipment from the IRIS Instrument company (http://www.iris–instruments.com). The Larmor frequency, measured with the proton magnetometer in the field, was set at 2241.8 Hz, and the inclination of the earth's magnetic was set at 52° N. A square loop with a side length of 150 m or 100 m was used. Positions were measured with Unistrong MG858s, with a horizontal and vertical accuracy of 30 cm.

To estimate hydraulic conductivity, the decay time constant $T_d$ is used. There are three kinds of $T_d$: longitudinal decay time constant $T_1$, transverse decay time constant $T_2$, and free induction decay time constant $T_2^*$. With the current instrument, only $T_1$ (actually an approximate value $T_1^*$) and $T_2^*$ are available. The Seevers equation (Seevers, 1966) (Eq. 1) and the Kenyon equation (Kenyon et al., 1989) (Eq. 2) can be used for estimating hydraulic conductivity $K$ (m/d):

$$K = C_p \theta_{MRS} T_d^2 \tag{1}$$

$$K = C_p \theta_{MRS}^4 T_d^2 \tag{2}$$

where $C_p$ is the calibration coefficient, which is a lithology dependent factor that needs to be calibrated from the pumping test (dimensionless). $\theta_{MRS}$ is the MRS estimated water content (%). Compared to the Kenyon equation, Seevers equation is more accurate (Plata and Rubio, 2008) and has been widely used (e.g., Legchenko et al. (2002), Vouillamoz et al. (2007), Nielsen et al. (2011)) and is used in this study. Once $K$ is estimated, the transmissivity $T$ (m²/d) can be calculated using the equation:

$$T = K \cdot \Delta z \tag{3}$$

where $\Delta z$ is the layer thickness (m) derived from MRS inversion.

Based on the study from Vouillamoz et al. (2008), MRS transmissivities are close to transmissivities estimated from pumping tests, the uncertainties in transmissivity estimated from MRS and pumping tests are comparable, and the mean relative uncertainty of the MRS determined water content is 20%. Boucher et al. (2009) and Vouillamoz et al. (2014) confirmed that

aquifer transmissivity could be estimated from MRS results with an averaged uncertainty of about 70%.

MRS data were interpreted with an open-access software Samovar V6.6 from the IRIS Instrument company (http://www.iris–instruments.com), which is based on the Tikhonov regularization method (Legchenko and Shushakov, 1998). Samovar assumes the default calibration coefficient $C_p$ of 7E–09 for sandy aquifers and aquifers composed of weathered and highly



fractured rock based on MRS calibration experience in France (Legchenko et al., 2004). In this study, $C_p$ was estimated using
pumping test data.

### 3.5.4 TEM

Compared to ERT, TEM also provides subsurface resistivity but with deeper penetration, a relatively lower resolution, and a shorter time of data acquisition. TEM instrument is usually operated in a 1D sounding mode as compared to the ERT 2D profiling mode. Since magnetic fields propagate faster in resistive media than in conductive ones, TEM is advantaged in low
resistivity media and mapping deep conductive targets. Similarly to MRS but with different constraints, there is a dead time between the excitation or transmitter function and the detection or receiver function which are time-shared. Such TEM deadtime is much shorter than in the case of MRS. TEM commonly involves placing a square loop on the targeted place and performing soundings. It generates a primary magnetic field that is abruptly interrupted to produce induced eddy currents in the subsurface. The eddy currents will lead to a secondary magnetic field, which can be detected by the loop on the ground
surface. The received signals can be used to estimate subsurface resistivities by using appropriate inversion techniques (Nabighian and Macnae, 1991).

The TEM soundings were performed at ten locations (Fig. 4) using TEM instrument TEM–FAST 48. Developed by Applied Electromagnetic Research Limited (http://www.aemr.net), TEM–FAST 48 is very small, compact, portable, and easy to deploy and apply in the field (Gonçalves, 2012). Only one TEM configuration was used, i.e., coincident square loop, of one loop that
combines functions of the transmitter and receiver. At each location, different loop sizes (3 m – 95 m), time ranges (3 – 9), stacks (5 – 10), and currents (0.7 A – 1.1 A) were applied to select the optimal data set, which has the maximum investigation depth. If abrupt changes occurred in the obtained curve, presenting the relation between apparent specific resistivity and time, the measurement was repeated to ensure data quality. After field collection, data were processed using TEM–Researcher proprietary software (http://www.aemr.net) based on the solution of the inverse problem in time domain electromagnetic
sounding.

## 4 Results and Discussion

### 4.1 Borehole core lithology

The lithology is shown in Table 3. The top layer is eolian sand and loam. There are dunes that have been blown out of the river bed on top of the terraces. The deep layer is fluvial sediment. Based on the lithology information, the range of lithology related
parameters can be estimated. According to Chen et al. (1999), the Ruoergai Basin was occupied by a large inland lake during the Quaternary before around 40 ka BP, while currently, it is a dry lake basin, with lake deposits exceeding 300 m in thickness. The extend of the ancient lake and Quaternary lake deposits are shown in Fig. 5. Based on Fig. 5 and the log of the ITC_Maqu_1 borehole shown in Table 3, the east of our study area is covered with thick lake sediments at depth, while the shallower part



would be covered with the Yellow River deposits with the thickness larger than 32 m. This conclusion is consistent with the
log of two other boreholes located to the east of the study area in Ruoergai Basin, RM (33°57′, 102°21′) and RH (33°54′, 102°33′) (Fig. 5). RH is about 40 km east of the study area, with a depth of 120 m, not reaching bedrock. The top 12.4 m of coarse sediment, i.e., sands, was deposited by rivers, while the deeper deposits are lake sediments, mainly composed of silt clay, clay silt, and clay (Wang et al., 1995). RM is about 20 km east of the study area, with a depth of 310 m. Like RH, RM core also reveals thick lake sediments, with thin river deposits on the top (Xue et al., 1998).

**Table 3. The core lithology of borehole ITC_Maqu_1.**

| Depth (m) | Thickness (m) | Lithology |
|-----------|---------------|-----------|
| 0.0 ~ 0.8 | 0.8 | sandy loam |
| 0.8 ~ 25.5 | 24.7 | fine sand |
| 25.5 ~32.0 | 6.5 | fine sand with gravel |

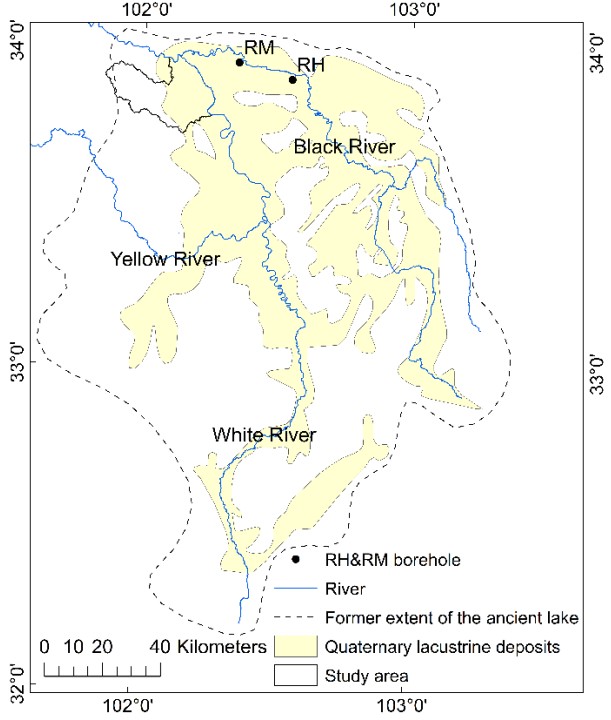

**Figure 5. Location of boreholes RM and RH (after Chen et al. (1999) ).**

**4.2 Altitude survey**

46 elevations were measured, 33 in the flat east, 13 in the mountainous west, and were used to evaluate the accuracies of seven DEM datasets (Fig. 6) and select the most accurate one. The statistical analysis results of the seven DEMs in the study area are shown in Table 4. The root mean squared error (RMSE) of ALOS RT1 and ALOS RT2 are 5.695 m and 5.477 m,





respectively, much smaller than the RMSE of the other five DEMs. The correlation coefficient, the mean error, and the mean absolute errors of ALOS RT1 and ALOS RT2 also show better performance than those of the other five DEMs. Comparing ALOS RT1 with ALOS RT2, ALOS RT1 slightly outperforms ALOS RT2 with regards to RMSE, correlation coefficient, and the mean error. Table A1 and Table A2 in the Appendix list the statistical analysis results of seven DEMs, separately for the flat eastern area and the mountainous western area. Seven DEMs all behave better in the west than the east in terms of the correlation coefficient. In the west, the correlation coefficients of seven DEMs are all larger than 0.94, while in the east, the correlation coefficients are all lower than 0.24. This is because the range of elevation in the flat east is much smaller than the range of elevation in the mountainous west. With regard to the RMSE, mean error, and mean absolute error, all seven DEMs have better behavior in the east than in the west. In general, ALOS RT1 and ALOS RT2 also outperform the other five DEMs, according to Table A1 and Table A2.

Since ALOS RT1 performs slightly better than ALOS RT2 in the whole study area and has a higher resolution than ALOS RT2, it is the most suitable DEM to use in this study area. For ALOS RT1 in the flat east, 52% errors (DEM value – GPS-RTK value) are within the range of –3 m to 3 m, and 79% errors are within the scope of –5 m to 5 m. While in the mountainous west, 54% errors are within the range of –8 m to –12 m, and 46% errors are within the range of 0 m to 7 m.

Previous TP works about DEM evaluation mainly focused on SRTM and ASTER. Our results are generally consistent with previous studies in terms of RMSE of SRTM. Nan et al. (2015) evaluated the height accuracy of SRTM and ASTER in eastern TP with reference to the relatively high precision of 1:50,000 scale DEM surveyed and mapped by the State Bureau of Surveying and Mapping in China. As a result, RMSE of SRTM and ASTER are 35.3 m and 50.2 m, respectively. Ye et al. (2011) evaluated SRTM and ASTER in the Mt. Qomolangma (Mt. Everest) area on the TP, by comparing 211 elevation checkpoints on the 1:50,000 topographic maps surveyed and mapped by State Bureau of Surveying and Mapping in China, demonstrating an average height difference of 31.3 m and 44.9 m for SRTM and ASTER, respectively. However, there are other studies that have different evaluation results. Fujita et al. (2008) found that the elevation differences between DEMs and ground survey data from differential GPS were 11.0 m for ASTER and 11.3 m for SRTM in the Lunana region, Bhutan Himalaya. The DEM evaluation results also indicated that in different places over the TP, the satellite DEM estimates are acquired with varying accuracy. This may be due to different topographic complexity in different areas.





**Figure 6. DEM elevations vs. GPS-RTK elevations.**

305





**Table 4. Statistical analysis of seven DEMs in the study area.**

| DEM | Resolution | Min Error * (m) | Max Error (m) | Max Error – Min Error (m) | MAE (Mean Absolute Error) (m) | ME (Mean Error) (m) | Correlation coefficient | RMSE (m) |
|---|---|---|---|---|---|---|---|---|
| SRTM | 1 Arc–Second | 22 | 44 | 22 | 35.488 | 35.488 | 0.985 | 35.936 |
| ASTER V1 | 1 Arc–Second | −17 | 43 | 60 | 24.761 | 24.010 | 0.950 | 26.565 |
| ASTER V2 | 1 Arc–Second | −8 | 55 | 63 | 27.483 | 27.140 | 0.941 | 30.171 |
| ASTER V3 | 1 Arc–Second | 4 | 45 | 41 | 28.988 | 28.988 | 0.962 | 30.438 |
| AW3D30 | 30 m | 25 | 44 | 19 | 36.249 | 36.249 | 0.985 | 36.707 |
| ALOS RT2 | 30 m | −13 | 8 | 21 | 4.592 | −0.338 | 0.985 | 5.695 |
| ALOS RT1 | 12.5 m | −12 | 8 | 20 | 4.404 | −0.360 | 0.986 | 5.477 |

* Error = DEM value – GPS-RTK value

## 4.3 Soil thickness measurement

Soil thickness measurements (Table 5) indicate that in most cases, the soil thicknesses are within 1.2 m, and the soil thicknesses
310    increase from the mountain top to the slope bottom. Besides, the soil thickness decreases as the slope increases (Fig. 7). Under
the soil layer, a less weathered layer exists where water can also flow and needs to be taken into account in the conceptual
model. In the field, the difference between the less weathered layer and the soil layer is that the less weathered layer contains
partially weathered stones. According to the owners of three wells located in or near the valley, the depths of three wells are
larger than 10 m and do not reach bedrock. Studies from Yan et al. (2020) and Shangguan et al. (2017) estimated the depth to
315    bedrock within China and on a global scale, respectively. By combining the depth to bedrock information with our results, the
thickness of the less weathered layer can be estimated later when establishing the hydrogeological conceptual model.

**Table 5. Soil thickness measurements and the locations of each measurement can be found in Figure 3.**

| No. | 1 | 2 | 3 | 4 | 5 | 6 | 7 | 8 | 9 | 10 | 11 | 12 | 13 | 14 | 15 | 16 | 17 | 18 | 19 | 20 |
|---|---|---|---|---|---|---|---|---|---|---|---|---|---|---|---|---|---|---|---|---|
| Depth (cm) | 39 | 45 | 28 | 48 | 50 | 46 | 39 | 34 | 37 | 42 | 23 | 52 | 42 | 35 | 38 | 50 | 40 | 38 | 42 | 37 |
| Slope (°) | 9 | 20 | 25 | 16 | 22 | 14 | 25 | 41 | 22 | 19.5 | 20 | 0 | 3 | 3 | 4 | 9 | 10 | 10 | 15 | 8 |
| No. | 21 | 22 | 23 | 24 | 25 | 26 | 27 | 28 | 29 | 30 | 31 | 32 | 33 | 34 | 35 | 36 | 37 | 38 | 39 | 40 |
| Depth (cm) | 40 | 30 | 30 | 35 | 28 | 29 | 71 | 90 | >120 | 110 | >120 | >107 | >110 | 59 | 85 | 60 | 92 | 38 | 41 | 76 |
| Slope (°) | 10 | 5 | 4 | 4 | 1 | 0 | 10 | 11 | 5 | 5 | 5 | 2 | 4 | 13 | 13 | 20 | 13 | 10 | 20 | 30 |
| No. | 41 | 42 | 43 | 44 | 45 | 46 | 47 | 48 | 49 | 50 | 51 | 52 | 53 | 54 | 55 | 56 | 57 | 60 | 61 | 62 |
| Depth (cm) | 55 | 32 | 80 | 27 | 49 | 52 | 43 | 44 | 30 | 74 | 37 | 81 | 102 | 102 | 104 | 100 | 92 | 40 | 53 | 61 |
| Slope (°) | 30 | 40 | 35 | 30 | 30 | 30 | 20 | 22 | 25 | 14 | 12 | 6 | 6 | 14 | 6 | 13 | 10 | 9 | 6 | 15 |
| No. | 63 | 64 | 65 | 66 | 67 | 68 | 69 | 70 | 71 | 72 | 73 | 74 | 75 | 76 | 77 | 78 | 79 | | | |
| Depth (cm) | 70 | 63 | 61 | 87 | 60 | 63 | 68 | 87 | 30 | 85 | 41 | 83 | 67 | 63 | >110 | >110 | 42 | | | |
| Slope (°) | 7 | 14 | 9 | 10 | 5 | 7 | 15 | 18 | 14 | 20 | 17 | 13 | 27 | 20 | 20 | 10 | 15 | | | |



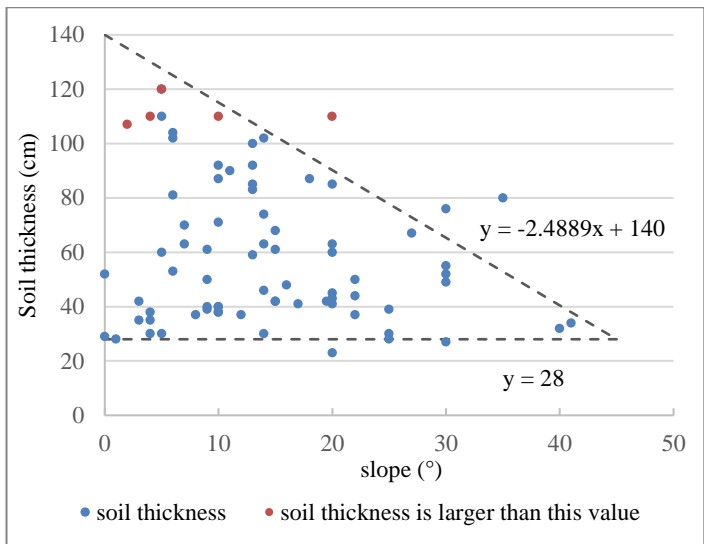

**Figure 7. Soil thickness (m) vs. slope (°).**

**4.4 Hydrogeological surveys**

**4.4.1 Water table depth measurement**

22 water table depths were measured in 2018, and 18 water table depths were measured in 2019 (Table 6). In the flat eastern area, the depths were interpolated in Surfer using the default Ordinary Kriging method with the linear variogram model
(slope=1, anisotropy ratio=1, anisotropy angle=0) which provides reasonable grids in most circumstances (Fig. 8a and Fig. 8b). Owing to the fact that most people living in the mountains use water from streams, only three wells were found and water table depths were measured in the mountainous west, but they were excluded during interpolation because water table depth is strongly controlled by topography, and the three measurements are far from enough to provide a reasonable estimation of water table depth in the west. In both 2018 and 2019, the interpolated water table depths show a similar trend that the depth
increases from the middle of the study area to the eastern boundary. However, the range of water table depth in 2018 is slightly larger than the range of water table depth in 2019. This is because the dam gates were open to lower the water level in the reservoir (Fig. 3a) in 2019 to facilitate nearby constructions. So water table depths at positions 1, 11, 21, and 22 were smaller in 2019 compared to 2018 (see Fig. 8a and Fig. 8b). In general, the range of water table depth is between 0.0 m to 19.1 m in 2018 and between 0.7 m to 18.0 m in 2019.
At 13 water table depth measurement locations, the elevations are available from the GPS-RTK survey and are shown in Table 7 with two decimal places, while ALOS RT1 extracted elevations are in integer form due to relatively low accuracy. These elevations are used to derive hydraulic heads by subtracting the water table depths from the ground surface elevation. Using the Kriging method, hydraulic heads were interpolated to obtain piezometric maps in the flat east (Fig. 8c and Fig. 8d). According to the map, in both 2018 and 2019, hydraulic heads decrease from the middle of the study area to the eastern





boundary. The difference of water table depth in 2018 and 2019 (Fig. 8c) is mainly caused by 1) different positions and amount of control points; 2) the gates were open to lower the water level in the reservoir in 2019;

In the study area, the western part plays a vital role in collecting water, whereas the east is mainly for storing water. Streams flow from the mountainous west to the flat east, and also, groundwater flows from west to east, recharging the Yellow River. This is consistent with the conclusion from Chang (2009) that the groundwater in Maqu county is recharging the Yellow River.

**Table 6. Water table depth measurements. GPS-RTK measurements of elevations are given with two decimal places, while ALOS RT1 extracted ones given in integer form.**

| Borehole | Latitude (°) | Longitude (°) | Elevation (m) | Logger installed date (dd/mm/yy) | 2018 Measurement | | | 2019 Measurement | | |
|---|---|---|---|---|---|---|---|---|---|---|
| | | | | | Date (dd/mm) | Depth (m) | Head (m) | Date (dd/mm) | Depth (m) | Head (m) |
| 1 | 33.932 | 102.117 | 3401 | | 05/08 – 08/08 | 18.80 | 3382 | 24/08 | 17.95 | 3383 |
| 2 | 33.921 | 102.149 | 3395 | | 05/08 – 08/08 | 13.22 | 3382 | | | |
| 3 | 33.918 | 102.136 | 3394 | | 05/08 – 08/08 | 13.65 | 3380 | | | |
| 4 | 33.904 | 102.127 | 3396 | | 05/08 – 08/08 | 8.40 | 3388 | | | |
| 5 | 33.890 | 102.128 | 3395 | | 05/08 – 08/08 | 1.20 | 3394 | | | |
| 6 | 33.876 | 102.093 | 3406 | | 05/08 – 08/08 | 2.50 | 3404 | 23/08 | 2.40 | 3404 |
| 7 | 33.864 | 102.126 | 3393 | | 05/08 – 08/08 | 0.68 | 3392 | | | |
| 8 | 33.864 | 102.146 | 3398 | | 05/08 – 08/08 | 2.00 | 3396 | | | |
| 9 | 33.863 | 102.147 | 3394 | | 05/08 – 08/08 | 1.96 | 3392 | | | |
| 10 | 33.877 | 102.172 | 3397 | | 05/08 – 08/08 | 9.13 | 3388 | | | |
| 11 | 33.884 | 102.198 | 3390.25 | | 05/08 – 08/08 | 9.90 | 3380.35 | 27/08 | 9.50 | 3380.75 |
| 12 | 33.860 | 102.190 | 3393 | | 05/08 – 08/08 | 10.02 | 3383 | | | |
| 13 | 33.857 | 102.170 | 3395 | | 05/08 – 08/08 | 6.30 | 3389 | | | |
| 14 | 33.837 | 102.141 | 3394 | | 05/08 – 08/08 | 1.37 | 3393 | | | |
| 15 | 33.811 | 102.143 | 3401 | | 05/08 – 08/08 | 0.80 | 3400 | | | |
| 16 | 33.790 | 102.147 | 3405.67 | 29/08/2019 | 05/08 – 08/08 | 1.47 | 3404.20 | 29/08 | 1.48 | 3404.19 |
| 17 | 33.832 | 102.189 | 3396 | | 05/08 – 08/08 | 8.57 | 3387 | | | |
| 18 | 33.824 | 102.185 | 3395 | | 05/08 – 08/08 | 7.08 | 3388 | | | |
| 19 | 33.820 | 102.185 | 3398 | | 05/08 – 08/08 | 7.72 | 3390 | | | |
| 20 | 33.818 | 102.186 | 3401 | | 05/08 – 08/08 | 6.77 | 3394 | | | |
| 21 | 33.830 | 102.225 | 3392.64 | 28/08/2019 | 05/08 – 08/08 | 12.80 | 3379.84 | 28/08 | 12.08 | 3380.56 |
| 22 | 33.794 | 102.214 | 3395.64 | | 05/08 – 08/08 | 10.51 | 3385.13 | 29/08 | 9.75 | 3385.89 |
| 23 | 33.947 | 102.135 | 3398.92 | 27/08/2019 | | | | 27/08 | 11.16 | 3387.76 |



| 24 | 33.916 | 102.155 | 3394.00 | | | 04/09 | 11.70 | 3382.30 |
|---|---|---|---|---|---|---|---|---|
| 25 | 33.872 | 102.143 | 3394.41 | | | 23/08 | 2.23 | 3392.18 |
| 26 | 33.865 | 102.132 | 3395.15 | 05/09/2019 | | 05/09 | 1.63 | 3393.52 |
| 27 | 33.860 | 102.194 | 3394.10 | 28/08/2019 | | 28/08 | 9.30 | 3384.80 |
| 28 | 33.774 | 102.187 | 3400 | | | 20/08 | 4.10 | 3396 |
| 29 | 33.776 | 102.168 | 3405.03 | | | 20/08 | 1.20 | 3403.83 |
| 30 | 33.794 | 102.129 | 3401 | | | 20/08 | 1.20 | 3400 |
| 31 | 33.815 | 102.117 | 3400 | | | 20/08 | 0.65 | 3399 |
| 32 | 33.817 | 102.080 | 3454.88 | 01/09/2019 | | 01/09 | 3.60 | 3451.28 |
| 33 | 33.866 | 101.983 | 3461.53 | 03/09/2019 | | 03/09 | 1.70 | 3459.83 |
| 34 | 33.884 | 101.927 | 3514.40 | 31/08/2019 | | 31/08 | 4.74 | 3509.66 |

(a)

(b)

(c)

(d)

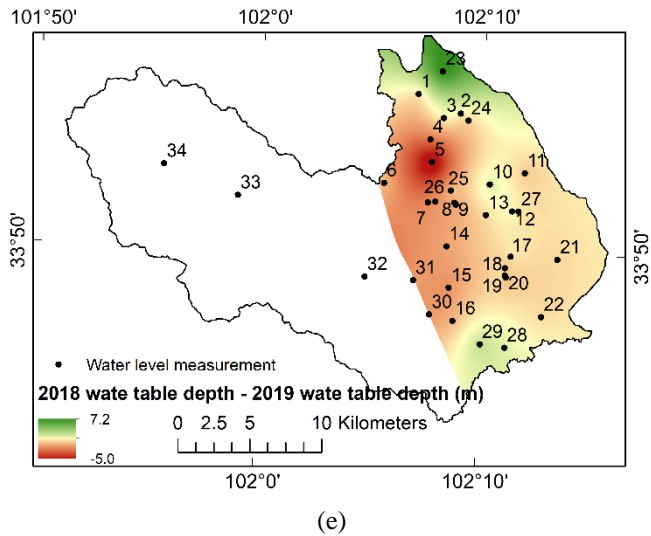

(e)

**Figure 8. (a) and (b) are water table depths (m) of east Maqu catchment in 2018 and 2019, respectively; (c) and (d) are piezometric heads (m a.s.l) of eastern Maqu catchment in 2018 and 2019, respectively; (e) is the difference (m) of water table depth between 2018 and 2019. Numbers from 1 to 34 indicate boreholes listed in Table 6.**

### 4.4.2 Aquifer tests

11 aquifer tests were conducted (Fig. 9) in unconfined aquifers, where the wells are partially penetrating. The pumping test data acquired from the borehole ITC_Maqu_1 were analyzed using the Boulton (1963) method as follows:

$$S_D = \frac{2\pi T(H-b)}{Q},$$ (4)

where $S_D$ is drawdown (m), $T$ is transmissivity (m²/d), $H$ is the average head along the saturated thickness (m), $b$ is the original saturated aquifer thickness (m), and $Q$ is pumping rate (m³/d).

Eight slug tests were done in boreholes numbered 16, 21, 24, 26, 27, 32, 33, 34 (Fig. 8) and the data were analyzed using the Bouwer and Rice (1976) method for hydraulic conductivity as follows:

$$K = \frac{r^2 \ln\left(\frac{R_e}{R}\right)}{2L} \cdot \frac{1}{t} \cdot \ln\left(\frac{h_0}{h_t}\right),$$ (5)

where $K$ is hydraulic conductivity (m/d), $r$ is the radius of the well casing (m), $R_e$ is the effective radial distance over which the head difference is dissipated (m), $R$ is radius measured from well center to undisturbed aquifer (m), $L$ is the length of the screen (m), $t$ is time (d), $h_0$ is the water level at time 0 (m), and $h_t$ is the water level at time t (m).

Another two pumping tests were carried out at borehole 6 and 23. The water level recovery data were analyzed using the Boulton and Agarwal method. Agarwal (1980) defines the recovery drawdown $S_r$ (m) as the difference between the head $h_p$ (m) at the end of the pumping period and the head $h$ (m) during the recovery period.

$$S_r = h - h_p,$$ (6)





The recovery time $t_r$ (d) is the time since the recovery started calculated as the difference between the duration of pumping $t_p$ (d) and the time $t$ (d) since pumping started.

$$t_r = t - t_p,\tag{7}$$

Data were processed automatically in AquiferTest software with assumptions made considering the average conditions in the study area: aquifer is unconfined and 35 m thick; well is partially penetrating; screen radius is 0.27 m; screen length is 15 m; the distance from aquifer top to screen bottom is 15 m; casing radius is 0.25 m; borehole radius is 0.3 m. As a result, the hydraulic conductivities range from 0.1 m/d to 15.6 m/d (Fig. 9 and Fig. A1). According to Healy et al. (2007), the hydraulic conductivity is roughly between 0.1 m/d - 100 m/d when the earth material changes from fine silty sand to coarse clean sand.

So the obtained hydraulic conductivities are acceptable. However, the slug test is likely to underestimate the hydraulic conductivity when the well is not used for a period of time. Compared to the slug test, the hydraulic conductivity obtained from the pumping test is more accurate and is a volumetric average, which makes it more suitable to calibrate $C_p$, because MRS results are also volumetric averages.

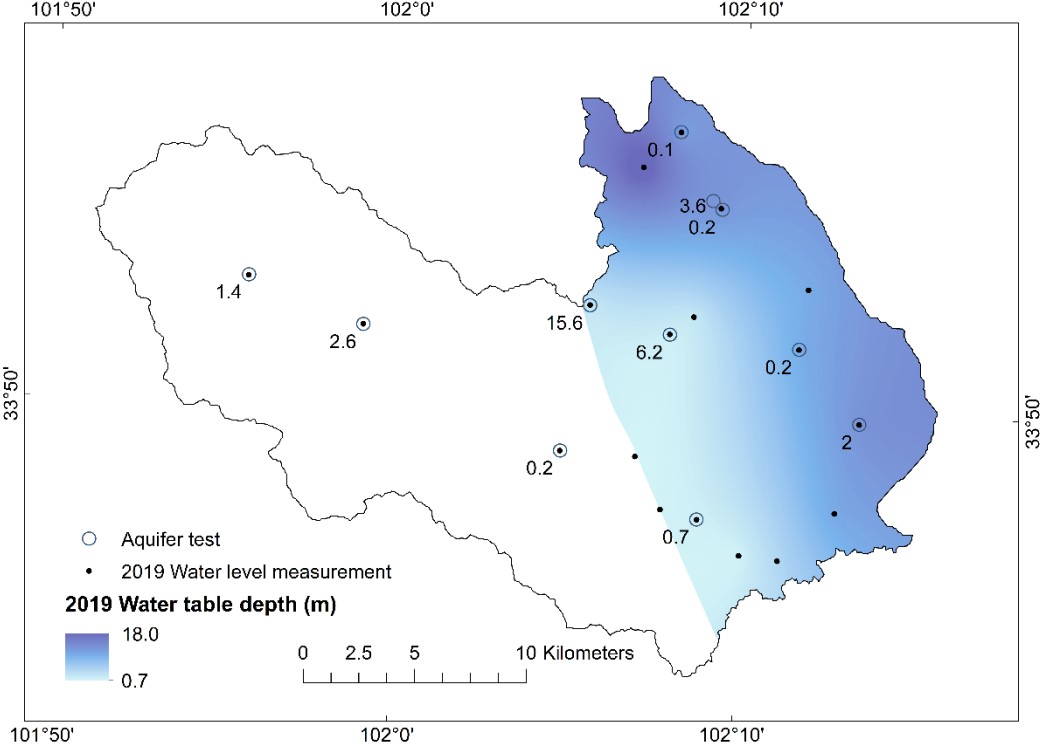

**Figure 9. Hydraulic conductivity (m/d) obtained from aquifer tests, east of Maqu catchment.**




## 4.5 Hydrogeophysical surveys

### 4.5.1 Magnetic susceptibility

The magnetic susceptibility measurements (Fig. 10) reveal very low susceptibility in the catchment with susceptibility values, all smaller than $1\times10^{-5}$ SI units with an average of $3\times10^{-6}$ SI units. A previous study from Chen et al. (1999) also reported
low magnetic susceptibility of the RH core (Fig. 5) with 120 m length located 40 km east of the study area in Ruoergai Basin. Thus, the low magnetic susceptibility ensured the suitability of applying MRS in the study area.

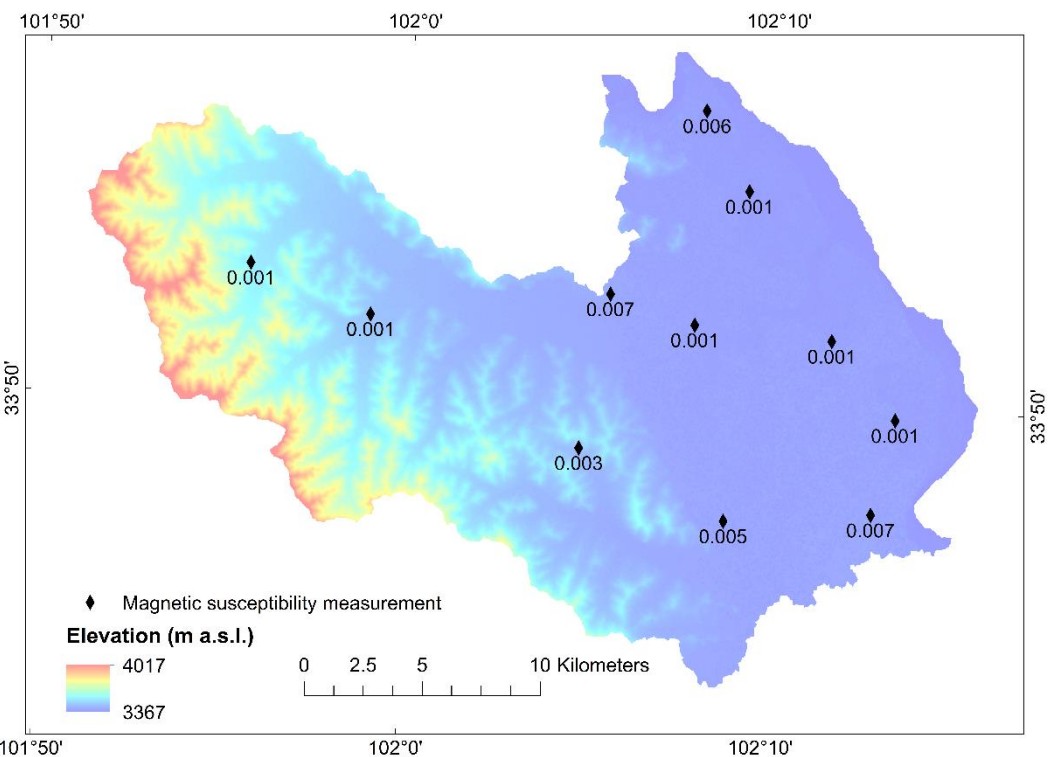

**Figure 10. Magnetic susceptibility measurements ($10^{-3}$ SI Units) ensured the suitability of applying MRS in the study area.**

### 4.5.2 ERT

Detailed information on ERT profiles and inversion parameters are listed in Table 7 and Table A3, respectively. The pseudosection plot in RES2DINV is useful for filtering out outlier data points, after which the least square method was used for the inversion. Results of ERT2 and ERT3 are shown in Fig. 11, and complete results are shown in Fig. A2 in the Appendix, with the root mean square (RMS) error less than 5%. A pattern of roughly regular parallel to surface electrostratigraphy is observed in all ERT profiles, except 0 m – 310 m of profile ERT5, where the pattern is dipping relatively to surface. This
means that strata are likely to be stratified in most parts of the study area. For ERT2, ERT3, ERT5, and ERT6, three electrostratigraphic layers can be identified: the first layer with the highest resistivity, the second layer with the lowest



resistivity, and the third layer with a medium resistivity. The second layer is likely to represent an aquifer. However, considering ERT4 and ERT7, there is a lack of marker electrostratum, i.e., a layer with high resistivity does not exist at the ground surface. This is probably due to high water content near the ground surface in the mountainous area where ERT4 and

ERT7 were located. As for ERT1, rainfall occurred during the field measurement. Rainwater accumulations occurred next to some of the electrodes, causing abnormal current distribution during the ERT measurements and about half of the data are missing in the filtering process. The ERT1 inversion results show a three – layers pattern similar to the one observed along the ERT2, ERT3, ERT5, and ERT6 profiles. One or more short wavelength anomalies (< 200 m) are observed along all profiles but particularly in the case of ERT1, ERT3 and ERT6. Short wavelength anomalies along ERT1 may be due to data acquisition

made during rainfall, while in the case of the other profiles, localized changes in water content or lithology variations are suspected.

Compared to the Dipole-Dipole configuration, the investigation depth of the Wenner configuration is deeper. So resistivity values obtained from Wenner configuration were used to establish geoelectrical models for MRS inversion. For ERT2, ERT3, ERT5, and ERT6, three-layer geoelectrical models were extracted, while for ERT4 and ERT7, two-layer geoelectrical models

were extracted. ERT1 was neglected due to the influence of rainfall. For ERT5, from 0 m to 310 m, there's a topographic change, the ground surface elevation decreases from 3395 m.a.s.l and stabilizes at around 3390 m.a.s.l. Ground surface with low resistivity exists along this 310 m transect. Since the MRS soundings were conducted in flat areas, so only resistivity from 310 m to 810 m was used for the first layer of the geoelectrical model. The geoelectrical models and corresponding MRS measurements are shown in Table A4. The depths of the last layer of geoelectrical models are extended to 1.5 times of the

MRS investigation depth since signal distortion due to subsurface resistivity is calculated down to that depth while making the MRS linear filter. In this particular version, MRS investigation depth was considered to be the MRS loop size, i.e., 150m and 100m. Nevertheless, like other geophysical methods, ERT has equivalence problems, i.e., non–uniqueness of inversion results. This can be better constrained with more information in the area, e.g., lithology and water content.

**Table 7. Detailed information on ERT.**

| Detailed information | | ERT1 | ERT2 | ERT3 | ERT4 | ERT5 | ERT6 | ERT7 |
|---|---|---|---|---|---|---|---|---|
| Length (m) | | 890 | 890 | 890 | 890 | 810 | 810 | 810 |
| Position (latitude°) (longitude°) | Start | 33.889 102.207 | 33.929 102.168 | 33.921 102.145 | 33.877 102.082 | 33.864 102.184 | 33.823 102.227 | 33.900 101.982 |
| | End | 33.881 102.209 | 33.925 102.160 | 33.918 102.136 | 33.881 102.074 | 33.860 102.191 | 33.822 102.218 | 33.903 101.990 |
| Orientation | | ES167° | SW242° | SW243° | WN307° | ES130° | SW261° | NE63° |




Earth System
Science
Data

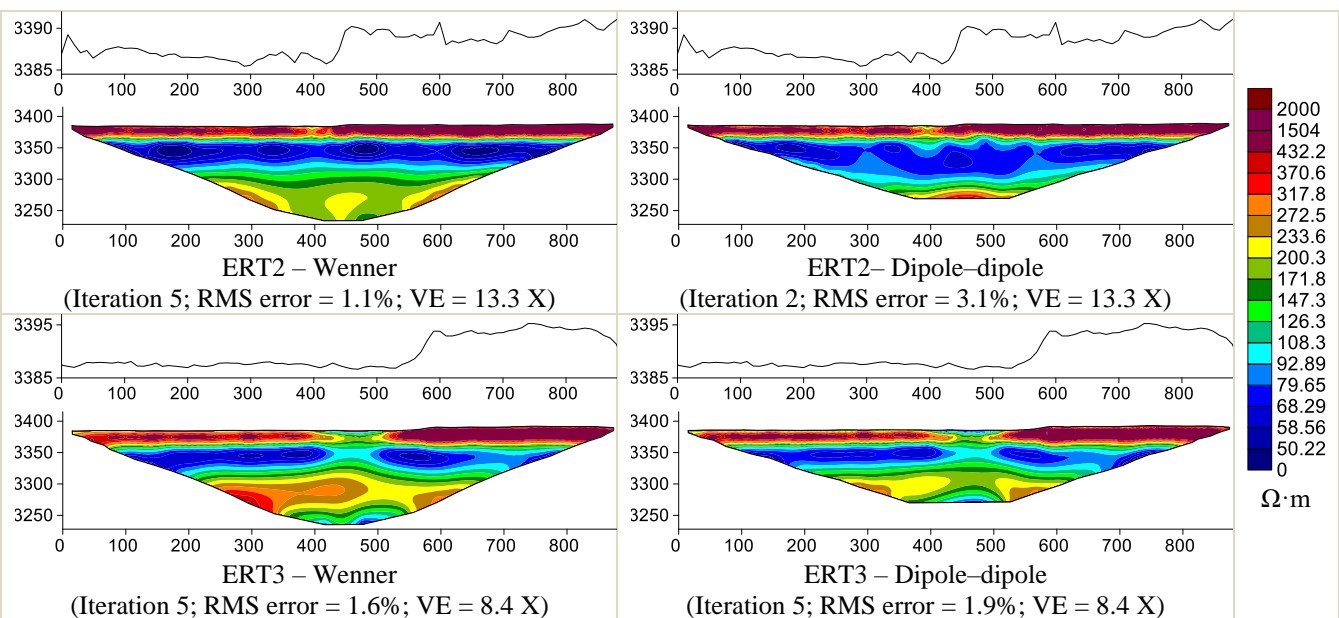

**Figure 11. ERT2 and ERT3 measurements and corresponding ground surface elevation and vertical exaggeration (VE).**

### 4.5.3 MRS

Alluvial deposits may be locally highly heterogeneous, but in the study area, they all have high permeability because they are braided river deposits. Besides, in the flat eastern area, there aren't big geographic or geomorphic variations, and the ERT

results suggest a roughly regular parallel stratification to surface electrostratigraphy. As such, generally horizontal aquifers are expected in the east, and we didn't use default inversion parameters because they sometimes result in abrupt changes or discontinuities of water content at two near MRS sounding sites. Some excitations were excluded during inversion based on S/N and the mismatch in terms of amplitude, Larmor frequency, and phase. The inversion parameters are listed in Table A5. The temperature of the water leads to different water densities and viscosities, and influences therefore also hydraulic

parameters. In Samovar V6.6, a default temperature of 20 °C is used. But in the study area, the average groundwater temperature is 6.2 °C. Therefore, it is necessary to take the groundwater temperature into account when estimating hydraulic parameters. Thus, based on the eq. 8, a correction factor of 0.69 was used during the inversion process to improve accuracy.

$$K = k\rho g/\eta, \tag{8}$$

Where $K$ is hydraulic conductivity (m/d), $k$ is the permeability of porous media (m$^2$), $\rho$ is water density (kg/m$^3$), $g$ is the

gravitational acceleration (m/s$^2$), and $\eta$ is water viscosity (Pa·s).

MRS3–1 was used to calculate the calibration coefficient $C_p$, because it is the nearest MRS sounding to the well (ITC_Maqu_1) for which pumping test data is available. Using a single point of calibration, the calibration coefficient $C_p$ can be estimated with the uncertainty ≤ 150% (Boucher et al., 2009). The calibrated $C_p$ is 8.78E–09 for $T_1$ and 8.13E–9 for $T_2^*$. Fig. 12 shows inversion results of water content and $T_1$ derived from MRS2–1, MRS3–1, and MRS3–2, and complete results are shown in

Fig. A3 in the Appendix. Except for MRS9–2, water mainly concentrates in upper layers, above the 60 meters depth. However,





still some of the in-situ water is missing on account of the depth and on account of the current 'window of the technique' sensitive to the larger pore fraction of the in-situ water.

Detailed results are listed in Table A6 in the Appendix, including $T_1$, $T_2^*$, water content, $T_1$ and $T_2^*$ derived hydraulic conductivities $K_{T1}$, $K_{T2*}$ and transmissivities $T_{T1}$, $T_{T2*}$. In the table, 0.00 and 1000.00 are invalid values for $T_2^*$. 0.00 and

3000.00 are invalid values for $T_1$. This un-determination of some parameters may be attributed to the hydrogeological conditions, such as highly heterogeneous lithology and too low signal/noise ratio, and may be eased using Samovar V11.4 which incorporates singular value decomposition. Nevertheless, in highly heterogeneous environments, the un-determination of some parameters may remain with current technology. According to Table A6, except for invalid values, $T_1$ derived hydraulic conductivity ($K_{T1}$) ranges from 0.00m/d to 19.64 m/d, $T_2^*$ derived hydraulic conductivity ($K_{T2*}$) ranges from 0.00

m/d to 210.98 m/d. An order of magnitude difference is observed between the range of $K_{T1}$ and the range of $K_{T2*}$, which may be due to the big difference between $T_1$ and $T_2^*$. Otherwise, more pumping test data are needed to further calibrate $C_p$. Derived hydraulic conductivity of 0.00 m/d is from the very low water content.

MRS has its own limitations in that the inversion involves equivalence problems, i.e., non–uniqueness of inversion results, and there is a decrease of resolution with depth. In this study, the most serious limitation is that part of the aquifer too deep for

the current technological performance of the MRS technique. Despite limitations observed, MRS does characterize non-invasively the subsurface hydrogeological properties. And there is no ambiguity in terms of quantifying the amount of free water (Lubczynski and Roy, 2003) compared to other hydrogeophysical methods. So information about the amount of free water is the most reliable result we could acquire from MRS. It is expected when more lithology and water content information becomes available in the area, and with the improvement of the MRS inversion technique, the results will become more

accurate.

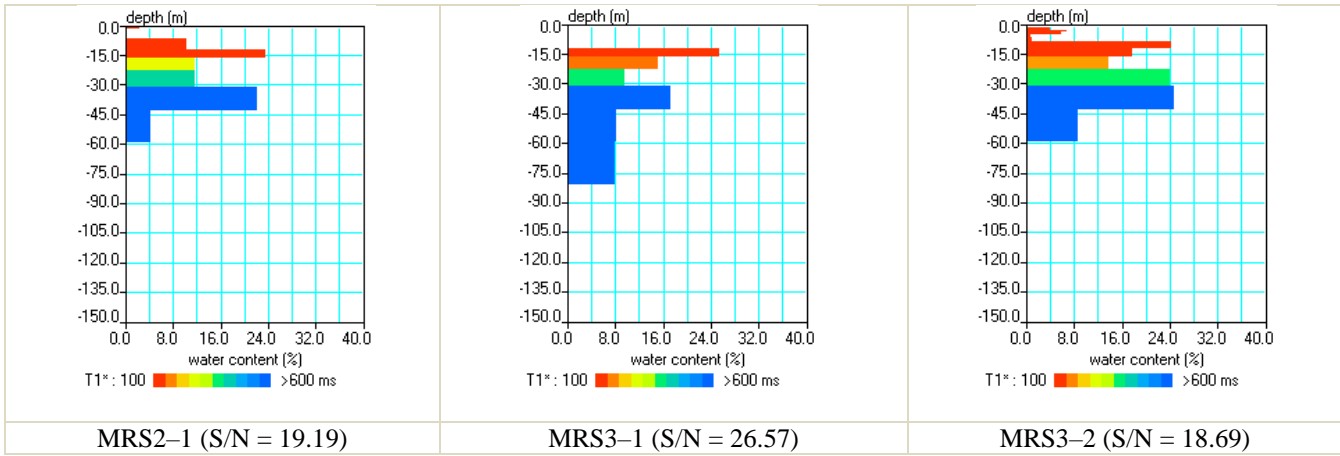

**Figure 12. Water content and $T_1$ derived from MRS2–1, MRS3–1, and MRS3–2.**



### 4.5.4 TEM

Detailed information of ten optimal TEM measurements and inversion parameters are listed in Table 8 and Table A7, respectively. The industrial noise filter was set at 50 Hz, and the amplifier was off. In the study area, using the square loop with a side length of 48 m or 95 m, the maximum time of 1 ms or 4 ms, stack between 5 – 10, and current of 0.8 A or 1.1 A, the TEM method can reach the maximum investigation depth ranging from 150 m to more than 1000 m. For data processing, the invalid data points in field data were first removed, then the field data were smoothed, and the initial model was constructed

based on apparent conductance S(h). After that, the process of the inverse problem solution was started. Induced polarization (IP) and superparamagnetic (SPM) effects were not considered in the inversion process. Because of the dead time and the fact that at most sites, a relatively dry layer of sediments exists near the ground surface with a corresponding high resistivity depth interval, the upper 15 m to 30 m of the sounding is lost, although subsequent layered earth modeling attempts filling the gap. The RMS error of the inversion results shown in Fig. 13 is below 2% in the flat area and below 10% in the mountainous area.

The results in the mountainous area, i.e., results of TEM6, TEM7, and TEM8, indicate that the resistivity becomes larger in the deep subsurface, and is consistent with our understanding that the bedrock is located at relatively shallow depth from the ground surface. The maximum investigation depth of TEM6 is shallow, only ten time windows were available and resulted in about 150 m investigation depth from the ground surface. This may be due to the local unknown geological condition. In addition to consolidated rock resistivity of the order of 2 kΩ·m to 4 kΩ·m, TEM7 and TEM8 responses may show instances of

fracturing, weathering or faulting so that several additional measurements will be needed in the future for confirmation.

The rest of the TEM measurements are scattered in the east where it is likely that lake deposits are covered by river deposits on the top. Because the clay silt lithology has a lower resistivity than sand-rich lithology, and Chen et al. (1999) suggested that the ancient lake in Ruoergai Basin was a freshwater or slightly saline lake for most of its life, the decrease of resistivity may indicate the change from river deposits to lake deposits. Table 9 listed the TEM derived depth of river deposits bottom in the

east. For TEM0, TEM1, TEM2, TEM3, TEM4, TEM9, the bottom of river deposits are deeper than 100 m, with lake deposits underneath. But for TEM5, the bottom of river deposits is at 50 m deep, followed by 64 m thick lake deposits, with the bedrock down most, and the nearest MRS sounding MRS6-1 indeed shows that there is no free water under 50 m depth.

**Table 8. Acquisition parameters of optimal TEM data.**

| Name | A side length of TEM loop (m) | Latitude (°) | Longitude (°) | Max Time (ms) | Stack | Adjustment of the high voltage protection system (μs) | Current in the transmitting loop (A) |
|---|---|---|---|---|---|---|---|
| TEM0 | 48 | 33.876 | 102.093 | 1 | 6 | 5 | 1.1 |
| TEM1 | 95 | 33.947 | 102.135 | 1 | 10 | 7 | 0.8 |
| TEM2 | 95 | 33.865 | 102.132 | 4 | 5 | 7 | 0.8 |
| TEM3 | 95 | 33.860 | 102.194 | 1 | 10 | 7 | 0.8 |
| TEM4 | 95 | 33.830 | 102.225 | 1 | 10 | 7 | 0.8 |
| TEM5 | 48 | 33.790 | 102.147 | 1 | 10 | 5 | 1.1 |
| TEM6 | 95 | 33.817 | 102.080 | 1 | 5 | 7 | 0.8 |
| TEM7 | 95 | 33.866 | 101.983 | 1 | 10 | 7 | 0.8 |
| TEM8 | 95 | 33.884 | 101.927 | 1 | 5 | 7 | 0.8 |
| TEM9 | 95 | 33.916 | 102.155 | 1 | 10 | 7 | 0.8 |

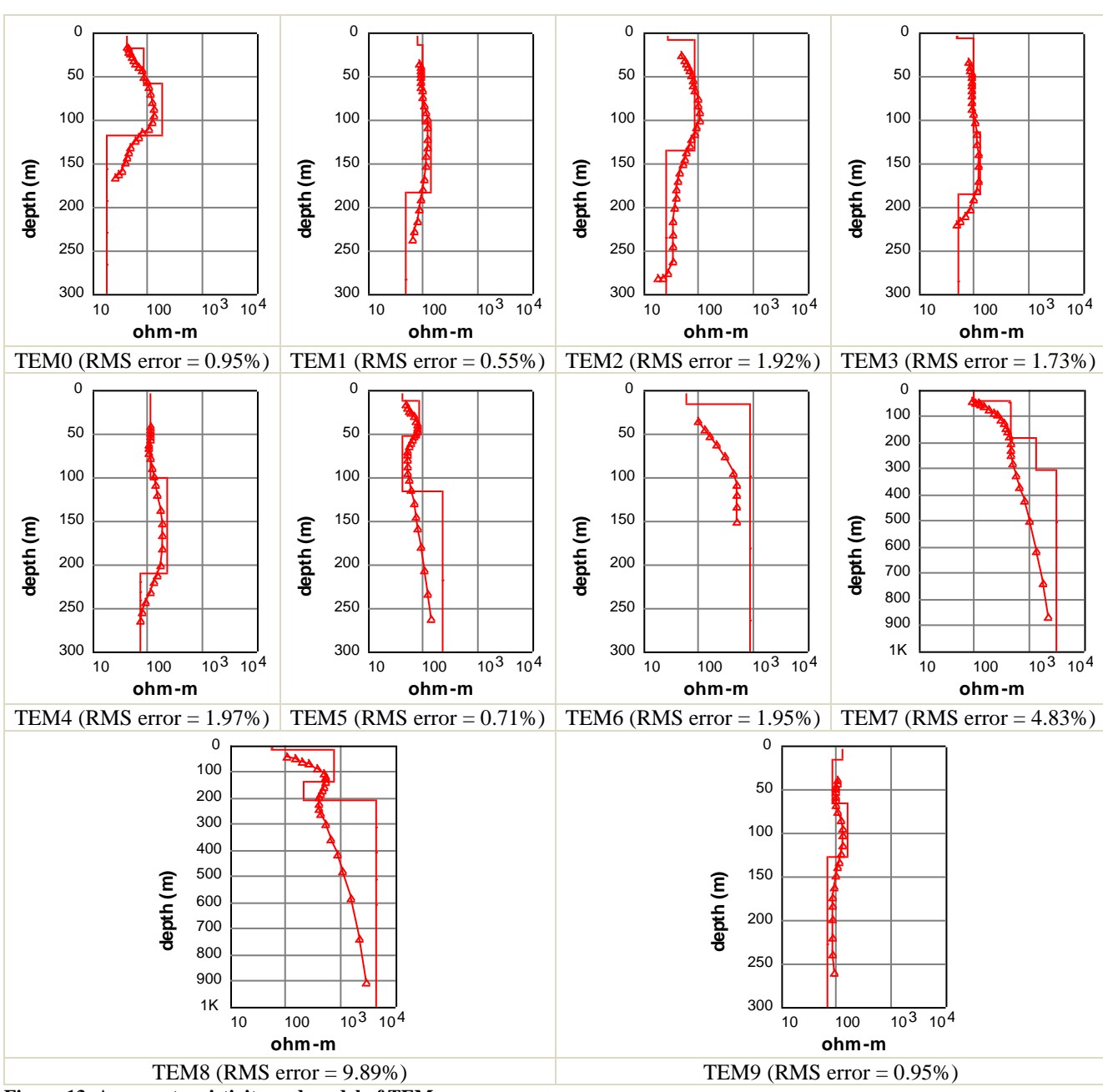

Figure 13. Apparent resistivity and model of TEM.



**Table 9. TEM derived depth of river deposits bottom in the east.**

| Name | Depth of river deposits bottom (m) |
|------|------------------------------------|
| TEM0 | 116 |
| TEM1 | 181 |
| TEM2 | 132 |
| TEM3 | 183 |
| TEM4 | 208 |
| TEM5 | 50 |
| TEM9 | 125 |

**5 Data availability**

The raw dataset is archived and freely available in the DANS repository under the link https://doi.org/10.17026/dans-z6t-zpn7
(Li et al., 2020).

**6 Conclusion**

In this study, we conducted borehole core lithology analysis, altitude survey, soil thickness measurement, hydrogeological
survey, and hydrogeophysical survey in the Maqu catchment of the Yellow River source region in the Tibetan Plateau, where
little aquifer data are available. The lithology is available in borehole ITC_Maqu_1, and it is mainly composed of sand. Seven
DEMs were evaluated based on measured elevation, ALOS RT1 and ALOS RT2 were proven to have the best overall
performance. ALOS RT1 is suggested to be used in future studies because of its slightly better performance and a higher
resolution than ALOS RT2. Soil thicknesses are within 1.2 m in most cases in the west, and the soil thickness decreases as the
slope increases based on soil thickness measurements. The hydrogeological survey reveals that groundwater flows from the
west to the east, recharging the Yellow River, and the hydraulic conductivity ranges from 0.2 m/d to 12.4 m/d. The
hydrogeophysical survey demonstrates the presence of an unconfined aquifer in the east, and water content and hydraulic
parameters were estimated at MRS sounding locations. The depth of the Yellow River deposits was derived at TEM sounding
positions in the flat eastern area. The raw data set is freely available at https://doi.org/10.17026/dans-z6t-zpn7 (Li et al., 2020).
Although water table depths were only measured once or twice, and hydrogeophysical methods, like ERT, TEM, and MRS,
have inherent non–uniqueness problems during the inversion process, they all provide valuable information, especially in the
data–scarce area. The data in this paper can be used for future set up of a hydrogeological conceptual model and groundwater
modeling which will be presented in follow up papers. To our knowledge, this is the first time to conduct such detailed surveys
in a TP catchment in order to set up a hydrogeological conceptual model. This paper is expected to contribute not only to the
hydrogeological conceptual model of the Maqu catchment over TP, but also to provide data for hydrogeological and
hydrogeophysical communities, and promote interdisciplinary research.





## Appendix A

### A1 Statistical analysis of seven DEMs in the flat eastern area and the mountainous western area

Table A1. Statistical analysis of seven DEMs in the flat eastern area.

|  | Min Error * (m) | Max Error (m) | Max Error –Min Error (m) | MAE (Mean Absolute Error) (m) | ME (Mean Error) (m) | Correlation coefficient | RMSE (m) |
|---|---|---|---|---|---|---|---|
| SRTM | 28 | 44 | 16 | 36.916 | 36.916 | 0.205 | 37.148 |
| ASTER V1 | –17 | 41 | 58 | 23.539 | 22.492 | 0.001 | 24.902 |
| ASTER V2 | –8 | 52 | 59 | 25.455 | 24.977 | 0.008 | 27.626 |
| ASTER V3 | 4 | 45 | 41 | 28.765 | 28.765 | 0.040 | 30.052 |
| AW3D30 | 27 | 43 | 17 | 37.522 | 37.522 | 0.086 | 37.788 |
| ALOS RT2 | –8 | 7 | 15 | 3.449 | 1.007 | 0.234 | 4.100 |
| ALOS RT1 | –8 | 8 | 16 | 3.394 | 0.947 | 0.216 | 4.145 |

* Error = DEM value – GPS-RTK value

Table A2. Statistical analysis of seven DEMs in the mountainous western area.

|  | Min Error * (m) | Max Error (m) | Max Error –Min Error (m) | MAE (Mean Absolute Error) (m) | ME (Mean Error) (m) | Correlation coefficient | RMSE (m) |
|---|---|---|---|---|---|---|---|
| SRTM | 22 | 42 | 20 | 31.862 | 31.862 | 0.985 | 32.660 |
| ASTER V1 | 3 | 43 | 39 | 27.862 | 27.862 | 0.956 | 30.381 |
| ASTER V2 | 10 | 55 | 45 | 32.631 | 32.631 | 0.945 | 35.828 |
| ASTER V3 | 13 | 42 | 28 | 29.554 | 29.554 | 0.967 | 31.396 |
| AW3D30 | 25 | 44 | 19 | 33.016 | 33.016 | 0.982 | 33.807 |
| ALOS RT2 | –13 | 8 | 21 | 7.494 | –3.753 | 0.984 | 8.489 |
| ALOS RT1 | –12 | 7 | 19 | 6.968 | –3.676 | 0.985 | 7.908 |

* Error = DEM value – GPS-RTK value


### A2 Aquifer tests data and derived hydraulic conductivity

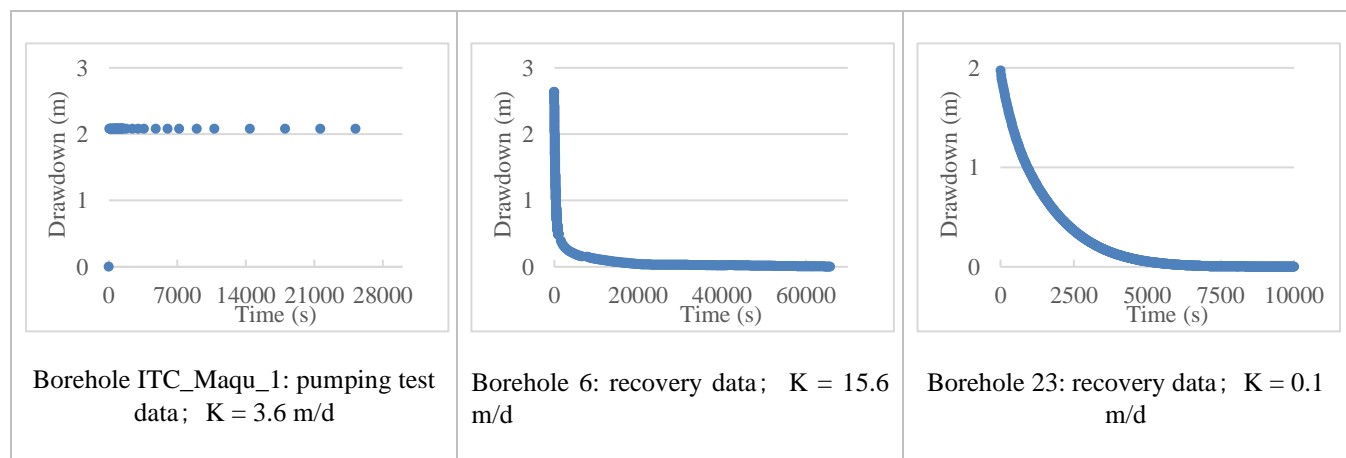

Borehole ITC_Maqu_1: pumping test data;  K = 3.6 m/d

Borehole 6: recovery data;  K = 15.6 m/d

Borehole 23: recovery data;  K = 0.1 m/d





Borehole 16: slug test data ;  K = 0.7 m/d

Borehole 21: slug test data ;  K = 2.0 m/d

Borehole 24: slug test data ; K = 0.2 m/d

Borehole 26: slug test data ;  K = 6.2 m/d

Borehole 27: slug test data ;  K = 0.2 m/d

Borehole 32: slug test data ;  K = 0.2 m/d

Borehole 33: slug test data; K = 2.6 m/d

Borehole 34: slug test data; K = 1.4 m/d

**Figure A1. Aquifer test data and derived hydraulic conductivity (K).**





**A3 Inversion parameters for ERT**

**Table A3. Inversion parameters for ERT.**

| Parameter | Value |
|---|---|
| Initial damping factor | 0.16 |
| Minimum damping factor | 0.015 |
| Convergence limit | 5 |
| The minimum change in RMS error | 0.4% |
| Number of iterations | 5 |
| Vertical to horizontal flatness filter ratio | 1 |
| Number of nodes between adjacent electrodes | 2 |
| Increasing of damping factor with depth | 1.05 |
| The thickness of the first layer | 0.5 m |
| Factor to increase thickness layer with depth | 1.1 |

**A4 Geoelectrical models used for MRS inversion**

**Table A4. Geoelectrical models used for MRS inversion.**

| MRS | ERT | Depth (m) from | Depth (m) to | Resistivity from Wenner configuration (ohm–m) |
|---|---|---|---|---|
| | | 0 | 20 | 526 |
| 2–1 | 2 | 20 | 75 | 86 |
| | | 75 | 225 | 185 |
| | | 0 | 25 | 385 |
| 3–1, 3–2, 4–1, 4–2 | 3 | 25 | 70 | 93 |
| | | 70 | 225 | 213 |
| 5–2 | 4 | 0 | 40 | 90 |
| | | 40 | 225 | 123 |
| | | 0 | 20 | 290 |
| 1–1, 5–1 8–1, 8–2, 11–1*, 11–2* | 5 | 20 | 70 | 97 |
| | | 70 | 225 | 127 |
| | | 0 | 20 | 441 |
| 6–1, 7–1, 7–2, 9–1, 9–2 | 6 | 20 | 60 | 81 |
| | | 60 | 225 | 193 |
| 10–1 | 7 | 0 | 20 | 99 |
| | | 20 | 225 | 323 |

* The depth of the third layer is 150m rather than 225m.



## 540  A5 ERT measurements and ground surface elevation

ERT1 – Wenner
(Iteration 5; RMS error = 1.32%; VE = 16.2 X)

ERT1 – Dipole–dipole
(Iteration 2; RMS error = 3.1%; VE = 16.2 X)

ERT2 – Wenner
(Iteration 5; RMS error = 1.1%; VE = 13.3 X)

ERT2– Dipole–dipole
(Iteration 2; RMS error = 3.1%; VE = 13.3 X)

ERT3 – Wenner
(Iteration 5; RMS error = 1.6%; VE = 8.4 X)

ERT3 – Dipole–dipole
(Iteration 5; RMS error = 1.9%; VE = 8.4 X)

ERT4 – Wenner
(Iteration 2; RMS error = 2.7%; VE = 6.3 X)

ERT4 – Dipole–dipole
(Iteration 1; RMS error = 4.4%; VE = 6.3 X)





**Figure A2.** ERT measurements and corresponding ground surface elevation and vertical exaggeration (VE).

**A6 Inversion parameters for MRS**

**Table A5.  Inversion parameters for MRS.**

| MRS | Latitude (°) | Longitude (°) | excluded excitation | Signal processing (200 ms) | | | | Inversion parameters | | Model layers |
|---|---|---|---|---|---|---|---|---|---|---|
| | | | | Running aver. filter | Notch filter (50Hz, narrow) | Notch band | Filt. Correction & Centre fixed | Regularization | | |
| | | | | | | | | $E, T_2^*$ | $T_1^*$ | |
| 1–1 | 33.893 | 102.205 | | 15 | | | | 20 | 1000 | 16 |
| 2–1 | 33.930 | 102.171 | | 10 | | | | 1000 | 500 | 15 |
| 3–1 | 33.923 | 102.149 | 1 | 15 | √ | 3.0 | | 500 | 500 | 16 |
| 3–2 | 33.922 | 102.144 | | 15 | | | | 500 | 500 | 16 |
| 4–1 | 33.916 | 102.135 | | 15 | √ | 3.0 | | 1000 | 500 | 16 |
| 4–2 | 33.919 | 102.124 | 2 | 15 | | | | 1000 | 500 | 15 |



| 5–1 | 33.869 | 102.123 | | 20 | | | | | 500 | 500 | 13 |
|---|---|---|---|---|---|---|---|---|---|---|---|
| 5–2 | 33.875 | 102.079 | 1, 16 | 11 | | | | | 500 | 500 | 14 |
| 6–1 | 33.799 | 102.129 | 12, 14, 15 | 15 | √ | 3.0 | | | 500 | 500 | 16 |
| 7–1 | 33.812 | 102.197 | | 15 | | | | | 500 | 500 | 16 |
| 7–2 | 33.822 | 102.230 | | 15 | √ | 3.0 | √ | | 1000 | 500 | 16 |
| 8–1 | 33.863 | 102.186 | | 15 | √ | 3.0 | | | 500 | 500 | 15 |
| 8–2 | 33.883 | 102.209 | 5, 10 | 15 | √ | 3.0 | | | 1000 | 500 | 15 |
| 9–1 | 33.816 | 102.165 | | 15 | √ | 3.0 | | | 1000 | 500 | 15 |
| 9–2 | 33.823 | 102.240 | 13 | 15 | | | | | 1000 | 500 | 16 |
| 10–1 | 33.901 | 101.983 | 16 | 15 | √ | 3.0 | √ | | 500 | 500 | 16 |
| 11–1 | 33.875 | 102.211 | | 15 | | | | | 500 | 500 | 16 |
| 11–2 | 33.860 | 102.164 | 1 | 15 | √ | 3.0 | | | 1000 | 500 | 14 |

## A7 MRS inversion results



MRS1–1 (S/N = 11.35)

MRS2–1 (S/N = 19.19)

MRS3–1 (S/N = 26.57)

MRS3–2 (S/N = 18.69)

MRS4–1 (S/N = 13.74)

MRS4–2 (S/N = 11.35)





MRS5–1 (S/N = 34.46)    MRS5–2 (S/N = 2.68)    MRS6–1 (S/N = 1.38)

MRS7–1 (S/N = 32.65)    MRS7–2 (S/N = 15.61)    MRS8–1 (S/N = 22.66)

MRS8–2 (S/N = 0.90)    MRS9–1 (S/N = 5.21)    MRS9–2 (S/N = 1.86)



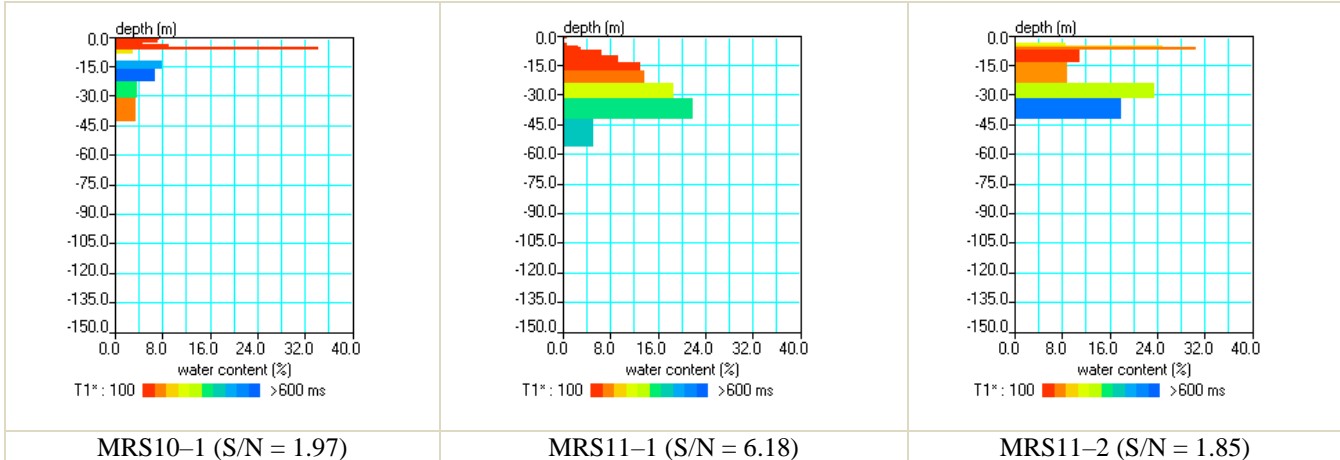

| MRS10–1 (S/N = 1.97) | MRS11–1 (S/N = 6.18) | MRS11–2 (S/N = 1.85) |

**Figure A3. Water content and T1 derived from MRS measurements.**

**Table A6. MRS inversion results.**

| MRS | Depth from (m) | Depth to (m) | $T_2^*$ (ms) | $T_1$ (ms) | Water content extrapol (%) | $K_{T2*}$ (m/d) | $T_{T2*}$ (m²/d) | $K_{T1}$ (m/d) | $T_{T1}$ (m²/d) |
|---|---|---|---|---|---|---|---|---|---|
| MRS1–1 | 0.00 | 1.00 | 0.00 | 0.00 | 0.00 | 0.00 | 0.00 | 0.00 | 0.00 |
| | 1.00 | 2.00 | 30.00 | 50.00 | 9.84 | 0.04 | 0.04 | 0.13 | 0.13 |
| | 2.00 | 3.00 | 0.00 | 0.00 | 0.00 | 0.00 | 0.00 | 0.00 | 0.00 |
| | 3.00 | 4.00 | 0.00 | 0.00 | 0.00 | 0.00 | 0.00 | 0.00 | 0.00 |
| | 4.00 | 5.00 | 0.00 | 0.00 | 0.00 | 0.00 | 0.00 | 0.00 | 0.00 |
| | 5.00 | 6.60 | 0.00 | 0.00 | 0.00 | 0.00 | 0.00 | 0.00 | 0.00 |
| | 6.60 | 9.00 | 55.20 | 50.00 | 10.63 | 0.16 | 0.38 | 0.14 | 0.34 |
| | 9.00 | 12.30 | 67.80 | 50.00 | 27.25 | 0.61 | 2.00 | 0.36 | 1.18 |
| | 12.30 | 16.80 | 45.00 | 65.50 | 12.18 | 0.12 | 0.54 | 0.27 | 1.24 |
| | 16.80 | 23.00 | 125.10 | 202.70 | 12.23 | 0.93 | 5.75 | 2.63 | 16.28 |
| | 23.00 | 31.50 | 56.90 | 372.00 | 13.38 | 0.21 | 1.78 | 9.69 | 82.13 |
| | 31.50 | 43.10 | 60.20 | 714.90 | 30.31 | 0.53 | 6.17 | 81.08 | 940.08 |
| | 43.10 | 58.90 | 147.80 | 895.90 | 3.15 | 0.33 | 5.28 | 13.25 | 210.28 |
| | 58.90 | 80.70 | 0.00 | 0.00 | 0.00 | 0.00 | 0.00 | 0.00 | 0.00 |
| | 80.70 | 110.40 | 0.00 | 0.00 | 0.00 | 0.00 | 0.00 | 0.00 | 0.00 |
| | 110.40 | 150.00 | 0.00 | 0.00 | 0.00 | 0.00 | 0.00 | 0.00 | 0.00 |
| MRS2–1 | 0.00 | 1.00 | 0.00 | 0.00 | 0.00 | 0.00 | 0.00 | 0.00 | 0.00 |
| | 1.00 | 2.00 | 30.00 | 50.00 | 2.25 | 0.01 | 0.01 | 0.03 | 0.03 |
| | 2.00 | 3.00 | 0.00 | 0.00 | 0.00 | 0.00 | 0.00 | 0.00 | 0.00 |
| | 3.00 | 4.00 | 0.00 | 0.00 | 0.00 | 0.00 | 0.00 | 0.00 | 0.00 |
| | 4.00 | 5.00 | 0.00 | 0.00 | 0.00 | 0.00 | 0.00 | 0.00 | 0.00 |
| | 5.00 | 6.60 | 0.00 | 0.00 | 0.00 | 0.00 | 0.00 | 0.00 | 0.00 |
| | 6.60 | 12.30 | 133.70 | 50.00 | 10.15 | 0.88 | 5.01 | 0.13 | 0.76 |

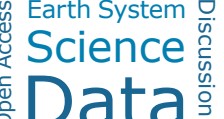

| | | | | | | | | |
|---|---|---|---|---|---|---|---|---|
| | 12.30 | 16.80 | 46.60 | 50.00 | 23.47 | 0.25 | 1.11 | 0.31 | 1.38 |
| | 16.80 | 23.00 | 57.00 | 240.40 | 11.75 | 0.18 | 1.15 | 3.55 | 22.03 |
| | 23.00 | 31.50 | 119.20 | 417.30 | 11.70 | 0.81 | 6.84 | 10.66 | 90.63 |
| | 31.50 | 43.10 | 47.80 | 803.80 | 22.22 | 0.25 | 2.85 | 75.14 | 871.64 |
| | 43.10 | 58.90 | 42.10 | 1274.40 | 4.14 | 0.04 | 0.56 | 35.22 | 556.55 |
| | 58.90 | 80.70 | 0.00 | 0.00 | 0.00 | 0.00 | 0.00 | 0.00 | 0.00 |
| | 80.70 | 110.40 | 0.00 | 0.00 | 0.00 | 0.00 | 0.00 | 0.00 | 0.00 |
| | 110.40 | 150.00 | 0.00 | 0.00 | 0.00 | 0.00 | 0.00 | 0.00 | 0.00 |
| MRS3–1 | 0.00 | 1.00 | 1000.00 | 50.00 | 3.20 | 15.52 | 15.52 | 0.04 | 0.04 |
| | 1.00 | 2.00 | 0.00 | 0.00 | 0.00 | 0.00 | 0.00 | 0.00 | 0.00 |
| | 2.00 | 3.00 | 0.00 | 0.00 | 0.00 | 0.00 | 0.00 | 0.00 | 0.00 |
| | 3.00 | 4.00 | 101.20 | 50.00 | 0.02 | 0.00 | 0.00 | 0.00 | 0.00 |
| | 4.00 | 5.00 | 74.40 | 50.00 | 0.03 | 0.00 | 0.00 | 0.00 | 0.00 |
| | 5.00 | 6.60 | 30.00 | 50.00 | 0.12 | 0.00 | 0.00 | 0.00 | 0.00 |
| | 6.60 | 9.00 | 30.00 | 50.00 | 0.07 | 0.00 | 0.00 | 0.00 | 0.00 |
| | 9.00 | 12.30 | 222.90 | 50.00 | 0.00 | 0.00 | 0.00 | 0.00 | 0.00 |
| | 12.30 | 16.80 | 65.80 | 50.00 | 25.34 | 0.53 | 2.39 | 0.33 | 1.50 |
| | 16.80 | 23.00 | 138.10 | 145.00 | 15.18 | 1.40 | 8.69 | 1.67 | 10.34 |
| | 23.00 | 31.50 | 393.60 | 376.90 | 9.49 | 7.12 | 60.51 | 7.05 | 59.76 |
| | 31.50 | 43.10 | 119.20 | 648.60 | 17.15 | 1.18 | 13.69 | 37.76 | 437.83 |
| | 43.10 | 58.90 | 58.10 | 744.20 | 8.21 | 0.13 | 2.12 | 23.79 | 377.49 |
| | 58.90 | 80.70 | 44.60 | 782.20 | 7.98 | 0.08 | 1.68 | 25.54 | 554.50 |
| | 80.70 | 110.40 | 0.00 | 0.00 | 0.00 | 0.00 | 0.00 | 0.00 | 0.00 |
| | 110.40 | 150.00 | 1000.00 | 50.00 | 0.00 | 0.00 | 0.00 | 0.00 | 0.00 |
| MRS3–2 | 0.00 | 1.00 | 0.00 | 0.00 | 0.00 | 0.00 | 0.00 | 0.00 | 0.00 |
| | 1.00 | 2.00 | 0.00 | 0.00 | 0.00 | 0.00 | 0.00 | 0.00 | 0.00 |
| | 2.00 | 3.00 | 63.90 | 50.00 | 3.99 | 0.08 | 0.08 | 0.05 | 0.05 |
| | 3.00 | 4.00 | 56.10 | 50.00 | 6.77 | 0.10 | 0.10 | 0.09 | 0.09 |
| | 4.00 | 5.00 | 66.10 | 50.00 | 5.83 | 0.12 | 0.12 | 0.08 | 0.08 |
| | 5.00 | 6.60 | 129.10 | 50.00 | 0.74 | 0.06 | 0.10 | 0.01 | 0.02 |
| | 6.60 | 9.00 | 663.70 | 50.00 | 0.93 | 1.99 | 4.78 | 0.01 | 0.03 |
| | 9.00 | 12.30 | 46.00 | 50.00 | 24.29 | 0.25 | 0.82 | 0.32 | 1.05 |
| | 12.30 | 16.80 | 80.80 | 50.00 | 17.68 | 0.56 | 2.52 | 0.23 | 1.05 |
| | 16.80 | 23.00 | 167.20 | 172.50 | 13.65 | 1.85 | 11.46 | 2.13 | 13.17 |
| | 23.00 | 31.50 | 106.50 | 365.00 | 24.07 | 1.32 | 11.24 | 16.78 | 142.22 |
| | 31.50 | 43.10 | 90.10 | 742.70 | 24.70 | 0.97 | 11.27 | 71.31 | 826.83 |
| | 43.10 | 58.90 | 94.90 | 1595.40 | 8.71 | 0.38 | 6.01 | 116.07 | 1841.47 |
| | 58.90 | 80.70 | 30.00 | 1608.80 | 0.01 | 0.00 | 0.00 | 0.10 | 2.11 |
| | 80.70 | 110.40 | 0.00 | 0.00 | 0.00 | 0.00 | 0.00 | 0.00 | 0.00 |
| | 110.40 | 150.00 | 0.00 | 0.00 | 0.00 | 0.00 | 0.00 | 0.00 | 0.00 |

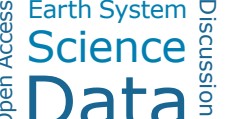

| | | | | | | | | |
|---|---|---|---|---|---|---|---|---|
| MRS4–1 | 0.00 | 1.00 | 81.10 | 50.00 | 3.07 | 0.10 | 0.10 | 0.04 | 0.04 |
| | 1.00 | 2.00 | 62.00 | 50.00 | 2.66 | 0.05 | 0.05 | 0.03 | 0.03 |
| | 2.00 | 3.00 | 0.00 | 0.00 | 0.00 | 0.00 | 0.00 | 0.00 | 0.00 |
| | 3.00 | 4.00 | 0.00 | 0.00 | 0.00 | 0.00 | 0.00 | 0.00 | 0.00 |
| | 4.00 | 5.00 | 0.00 | 0.00 | 0.00 | 0.00 | 0.00 | 0.00 | 0.00 |
| | 5.00 | 6.60 | 0.00 | 0.00 | 0.00 | 0.00 | 0.00 | 0.00 | 0.00 |
| | 6.60 | 9.00 | 0.00 | 0.00 | 0.00 | 0.00 | 0.00 | 0.00 | 0.00 |
| | 9.00 | 12.30 | 1000.00 | 50.00 | 0.46 | 2.22 | 7.34 | 0.01 | 0.02 |
| | 12.30 | 16.80 | 160.70 | 50.00 | 5.39 | 0.67 | 3.03 | 0.07 | 0.32 |
| | 16.80 | 23.00 | 254.80 | 50.00 | 9.27 | 2.92 | 18.08 | 0.12 | 0.75 |
| | 23.00 | 31.50 | 208.00 | 253.90 | 14.00 | 2.93 | 24.93 | 4.72 | 40.02 |
| | 31.50 | 43.10 | 170.80 | 467.20 | 12.38 | 1.75 | 20.29 | 14.14 | 163.90 |
| | 43.10 | 58.90 | 115.80 | 594.00 | 7.50 | 0.49 | 7.70 | 13.85 | 219.71 |
| | 58.90 | 80.70 | 75.60 | 675.70 | 5.51 | 0.15 | 3.33 | 13.18 | 286.11 |
| | 80.70 | 110.40 | 73.60 | 725.90 | 5.03 | 0.13 | 3.92 | 13.86 | 411.77 |
| | 110.40 | 150.00 | 79.10 | 747.30 | 4.88 | 0.15 | 5.86 | 14.28 | 566.03 |
| MRS4–2 | 0.00 | 1.00 | 757.10 | 294.20 | 5.37 | 14.91 | 14.91 | 2.43 | 2.43 |
| | 1.00 | 2.00 | 91.10 | 240.90 | 10.27 | 0.41 | 0.41 | 3.12 | 3.12 |
| | 2.00 | 3.00 | 78.80 | 149.70 | 4.57 | 0.14 | 0.14 | 0.54 | 0.54 |
| | 3.00 | 4.00 | 289.60 | 50.00 | 3.36 | 1.36 | 1.36 | 0.04 | 0.04 |
| | 4.00 | 5.00 | 0.00 | 0.00 | 0.00 | 0.00 | 0.00 | 0.00 | 0.00 |
| | 5.00 | 6.60 | 1000.00 | 50.00 | 0.18 | 0.86 | 1.38 | 0.00 | 0.00 |
| | 6.60 | 9.00 | 98.90 | 50.00 | 5.64 | 0.27 | 0.64 | 0.07 | 0.18 |
| | 9.00 | 12.30 | 169.90 | 50.00 | 2.99 | 0.42 | 1.38 | 0.04 | 0.13 |
| | 12.30 | 16.80 | 36.00 | 50.00 | 26.79 | 0.17 | 0.76 | 0.35 | 1.58 |
| | 16.80 | 31.50 | 139.10 | 248.90 | 6.21 | 0.58 | 8.56 | 2.01 | 29.61 |
| | 31.50 | 43.10 | 48.50 | 1139.20 | 17.87 | 0.20 | 2.36 | 121.37 | 1407.91 |
| | 43.10 | 58.90 | 30.00 | 1798.40 | 2.12 | 0.01 | 0.15 | 35.95 | 568.03 |
| | 58.90 | 80.70 | 0.00 | 0.00 | 0.00 | 0.00 | 0.00 | 0.00 | 0.00 |
| | 80.70 | 110.40 | 0.00 | 0.00 | 0.00 | 0.00 | 0.00 | 0.00 | 0.00 |
| | 110.40 | 150.00 | 0.00 | 0.00 | 0.00 | 0.00 | 0.00 | 0.00 | 0.00 |
| MRS5–1 | 0.00 | 1.00 | 0.00 | 0.00 | 0.00 | 0.00 | 0.00 | 0.00 | 0.00 |
| | 1.00 | 2.00 | 0.00 | 0.00 | 0.00 | 0.00 | 0.00 | 0.00 | 0.00 |
| | 2.00 | 3.00 | 0.00 | 0.00 | 0.00 | 0.00 | 0.00 | 0.00 | 0.00 |
| | 3.00 | 4.00 | 0.00 | 0.00 | 0.00 | 0.00 | 0.00 | 0.00 | 0.00 |
| | 4.00 | 5.00 | 0.00 | 0.00 | 0.00 | 0.00 | 0.00 | 0.00 | 0.00 |
| | 5.00 | 9.00 | 168.40 | 50.00 | 28.75 | 3.95 | 15.80 | 0.38 | 1.50 |
| | 9.00 | 12.30 | 166.00 | 50.00 | 26.56 | 3.55 | 11.70 | 0.35 | 1.15 |
| | 12.30 | 16.80 | 112.40 | 179.60 | 14.43 | 0.88 | 3.97 | 2.43 | 10.96 |
| | 16.80 | 43.10 | 129.40 | 1666.40 | 14.52 | 1.18 | 30.97 | 210.98 | 5548.77 |





|  |  |  |  |  |  |  |  |  |
|---|---|---|---|---|---|---|---|---|
|  | 43.10 | 58.90 | 0.00 | 0.00 | 0.00 | 0.00 | 0.00 | 0.00 | 0.00 |
|  | 58.90 | 80.70 | 0.00 | 0.00 | 0.00 | 0.00 | 0.00 | 0.00 | 0.00 |
|  | 80.70 | 110.40 | 0.00 | 0.00 | 0.00 | 0.00 | 0.00 | 0.00 | 0.00 |
|  | 110.40 | 150.00 | 0.00 | 0.00 | 0.00 | 0.00 | 0.00 | 0.00 | 0.00 |
| MRS5–2 | 0.00 | 1.00 | 0.00 | 0.00 | 0.00 | 0.00 | 0.00 | 0.00 | 0.00 |
|  | 1.00 | 2.00 | 0.00 | 0.00 | 0.00 | 0.00 | 0.00 | 0.00 | 0.00 |
|  | 2.00 | 3.00 | 0.00 | 0.00 | 0.00 | 0.00 | 0.00 | 0.00 | 0.00 |
|  | 3.00 | 4.00 | 0.00 | 0.00 | 0.00 | 0.00 | 0.00 | 0.00 | 0.00 |
|  | 4.00 | 5.00 | 0.00 | 0.00 | 0.00 | 0.00 | 0.00 | 0.00 | 0.00 |
|  | 5.00 | 6.60 | 30.00 | 50.00 | 2.08 | 0.01 | 0.01 | 0.03 | 0.04 |
|  | 6.60 | 9.00 | 30.00 | 50.00 | 5.83 | 0.03 | 0.06 | 0.08 | 0.18 |
|  | 9.00 | 12.30 | 0.00 | 0.00 | 0.00 | 0.00 | 0.00 | 0.00 | 0.00 |
|  | 12.30 | 23.00 | 198.90 | 50.00 | 11.21 | 2.15 | 22.99 | 0.15 | 1.57 |
|  | 23.00 | 43.10 | 204.40 | 3000.00 | 12.31 | 2.49 | 50.09 | 579.99 | 11657.71 |
|  | 43.10 | 58.90 | 1000.00 | 885.20 | 10.05 | 48.67 | 768.94 | 41.19 | 650.86 |
|  | 58.90 | 80.70 | 0.00 | 0.00 | 0.00 | 0.00 | 0.00 | 0.00 | 0.00 |
|  | 80.70 | 110.40 | 0.00 | 0.00 | 0.00 | 0.00 | 0.00 | 0.00 | 0.00 |
|  | 110.40 | 150.00 | 0.00 | 0.00 | 0.00 | 0.00 | 0.00 | 0.00 | 0.00 |
| MRS6–1 | 0.00 | 1.00 | 0.00 | 0.00 | 0.00 | 0.00 | 0.00 | 0.00 | 0.00 |
|  | 1.00 | 2.00 | 0.00 | 0.00 | 0.00 | 0.00 | 0.00 | 0.00 | 0.00 |
|  | 2.00 | 3.00 | 0.00 | 0.00 | 0.00 | 0.00 | 0.00 | 0.00 | 0.00 |
|  | 3.00 | 4.00 | 1000.00 | 50.00 | 2.67 | 12.91 | 12.91 | 0.03 | 0.03 |
|  | 4.00 | 5.00 | 1000.00 | 67.70 | 7.47 | 36.18 | 36.18 | 0.18 | 0.18 |
|  | 5.00 | 6.60 | 490.70 | 165.00 | 16.84 | 19.64 | 31.42 | 2.40 | 3.75 |
|  | 6.60 | 9.00 | 136.10 | 334.90 | 21.46 | 1.93 | 4.62 | 12.59 | 30.45 |
|  | 9.00 | 12.30 | 30.00 | 872.30 | 12.48 | 0.05 | 0.18 | 49.70 | 164.38 |
|  | 12.30 | 16.80 | 152.00 | 2859.80 | 1.68 | 0.19 | 0.85 | 71.88 | 325.33 |
|  | 16.80 | 23.00 | 0.00 | 0.00 | 0.00 | 0.00 | 0.00 | 0.00 | 0.00 |
|  | 23.00 | 31.50 | 61.20 | 3000.00 | 4.28 | 0.08 | 0.66 | 201.54 | 1707.85 |
|  | 31.50 | 43.10 | 0.00 | 0.00 | 0.00 | 0.00 | 0.00 | 0.00 | 0.00 |
|  | 43.10 | 58.90 | 0.00 | 0.00 | 0.00 | 0.00 | 0.00 | 0.00 | 0.00 |
|  | 58.90 | 80.70 | 0.00 | 0.00 | 0.00 | 0.00 | 0.00 | 0.00 | 0.00 |
|  | 80.70 | 110.40 | 0.00 | 0.00 | 0.00 | 0.00 | 0.00 | 0.00 | 0.00 |
|  | 110.40 | 150.00 | 0.00 | 0.00 | 0.00 | 0.00 | 0.00 | 0.00 | 0.00 |
| MRS7–1 | 0.00 | 1.00 | 0.00 | 0.00 | 0.00 | 0.00 | 0.00 | 0.00 | 0.00 |
|  | 1.00 | 2.00 | 31.30 | 50.00 | 3.91 | 0.02 | 0.02 | 0.05 | 0.05 |
|  | 2.00 | 3.00 | 30.00 | 50.00 | 8.56 | 0.04 | 0.04 | 0.11 | 0.11 |
|  | 3.00 | 4.00 | 42.20 | 50.00 | 6.63 | 0.06 | 0.06 | 0.09 | 0.09 |
|  | 4.00 | 5.00 | 0.00 | 0.00 | 0.00 | 0.00 | 0.00 | 0.00 | 0.00 |
|  | 5.00 | 6.60 | 0.00 | 0.00 | 0.00 | 0.00 | 0.00 | 0.00 | 0.00 |





|  | | | | | | | | | |
|---|---|---|---|---|---|---|---|---|---|
|  | 6.60 | 9.00 | 1000.00 | 50.00 | 2.22 | 10.75 | 25.79 | 0.03 | 0.07 |
|  | 9.00 | 12.30 | 110.00 | 50.00 | 15.57 | 0.91 | 3.01 | 0.20 | 0.67 |
|  | 12.30 | 16.80 | 51.10 | 50.00 | 22.97 | 0.29 | 1.31 | 0.30 | 1.36 |
|  | 16.80 | 23.00 | 114.00 | 50.00 | 13.79 | 0.87 | 5.38 | 0.18 | 1.12 |
|  | 23.00 | 31.50 | 347.10 | 215.80 | 18.17 | 10.61 | 90.16 | 4.43 | 37.53 |
|  | 31.50 | 43.10 | 177.50 | 424.70 | 28.90 | 4.41 | 51.17 | 27.28 | 316.35 |
|  | 43.10 | 58.90 | 73.40 | 477.30 | 31.78 | 0.83 | 13.11 | 37.90 | 601.29 |
|  | 58.90 | 80.70 | 96.30 | 434.40 | 17.70 | 0.80 | 17.33 | 17.48 | 379.47 |
|  | 80.70 | 110.40 | 583.80 | 415.50 | 7.35 | 12.13 | 360.34 | 6.64 | 197.23 |
|  | 110.40 | 150.00 | 1000.00 | 412.70 | 4.80 | 23.26 | 920.91 | 4.28 | 169.67 |
| MRS7–2 | 0.00 | 1.00 | 0.00 | 0.00 | 0.00 | 0.00 | 0.00 | 0.00 | 0.00 |
|  | 1.00 | 2.00 | 0.00 | 0.00 | 0.00 | 0.00 | 0.00 | 0.00 | 0.00 |
|  | 2.00 | 3.00 | 0.00 | 0.00 | 0.00 | 0.00 | 0.00 | 0.00 | 0.00 |
|  | 3.00 | 4.00 | 0.00 | 0.00 | 0.00 | 0.00 | 0.00 | 0.00 | 0.00 |
|  | 4.00 | 5.00 | 30.00 | 50.00 | 2.60 | 0.01 | 0.01 | 0.03 | 0.03 |
|  | 5.00 | 6.60 | 33.90 | 50.00 | 12.48 | 0.07 | 0.11 | 0.16 | 0.26 |
|  | 6.60 | 9.00 | 70.70 | 50.00 | 1.80 | 0.04 | 0.10 | 0.02 | 0.06 |
|  | 9.00 | 12.30 | 235.30 | 50.00 | 9.60 | 2.57 | 8.49 | 0.13 | 0.42 |
|  | 12.30 | 16.80 | 177.50 | 50.00 | 20.48 | 3.13 | 14.07 | 0.27 | 1.21 |
|  | 16.80 | 23.00 | 95.50 | 50.00 | 16.44 | 0.73 | 4.50 | 0.22 | 1.33 |
|  | 23.00 | 31.50 | 105.80 | 276.70 | 23.28 | 1.26 | 10.73 | 9.33 | 79.06 |
|  | 31.50 | 43.10 | 144.90 | 326.40 | 29.58 | 3.01 | 34.91 | 16.49 | 191.24 |
|  | 43.10 | 58.90 | 158.90 | 332.60 | 18.42 | 2.25 | 35.60 | 10.67 | 169.25 |
|  | 58.90 | 80.70 | 181.90 | 385.10 | 11.98 | 1.92 | 41.87 | 9.30 | 201.85 |
|  | 80.70 | 110.40 | 244.10 | 435.20 | 10.57 | 3.05 | 90.59 | 10.47 | 311.14 |
|  | 110.40 | 150.00 | 372.10 | 459.20 | 10.32 | 6.92 | 274.04 | 11.39 | 451.46 |
| MRS8–1 | 0.00 | 1.00 | 30.00 | 50.00 | 7.08 | 0.03 | 0.03 | 0.09 | 0.09 |
|  | 1.00 | 2.00 | 30.00 | 50.00 | 1.57 | 0.01 | 0.01 | 0.02 | 0.02 |
|  | 2.00 | 3.00 | 0.00 | 0.00 | 0.00 | 0.00 | 0.00 | 0.00 | 0.00 |
|  | 3.00 | 4.00 | 0.00 | 0.00 | 0.00 | 0.00 | 0.00 | 0.00 | 0.00 |
|  | 4.00 | 5.00 | 0.00 | 0.00 | 0.00 | 0.00 | 0.00 | 0.00 | 0.00 |
|  | 5.00 | 6.60 | 0.00 | 0.00 | 0.00 | 0.00 | 0.00 | 0.00 | 0.00 |
|  | 6.60 | 9.00 | 106.20 | 50.00 | 7.50 | 0.41 | 0.98 | 0.10 | 0.24 |
|  | 9.00 | 12.30 | 30.00 | 50.00 | 11.66 | 0.05 | 0.17 | 0.15 | 0.50 |
|  | 12.30 | 23.00 | 141.50 | 50.00 | 9.53 | 0.92 | 9.90 | 0.12 | 1.33 |
|  | 23.00 | 31.50 | 84.90 | 288.60 | 19.51 | 0.68 | 5.79 | 8.51 | 72.31 |
|  | 31.50 | 43.10 | 172.70 | 683.20 | 12.46 | 1.80 | 20.88 | 30.44 | 353.13 |
|  | 43.10 | 58.90 | 45.50 | 1421.80 | 8.72 | 0.09 | 1.38 | 92.23 | 1457.22 |
|  | 58.90 | 80.70 | 0.00 | 0.00 | 0.00 | 0.00 | 0.00 | 0.00 | 0.00 |
|  | 80.70 | 110.40 | 0.00 | 0.00 | 0.00 | 0.00 | 0.00 | 0.00 | 0.00 |



| | 110.40 | 150.00 | 0.00 | 0.00 | 0.00 | 0.00 | 0.00 | 0.00 | 0.00 |
|---|---|---|---|---|---|---|---|---|---|
| MRS8–2 | 0.00 | 1.00 | 0.00 | 0.00 | 0.00 | 0.00 | 0.00 | 0.00 | 0.00 |
| | 1.00 | 2.00 | 0.00 | 0.00 | 0.00 | 0.00 | 0.00 | 0.00 | 0.00 |
| | 2.00 | 3.00 | 0.00 | 0.00 | 0.00 | 0.00 | 0.00 | 0.00 | 0.00 |
| | 3.00 | 4.00 | 0.00 | 0.00 | 0.00 | 0.00 | 0.00 | 0.00 | 0.00 |
| | 4.00 | 5.00 | 30.00 | 50.00 | 20.10 | 0.09 | 0.09 | 0.26 | 0.26 |
| | 5.00 | 6.60 | 0.00 | 0.00 | 0.00 | 0.00 | 0.00 | 0.00 | 0.00 |
| | 6.60 | 9.00 | 0.00 | 0.00 | 0.00 | 0.00 | 0.00 | 0.00 | 0.00 |
| | 9.00 | 12.30 | 472.90 | 50.00 | 7.67 | 8.31 | 27.41 | 0.10 | 0.33 |
| | 12.30 | 23.00 | 30.00 | 50.00 | 16.64 | 0.07 | 0.78 | 0.22 | 2.33 |
| | 23.00 | 31.50 | 61.00 | 501.30 | 36.77 | 0.66 | 5.63 | 48.37 | 411.11 |
| | 31.50 | 43.10 | 46.00 | 3000.00 | 12.13 | 0.12 | 1.44 | 571.34 | 6627.56 |
| | 43.10 | 58.90 | 89.50 | 3000.00 | 0.12 | 0.00 | 0.08 | 5.82 | 91.96 |
| | 58.90 | 80.70 | 0.00 | 0.00 | 0.00 | 0.00 | 0.00 | 0.00 | 0.00 |
| | 80.70 | 110.40 | 1000.00 | 3000.00 | 0.27 | 1.31 | 38.82 | 12.71 | 377.45 |
| | 110.40 | 150.00 | 1000.00 | 3000.00 | 0.46 | 2.21 | 87.61 | 21.51 | 851.96 |
| MRS9–1 | 0.00 | 1.00 | 0.00 | 0.00 | 0.00 | 0.00 | 0.00 | 0.00 | 0.00 |
| | 1.00 | 2.00 | 77.80 | 200.70 | 1.55 | 0.05 | 0.05 | 0.33 | 0.33 |
| | 2.00 | 3.00 | 100.10 | 194.30 | 8.55 | 0.42 | 0.42 | 1.69 | 1.69 |
| | 3.00 | 4.00 | 135.40 | 172.30 | 16.88 | 1.50 | 1.50 | 2.62 | 2.62 |
| | 4.00 | 5.00 | 119.00 | 136.20 | 21.18 | 1.45 | 1.45 | 2.06 | 2.06 |
| | 5.00 | 6.60 | 108.30 | 97.80 | 26.89 | 1.53 | 2.44 | 1.35 | 2.15 |
| | 6.60 | 9.00 | 178.40 | 50.00 | 26.76 | 4.13 | 9.90 | 0.35 | 0.84 |
| | 9.00 | 12.30 | 151.60 | 50.00 | 19.09 | 2.13 | 7.01 | 0.25 | 0.82 |
| | 12.30 | 16.80 | 54.70 | 50.00 | 20.19 | 0.29 | 1.32 | 0.26 | 1.19 |
| | 16.80 | 31.50 | 185.30 | 194.80 | 10.32 | 1.72 | 25.24 | 2.05 | 30.15 |
| | 31.50 | 43.10 | 107.10 | 839.60 | 37.63 | 2.09 | 24.25 | 138.81 | 1610.20 |
| | 43.10 | 58.90 | 30.00 | 1165.00 | 1.21 | 0.01 | 0.08 | 8.58 | 135.62 |
| | 58.90 | 80.70 | 0.00 | 0.00 | 0.00 | 0.00 | 0.00 | 0.00 | 0.00 |
| | 80.70 | 110.40 | 0.00 | 0.00 | 0.00 | 0.00 | 0.00 | 0.00 | 0.00 |
| | 110.40 | 150.00 | 0.00 | 0.00 | 0.00 | 0.00 | 0.00 | 0.00 | 0.00 |
| MRS9–2 | 0.00 | 1.00 | 57.70 | 569.30 | 27.56 | 0.44 | 0.44 | 46.75 | 46.75 |
| | 1.00 | 2.00 | 196.70 | 630.50 | 5.64 | 1.06 | 1.06 | 11.74 | 11.74 |
| | 2.00 | 3.00 | 0.00 | 0.00 | 0.00 | 0.00 | 0.00 | 0.00 | 0.00 |
| | 3.00 | 4.00 | 0.00 | 0.00 | 0.00 | 0.00 | 0.00 | 0.00 | 0.00 |
| | 4.00 | 5.00 | 0.00 | 0.00 | 0.00 | 0.00 | 0.00 | 0.00 | 0.00 |
| | 5.00 | 6.60 | 0.00 | 0.00 | 0.00 | 0.00 | 0.00 | 0.00 | 0.00 |
| | 6.60 | 9.00 | 0.00 | 0.00 | 0.00 | 0.00 | 0.00 | 0.00 | 0.00 |
| | 9.00 | 12.30 | 30.00 | 727.20 | 12.05 | 0.05 | 0.17 | 33.33 | 110.25 |
| | 12.30 | 16.80 | 34.70 | 400.90 | 11.97 | 0.07 | 0.31 | 10.07 | 45.59 |





| | | | | | | | | |
|---|---|---|---|---|---|---|---|---|
| 16.80 | 23.00 | 45.20 | 148.00 | 9.48 | 0.09 | 0.58 | 1.09 | 6.73 |
| 23.00 | 31.50 | 37.40 | 50.00 | 13.15 | 0.09 | 0.76 | 0.17 | 1.46 |
| 31.50 | 43.10 | 335.50 | 50.00 | 1.71 | 0.93 | 10.81 | 0.02 | 0.26 |
| 43.10 | 58.90 | 0.00 | 0.00 | 0.00 | 0.00 | 0.00 | 0.00 | 0.00 |
| 58.90 | 80.70 | 102.60 | 50.00 | 3.48 | 0.18 | 3.86 | 0.05 | 0.99 |
| 80.70 | 110.40 | 38.70 | 163.60 | 19.05 | 0.14 | 4.11 | 2.67 | 79.28 |
| 110.40 | 150.00 | 33.40 | 231.40 | 28.71 | 0.16 | 6.14 | 8.05 | 319.00 |
| **MRS10–1** 0.00 | 1.00 | 1000.00 | 50.00 | 0.86 | 4.16 | 4.16 | 0.01 | 0.01 |
| 1.00 | 2.00 | 1000.00 | 50.00 | 7.42 | 35.95 | 35.95 | 0.10 | 0.10 |
| 2.00 | 3.00 | 1000.00 | 50.00 | 7.16 | 34.68 | 34.68 | 0.09 | 0.09 |
| 3.00 | 4.00 | 1000.00 | 50.00 | 4.78 | 23.14 | 23.14 | 0.06 | 0.06 |
| 4.00 | 5.00 | 64.80 | 50.00 | 9.16 | 0.19 | 0.19 | 0.12 | 0.12 |
| 5.00 | 6.60 | 54.30 | 50.00 | 34.20 | 0.49 | 0.78 | 0.45 | 0.70 |
| 6.60 | 9.00 | 499.50 | 225.60 | 3.07 | 3.71 | 8.91 | 0.82 | 1.98 |
| 9.00 | 12.30 | 0.00 | 0.00 | 0.00 | 0.00 | 0.00 | 0.00 | 0.00 |
| 12.30 | 16.80 | 74.20 | 489.60 | 7.83 | 0.21 | 0.94 | 9.82 | 44.44 |
| 16.80 | 23.00 | 34.60 | 681.00 | 6.68 | 0.04 | 0.24 | 16.21 | 100.41 |
| 23.00 | 31.50 | 371.10 | 371.60 | 3.64 | 2.43 | 20.62 | 2.63 | 22.27 |
| 31.50 | 43.10 | 1000.00 | 152.10 | 3.42 | 16.54 | 191.91 | 0.41 | 4.80 |
| 43.10 | 58.90 | 0.00 | 0.00 | 0.00 | 0.00 | 0.00 | 0.00 | 0.00 |
| 58.90 | 80.70 | 0.00 | 0.00 | 0.00 | 0.00 | 0.00 | 0.00 | 0.00 |
| 80.70 | 110.40 | 0.00 | 0.00 | 0.00 | 0.00 | 0.00 | 0.00 | 0.00 |
| 110.40 | 150.00 | 0.00 | 0.00 | 0.00 | 0.00 | 0.00 | 0.00 | 0.00 |
| **MRS11–1** 0.00 | 1.00 | 30.00 | 50.00 | 0.05 | 0.00 | 0.00 | 0.00 | 0.00 |
| 1.00 | 2.00 | 1000.00 | 50.00 | 0.72 | 3.51 | 3.51 | 0.01 | 0.01 |
| 2.00 | 3.00 | 0.00 | 0.00 | 0.00 | 0.00 | 0.00 | 0.00 | 0.00 |
| 3.00 | 4.00 | 0.00 | 0.00 | 0.00 | 0.00 | 0.00 | 0.00 | 0.00 |
| 4.00 | 5.00 | 30.00 | 50.00 | 0.73 | 0.00 | 0.00 | 0.01 | 0.01 |
| 5.00 | 6.00 | 30.00 | 50.00 | 2.52 | 0.01 | 0.01 | 0.03 | 0.03 |
| 6.00 | 7.50 | 30.00 | 50.00 | 3.14 | 0.01 | 0.02 | 0.04 | 0.06 |
| 7.50 | 10.00 | 30.00 | 50.00 | 6.60 | 0.03 | 0.07 | 0.09 | 0.22 |
| 10.00 | 13.40 | 63.50 | 50.00 | 9.44 | 0.18 | 0.63 | 0.12 | 0.42 |
| 13.40 | 17.90 | 82.40 | 50.00 | 12.96 | 0.43 | 1.92 | 0.17 | 0.76 |
| 17.90 | 23.90 | 82.60 | 146.00 | 13.87 | 0.46 | 2.75 | 1.55 | 9.28 |
| 23.90 | 31.90 | 83.60 | 279.20 | 18.76 | 0.64 | 5.08 | 7.66 | 61.25 |
| 31.90 | 42.50 | 66.00 | 393.90 | 21.99 | 0.46 | 4.92 | 17.85 | 189.26 |
| 42.50 | 56.70 | 50.30 | 442.70 | 5.17 | 0.06 | 0.90 | 5.31 | 75.38 |
| 56.70 | 75.70 | 0.00 | 0.00 | 0.00 | 0.00 | 0.00 | 0.00 | 0.00 |
| 75.70 | 100.00 | 0.00 | 0.00 | 0.00 | 0.00 | 0.00 | 0.00 | 0.00 |

| MRS11–2 | 0.00 | 1.00 | 0.00 | 0.00 | 0.00 | 0.00 | 0.00 | 0.00 | 0.00 |
|---|---|---|---|---|---|---|---|---|---|
| | 1.00 | 2.00 | 0.00 | 0.00 | 0.00 | 0.00 | 0.00 | 0.00 | 0.00 |
| | 2.00 | 3.00 | 0.00 | 0.00 | 0.00 | 0.00 | 0.00 | 0.00 | 0.00 |
| | 3.00 | 4.00 | 0.00 | 0.00 | 0.00 | 0.00 | 0.00 | 0.00 | 0.00 |
| | 4.00 | 5.00 | 183.90 | 241.80 | 8.56 | 1.40 | 1.40 | 2.62 | 2.62 |
| | 5.00 | 6.00 | 105.40 | 214.00 | 24.84 | 1.34 | 1.34 | 5.95 | 5.95 |
| | 6.00 | 7.50 | 107.70 | 147.50 | 30.44 | 1.71 | 2.57 | 3.47 | 5.20 |
| | 7.50 | 13.40 | 497.60 | 50.00 | 11.07 | 13.28 | 78.35 | 0.14 | 0.85 |
| | 13.40 | 23.90 | 206.00 | 166.50 | 8.81 | 1.81 | 19.01 | 1.28 | 13.42 |
| | 23.90 | 31.90 | 149.30 | 322.90 | 23.44 | 2.53 | 20.25 | 12.79 | 102.35 |
| | 31.90 | 42.50 | 235.70 | 575.40 | 17.91 | 4.82 | 51.09 | 31.04 | 329.00 |
| | 42.50 | 56.70 | 0.00 | 0.00 | 0.00 | 0.00 | 0.00 | 0.00 | 0.00 |
| | 56.70 | 75.70 | 0.00 | 0.00 | 0.00 | 0.00 | 0.00 | 0.00 | 0.00 |
| | 75.70 | 100.00 | 0.00 | 0.00 | 0.00 | 0.00 | 0.00 | 0.00 | 0.00 |

## A8 TEM inversion parameters

**Table A7. TEM inversion parameters.**

| | | |
|---|---|---|
| Ignored time windows | Ignored time before (μs) | 4 |
| | Ignored time after (μs) | 16000 |
| | Use auto protection | yes |
| Adjust cut off ramp | Use cut–off ramp | yes |
| Regularizing algorithm | Low | |
| Variation's limits | Resistivity (ohm–m) | 0.1–4000 |
| | Thickness (m) | 0.25–1000 |
| Smooth field data | Styles | Limited |
| | Tension | Middle |
| Transformation resolution | Middle | |

**Author contributions.** ML, ZS, YZ, MWL, JR, LY, HQ, ZL, JC, LH, TV, JMS, HJHF contributed to the design of this research. ML, JR, LY, HQ, ZL, JC, KH, QZ, PX, FL, KL, YL carried out the fieldwork. All co-authors revised the manuscript and contributed to the writing.

**Competing interests.** All authors declare that there are no conflicts of interests.

**Acknowledgments.** We acknowledge the financial support from Fundamental Research Funds for the Central Universities (300102298307, 300102299305). We thank Chang'an University, ITC, the University of Twente, and Prof. Mark van der Meijde from the University of Twente for their supports, as well as the local government in Maqu for their assistance during



fieldwork. This work was also supported in part by the ESA MOST Dragon IV Program (Monitoring and Modelling Climate
Change in Water, Energy and Carbon Cycles in the Pan-Third Pole Environment).

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
