# Peer review of "A first investigation of hydrogeology and hydrogeophysics of the Maqu catchment in the Yellow River source region."

_Earth System Science Data, 2020_

## Referee Comment (RC1) · Anonymous Referee #1 · 27 Dec 2020

This study investigates the hydrogeology and hydrogeophysics in a catchment of the headwater area of the Yellow River Basin in Tibet Plateau (TP). Multiple analysis and field surveys were conducted including MRS, ERT, and TEM. Hydraulic conductivity is a poorly characterized, yet very important parameter in high mountainous areas, because it is used to quantify the subsurface flow. Studies which is leading to investigate the hydrogeology settings particularly in such a harsh environment are much needed. This study could potentially make a valuable contribution after some revision because so few field surveys have been conducted in TP. However, I do not believe the presentation of the manuscript at this stage is sufficiently good to warrant publication in the Earth System Science Data. Joint use of MRS, TEM, and ERT in TP is not it-

self note-worthy, rather that it is critical to clarify how combined multiple field surveys help to characterize the subsurface structure. Moreover, the manuscript is organized in disorder. The figures and tables fail to reach the standard of high profile scientific journals.

Specific comments:

1. The Introduction should be rewritten to highlight the significance of this study. That is the global map of hydraulic conductivity/permeability lacks realistic data points in TP, such as SoilksatDB (Gupta et al, ESSD, 2020) and permeability database (Gleeson et al., GRL, 2011). This study could fill the scientific and data gaps in a global view.

Gupta, S., Hengl, T., Lehmann, P., Bonetti, S., & Or, D. (2020). SoilKsatDB: global soil saturated hydraulic conductivity measurements for geoscience applications. Earth System Science Data Discussions, 1-26.

Gleeson T, Smith L, Moosdorf N, Hartmann J, Dürr HH, Manning AH, van Beek L P H, Jellinek A M 2011. Mapping permeability over the surface of the Earth. Geophysical Research Letters [J], 38: n/a-n/a.

2. Line 43: Since the hydraulic conductivity is a key parameter for the groundwater system, I would like to suggest using the groundwater model or integrated surface-groundwater model, instead of IHM.

3. Line 72: "Some investigations have been done on the TP based on DEMs." Investigations on what? How these previous works are related to your study? Need to clarify.

4. Line 94: what is the data source for geomorphology and geology? Need references.

5. Figure 1: Since not every reader is familiar with the position of TP, it is necessary to add the position of TP in the China map and its neighboring countries.

6. Lines 113-120: Authors should give an explanation of workflow for Figure 2 rather

than only listing methods. It is redundant to describe the time for each survey because all this information has been listed in Table 1.

7. Figure 6: It should redraw Figure 6 using professional tools which are used for scientific graphs in publication format.

8. Figure 8: There must be something wrong with water table depth (a) and (b). The eastern boundary is the Yellow River, and the elevation is decreasing from west to east, so the value of water table depth is supposed to be big in the western areas and small close to the river. However, the water table depth is 19m near to river while 0 m in the alluvial plain?? Same to Figure 9. Besides, the chromatogram should be changed to better present the gradient of results.

9. Section 4.4.2, Why did authors put equations of aquifer tests in the part of Results and Discussion? This should move to the Method part.

———————————————

---

## Referee Comment (RC2) · Anonymous Referee #2 · 28 Dec 2020

This study investigated the hydrogeology and hydrogeophysics in the Maqu catchment in the headwater area of the Yellow River Basin in the Tibetan Plateau. The investigation data can be useful for building hydrological models in the catchment to simulate the groundwater flow and potentially to understand interaction between surface-subsurface water as an example in the Tibetan Plateau with data-scarcity especially related to sub-surface media processes. while, for final publication of the manuscript, minor revisions are suggested as follows.

Specific comments: -section 4.1: better to show the pictures of the core sediment - section 4.2: please discuss the potential reason for the different accuracy of the seven

[Figure]

datasets. -section 4.3: are there any data, table or figure showing that the soil thicknesses increase from the mountain top to the slope bottom? -Table 5: better to include elevation information also -section 4.4.2: the equations should be described in section 3.

Technical correctionsïijŽ -Line 148: change "the data was" to "the data were" -Figure 9: should "2019 water table depth (m)" be "hydraulic conductivity (m/d)"?
* * *

---

## Editor Comment (EC1) · David Carlson (Editor) · 11 Jun 2021

Despite very good efforts of reviewers, editors and authors, this manuscript does not yet meet ESSD standards for publication. Without restarting the overall process, we ask the authors for additional revisions; details in .pdf attached. Most of our suggested and requested changes have to do with the data (too much that already reside in the repository also exists within the manuscript), with inclusion of the DEM inter-comparison (which we find tangential to the hydrological focus), and with the somewhat confusing mixture of proprietary and open-access software tools. We have notified authors about need for additional changes. Address questions to senior editor

D. Carlson (ipy.djc@gmail.com).

Please also note the supplement to this comment:
https://essd.copernicus.org/preprints/essd-2020-230/essd-2020-230-EC1-supplement.pdf

**Supplement:**

Overall: handled a lot of information from a difficult region, reasonably well-organized and well-written. This reader got the sense of the suite of usual geophysical and hydrologic measurements applied with great effort to this catchment, with better outcomes east vs. west, but overall with mostly tentative initial steps toward a working hydrogeological model. Heroic efforts but mostly to identify what's missing? Provide an overall status / uncertainty of budget components: what works, what works elsewhere but not so well in this particular catchment, what key uncertainties remain, how one might address those, how users should regard this preliminary data product.

The 'separation' / assignment of data seems confusing at best, counter-productive at worst. Most of the data presented in tables (text and appendices) here should in fact reside in the data set itself. A large section, on GPS-RTK 'validation' of various DEMs detracts from their overall focus on hydrogeology and should move to an appendix (for list of DEMs and validation strategy) with actual data in a DEM folder in the repositories. Keep focus on the model and its data needs, put all necessary data in the TP or DANS repositories, put a description of GPS-RTK validation of various DEMs, of interest to some users but not directly related to hydrological parameters, in an appendix with data itself in the repository. Many tables (e.g. Table 6, others) in text report data already included in the repository; no need to duplicate here! No need to include Excel tables here of data already the repository. If you have data in the repository sorted by folder, refer directly to those folders?

Very strong reliance on standard geophysical and hydrological proprietary commercial software not helpful, perhaps even unacceptable. Replace one of the data tables (now included in repository data set) with a list of software: free open-access, proprietary, etc. Show open-access options or substitutes for commercial products where those exist. Provide unfamiliar users with a guide to what they could find easily or develop themselves, what licenses they may already have accompanying which instruments, and what they would need to purchase.

Section 3, Material and methods

Figure 2 - give reader, via text changes (bold, italic, font, etc.), an indication of strengths (low uncertainties) and weaknesses (high or unknown uncertainties) of the various inputs. E.g. from text that follows this reader gets the sense that 'aquifer geometry' remains highly uncertain, almost unknown, due to weaknesses of ERT, MRS, etc. Highlighted uncertainties or places for needed improvements denoted in this figure will set up discussions (now scattered among various results sections) about impact of future instrument or measurement improvements. Come back to this figure in conclusion? How close are authors to having a reasonably well-constrained hydrogeological model and with what reliability should readers regard these measurements? Elevations and lithology strong but conductivities and aquifer geometries weak? Or some different combination of relative strengths and weaknesses that the authors should convey?

Section 4.2, Altitude survey

Necessary, perhaps skillful, but overall a substantial diversion / distraction from the hydrogeological focus. Authors made the case for accurate elevation data, but entire section could be replaced in this text by this (slightly modified) short summary "ALOS RT1, which performed slightly better than other available DEM product across the whole study area and had a higher resolution than ALOS RT2, was the most suitable DEM to use in this study area. For details, see Appendix XX". Please define all acronyms (e.g. satellite names). Text here refers to Table 4 but relevant information also included earlier in Table 2? All of this, including tables and figures, should move to appendix. Convey relatively high reliability factor as a feature of Figure 2? One could retain the locations of GPS-RTK validation points as shown in

Figure 3 at the same time as removing text from main narrative to an appendix and data to a DEM folder at the repository.

Section 4.3, Soil Thickness

Thickness of weathered layers. e.g depth to bedrock from other studies minus surface soil depths from this study will give a difference equal to the second lower weathered layer? But these calculations will happen later, subsequent to data gathered and described here? With what uncertainty? Plus/minus 1m? 10m?

Section 4.4.1, Water table depth measurement

Needs revisions to improve content and references. If "Surfer" and "Ordinary Kriging" represent formal tools, we need to know source, citation, and open availability. This refers to the commercial software 'Surfer'? Not available to most users. Text about linear variogram seems to come straight from GoldenSoftware website? Very standard tool, open access substitutes must exist?

Next paragraph induces confusion. Because people in the west use surface water, need/interest in ground water remains low and few wells drilled? As a consequence, few boreholes exist? These are boreholes numbered 32-34 in Figure 8? Because of rarity, authors decided to exclude these from interpolation. What interpolation? The 'Surfer' interpolation already mentioned? No details given. First mention of interpolation in this document. A good interpolation over a large area needs / uses every point, regardless of isolation? In this catchment, these points deleted for reasons of quality or for reasons of geographic isolation? First and only reference to a dam? Here authors assign lower water tables in 2019 vs 2018 to differences in dam storage, but in concluding sentence of the paragraph the authors mention different "control points" as well as different dam storage conditions. Need revision and clarity here!

Final short paragraph of this Section highly redundant. Remove it, or move it to Abstract?

Section 4.4.2, Aquifer tests

Here the authors accept / use data from isolated rare western stations.  Because they do not apply a software interpolation? Authors provide and justify a range of hydraulic conductivities (e.g. ranged from 0.1. to 15.6 m per day) but data provided includes only geographic coordinates and raw data, not these derived conductivities. Users will need to make their own conversions? Better that authors describe their calculations and provide derived conductivities in addition to raw data, directly in the repository product?

Section 4.5.1 Magnetic susceptibility

Low values of magnetic susceptibility needed only to assure validity of subsequent ERT or MRS measurements. Provide only a brief sentence of assurance here and refer to text / figures in an appendix as well as data in NTPDC for those who want?

Section 4.5.2 ERT

RES2INDV software mentioned here (first mentioned in Section 3.5.2, ERT) represents another proprietary commercial software not available to most readers / users. Perhaps common in geophysical methods but authors need to describe open-access alternatives. Or, we need a list of proprietary software dependencies that covers the entire measurement suite?

"Half of the data missing in the filtering process"? What filtering process? Part of the proprietary RES2INDV processing? Are these data flagged? We get no information on data needed to meet various quality control criteria; did so much data at so many sites fail? How do these failures affect overall conclusions? Affects only ERT1? But periodic rainfall occurred at other stations as well?

Overall, ERT measurements seem useful to or necessary for MRS measurements but not useful or reliable in the absence of other depth-resolved lithology information, for example. Authors say "ERT has equivalence problems, i.e., non–uniqueness of inversion results." Provide instead a short sentence assuring that ERT supports and allows valid MRS measurements but put ERT test in a separate Appendix? You already have ERT data in a repository?

Section 4.5.3 MRS

"at two near MRS sounding sites" - Authors mean at two 'adjacent' sites? Sufficient mention of ERT here, don't need a separate section?  Samovar V6.6 mentioned here represents open access software from IRIS instruments (e.g. described in Section 3.5.3) but a few sentences later in this section reader encounters Samovar V11.4? Different software version? Different instrument type? Because authors clearly assign interpretation differences to V6.6 vs V11.4, readers need to know source of those differences? How much "in situ" water is "missing". 10%? 50%? Not surprising, but how does a reader find this information? "Un-determination": what does this mean? Not resolved? Under-determined? Other instrumentation or lithological factors? We need a much different, much better discussion of sources and levels of uncertainty here; this reader found very little basis to accept any MRS data. Did MRS function effectively or not given these (supposed?, estimated?, measured?) aquifer depths. Not clear that MRS contributes valid information to hydrogeological model, e.g. more/better than borehole estimates. Authors do not provide information necessary to make that determination? By authors own admission, the best they / we can get from MRS remains amount of free water?

Section 4.5.4 TEM

Based again on a commercial proprietary software TEM-Researcher (e.g. mentioned in Section 3.5.4); can the authors explain or list open-access alternatives? In Figure 13 the linear red lines indicate the initial model with the connected red dots represent the interpolated values? At best, these represent preliminary data, e.g as the authors say "several additional measurements will be needed in the future". Need explicit uncertainties here!

Section 6 Conclusion

Authors write "data in this paper can be used for future set up of a hydrogeological conceptual model and groundwater modeling which will be presented in follow up papers." Good effort, no doubt, and thanks for an admirable effort to share, but reader never learns how close they got to a useful reliable groundwater model. What are their priorities for future efforts? Improve instruments / measurements in this catchment? Duplicate work in a second catchment? Focus on modeling rather than observations? How do they recommend that potential users consider or use these data? What do they consider strong or adequate? Where (everywhere?) do they recommend future improvements? Can we as users rely on their soil depths, their borehole pumping data, their "unconfined" aquifer conclusions? Authors give users very little basis for confidence in their efforts and their data.

---

## Author Response (AR1)

The authors thank anonymous reviewers for their useful comments, which will help improve the manuscript. Each comment from referees is repeated in black text here; our responses are given in green text and changes to the manuscript are in blue.

**Response to Reviewer #1**

1. The Introduction should be rewritten to highlight the significance of this study. That is the global map of hydraulic conductivity/permeability lacks realistic data points in TP, such as SoilksatDB (Gupta et al, ESSD, 2020) and permeability database (Gleeson et al., GRL, 2011). This study could fill the scientific and data gaps in a global view.
Gupta, S., Hengl, T., Lehmann, P., Bonetti, S., & Or, D. (2020). SoilKsatDB: global soil saturated hydraulic conductivity measurements for geoscience applications. Earth System Science Data Discussions, 1-26.
Gleeson T, Smith L, Moosdorf N, Hartmann J, Dürr HH, Manning AH, van Beek L P H, Jellinek A M 2011. Mapping permeability over the surface of the Earth. Geophysical Research Letters [J], 38: n/a-n/a.

Many thanks for this comment. A paragraph has been added and the introduction has been modified.

Efforts have been made to develop the global map of permeability (Gleeson et al., 2014; Gleeson et al., 2011), hydraulic conductivity (Gupta et al., 2020; Montzka et al., 2017), groundwater table depth (Fan et al., 2013), groundwater volume and distribution (Gleeson et al., 2016). Nevertheless, due to the remoteness and harsh environment over TP (Yao et al., 2019), the above studies lack reliable in situ data in TP.

`````` The paper is focusing on field hydrogeological, hydrogeophysical surveys, and corresponding datasets, aiming to fill the scientific and data gap in TP from a global view.``````

2. Line 43: Since the hydraulic conductivity is a key parameter for the groundwater system, I would like to suggest using the groundwater model or integrated surface-groundwater model, instead of IHM.

Thanks for this suggestion. IHM has been replaced by integrated surface-groundwater model.

An integrated surface-groundwater model is essential for improving the understanding of different processes quantitatively (Graham and Butts, 2005). To set up an integrated surface-groundwater model, different kinds of data are needed for parameterization of land surface and subsurface, for atmospheric forcing, and state variables are required for model calibration and validation.

3. Line 72: "Some investigations have been done on the TP based on DEMs." Investigations on what? How these previous works are related to your study? Need to clarify.

Thanks for this comment. The paragraph has been modified:

Investigations on various fields, such as geomorphology, climate change, glacier, and permafrost have been carried out on the TP based on different DEMs. Zhang et al. (2006) analyzed the geomorphic characteristics of the Minjiang drainage basin with SRTM (Shuttle Radar Topography Mission) data. Wei and Fang (2013) assessed the trends of climate change and temporal-spatial differences over the TP from 1961–2010, with a generalized temperature zone–elevation model and SRTM. Ye et al. (2015) calculated the glacier elevation change in the Rongbuk catchment from 1974 to 2006 based on topographic maps and ALOS. Niu et al. (2018) mapped permafrost distribution throughout the Qinghai–Tibet Engineering Corridor based on ASTER Global DEM. However, different DEMs used in different studies may lead to potential inconsistencies for understanding relevant physical processes. For Maqu catchment, it is crucial to understand the accuracy of different DEMs, since it controls the flow field of groundwater in this mountainous region. Therefore, we evaluate the accuracy of DEMs with a Real-time Kinematic-Global Positioning System (GPS-RTK), which has not been given attentions in many studies over the TP.

4. Line 94: what is the data source for geomorphology and geology? Need references.

Thanks for pointing out this issue. The sentence has been revised:

Based on the field survey of geomorphology and geology, the catchment can be divided into two parts.

5. Figure 1: Since not every reader is familiar with the position of TP, it is necessary to add the position of TP in the China map and its neighboring countries.

Thanks for this suggestion, Figure 1 has been modified accordingly.

[Figure]

Figure 1. The geographical location of Maqu catchment in the TP and geomorphologic map. (a) The geographical location and boundary of the TP (Zhang et al., 2014a; Zhang et al., 2014b), and the geographical location of Maqu catchment; (b) The geomorphologic map of Maqu catchment.

6. Lines 113-120: Authors should give an explanation of workflow for Figure 2 rather than only listing methods. It is redundant to describe the time for each survey because all this information has been listed in Table 1.

Thanks for this comment, this paragraph has been revised:

Figure 2 shows the fieldwork workflow towards establishing a hydrogeological conceptual model, which includes the borehole core lithology analysis, altitude survey, soil thickness measurement, hydrogeological survey, and hydrogeophysical survey (Table 1 and Fig. 2). Yellow boxes in Fig. 2 represent the fieldwork, green boxes represent the results of fieldwork, which finally contribute to the hydrogeological conceptual model shown in a blue box. The obtained information on lithology, soil thickness, and elevation provides basic knowledge in the study area. Hydrogeological measurements of water table depth and hydraulic conductivity provide important input that can be used to deduce the direction and rate of regional groundwater flow. For hydrogeophysical results, magnetic susceptibility ensures the suitability of applying MRS, which provides information on water content and transmissivity. Furthermore, ERT not only provides information on underground resistivity but also integrated with MRS for retrieving water content and transmissivity. The locations of the surveys and measurements are shown in Fig. 3 and Fig. 4.

7. Figure 6: It should redraw Figure 6 using professional tools which are used for scientific graphs in publication format.

Thanks a lot. Figure 6 has been redrawn:

[Figure]

[Figure]

**Figure 6. GPS-RTK elevations vs. DEM elevations.**

8. Figure 8: There must be something wrong with water table depth (a) and (b). The eastern boundary is the Yellow River, and the elevation is decreasing from west to east, so the value of water table depth is supposed to be big in the western areas and small close to the river. However, the water table depth is 19m near to river while 0 m in the alluvial plain?? Same to Figure 9. Besides, the chromatogram should be changed to better present the gradient of results.

Thanks for the query and suggestion. The eastern boundary is the Yellow River, and the elevation is decreasing from west to east. This is likely to result in big hydraulic heads in the west and small hydraulic heads in the east, so that groundwater flows from the west to the east according to Darcy's law. As for the water table depth, which is the distance from the ground surface to the groundwater table, is not necessarily big in the west and small in the east. The chromatogram has been adjusted to better present the gradient of the groundwater table in Figure 8.
The chromatogram of the 2019 water table depth (m) in Figure 9 has been removed to avoid confusion.

[Figure]

**Figure 8. Water table depths (m) and piezometric heads (m a.s.l) of east Maqu catchment. (a) and (b) are water table depths (m) of east Maqu catchment in 2018 and 2019, respectively; (c) and (d) are piezometric heads (m a.s.l) of eastern Maqu catchment in 2018 and 2019, respectively; (e) is the difference (m) of water table depth between 2018 and 2019. Numbers from 1 to 34 are identification numbers of boreholes listed in Table 6.**

[Figure]

**Figure 9. Hydraulic conductivity (m.d⁻¹) obtained from aquifer tests, east of Maqu catchment.**

9. Section 4.4.2, Why did authors put equations of aquifer tests in the part of Results and Discussion? This should move to the Method part.

Thanks for this comment. The equations have been moved to section 3.4.2 Aquifer tests part.

**3.4.2 Aquifer tests**

Aquifer tests, including pumping tests and slug tests, were conducted to obtain aquifer hydraulic conductivity (Fig. 3a). The first pumping test was done in 2017, in the borehole ITC_Maqu_1, where core lithology information is available. The pumping rate was constant 55.6 $m^3.d^{-1}$ measured with a flowmeter, and the pumping duration was about 30 minutes. The pumping rate was limited because the borehole ITC_Maqu_1 could easily collapse if the pumping rate were too high. The water level became stable soon after the start of pumping and was recorded every minute using a data logger (TD–Diver manufactured by Van Essen Instruments, with a range of 10 m). Other tests were carried out in 2019, including two pumping tests and eight slug tests (Fig. 3a). For the two pumping tests with the pumping rate of 31.6 $m^3.d^{-1}$ and 101.52 $m^3.d^{-1}$, due to practical reasons, only water level recovery data were analyzed. In the eight slug tests, the groundwater level was abruptly lowered by extracting 11.75 L water from the borehole. The water levels were recorded every second or two seconds in slug tests and every five seconds or 20 seconds in pumping tests using a data logger (3001 Levelogger Edge manufactured by Solinst, with a range of 10 m).

The pumping test data acquired from the borehole ITC_Maqu_1 were analyzed using the Boulton (1963) method as follows:

$$S = \frac{Q}{4\pi T} W(U_{AB}, r/D), \tag{1}$$

where $S_D$ is drawdown (m), $Q$ is pumping rate (m³.d⁻¹), $T$ is transmissivity (m².d⁻¹), $W(U_{AB}, r/D)$ is Boulton's well-function (dimensionless).

Slug tests data were analyzed using the Bouwer and Rice (1976) method for hydraulic conductivity as follows:

$$K = \frac{r^2 \ln\left(\frac{R_e}{R}\right)}{2L} \cdot \frac{1}{t} \cdot \ln\left(\frac{h_0}{h_t}\right),$$

(2)

where $K$ is hydraulic conductivity (m.d⁻¹), $r$ is the radius of the borehole casing (m), $R_e$ is the effective radial distance over which the head difference is dissipated (m), $R$ is radius measured from borehole center to undisturbed aquifer (m), $L$ is the length of the screen (m), $t$ is time (d), $h_0$ is the water level at time 0 (m), and $h_t$ is the water level at time t (m).

Another two pumping test data were analyzed using the Boulton and Agarwal method. Agarwal (1980) defines the recovery drawdown $S_r$ (m) as the difference between the head $h_p$ (m) at the end of the pumping period and the head $h$ (m) during the recovery period.

$$S_r = h - h_p,$$

(3)

The recovery time $t_r$ (d) is the time since the recovery started calculated as the difference between the duration of pumping $t_p$ (d) and the time $t$ (d) since pumping started.

$$t_r = t - t_p,$$

(4)

**Response to Reviewer #2**

-section 4.1: better to show the pictures of the core sediment

Thanks for this suggestion. The core lithology and a picture of the core sediment have been added to Figure 5.

[Figure]

Figure 5. Borehole information: (a) The core lithology of borehole ITC_Maqu_1; (b) A picture of the core sediment when the borehole was drilled; (c) Location of boreholes RM and RH (after Chen et al. (1999) ).

-section 4.2: please discuss the potential reason for the different accuracy of the seven datasets.

Thanks a lot for this comment. The potential reasons for the different accuracy of the seven datasets have been discussed and added in section 4.2:

The DEMs' quality can be influenced by several factors, such as sensor type, algorithm, terrain type, and grid spacing. (Hebeler and Purves, 2009). In this study, grid spacings of DEMs are similar except for ALOS RT1, so the main factors that affect the accuracy of the DEMs should be sensor types and algorithms. For SRTM, the issue inherent to the production method is mast oscillations, while for ASTER and AW3D30, the issue is scene mismatch (Grohmann, 2018). As for radiometrically terrain corrected (RTC) products ALOS RT1 and ALOS RT2, the quality is directly related to the quality of the source DEM SRTM which was used in the RTC process. This results in very similar correlation coefficients of SRTM, ALOS RT1, and ALOS RT2, and obvious improvements in RMSE, MAE, and ME (Table 4).

-section 4.3: are there any data, table or figure showing that the soil thicknesses increase from the mountain top to the slope bottom?

Many thanks for pointing out this mistake. The sentence was deleted because the soil thickness is more related to slopes, rather than mountain top or bottom.

**4.3 Soil thickness measurement**

Results of soil thickness measurements are listed in Table 5 (location shown in Fig.3). The soil thickness decreases as the slope increases, and are within 1.2 m in most cases (Fig. 7).

-Table 5: better to include elevation information also

Thanks for this suggestion, the elevation information has been added to Table 5.

Table 5. Soil thickness measurements, locations of each measurement can be found in Figure 3.

| No. | Depth (cm) | Slope (°) | Elevation* (m) | No. | Depth (cm) | Slope (°) | Elevation* (m) | No. | Depth (cm) | Slope (°) | Elevation* (m) |
|---|---|---|---|---|---|---|---|---|---|---|---|
| 1 | 39 | 9 | 3762 | 27 | 71 | 10 | 3509 | 53 | 102 | 6 | 3457 |
| 2 | 45 | 20 | 3769 | 28 | 90 | 11 | 3503 | 54 | 102 | 14 | 3459 |
| 3 | 28 | 25 | 3777 | 29 | >120 | 5 | 3493 | 55 | 104 | 6 | 3460 |
| 4 | 48 | 16 | 3784 | 30 | 110 | 5 | 3488 | 56 | 100 | 13 | 3462 |
| 5 | 50 | 22 | 3783 | 31 | >120 | 5 | 3482 | 57 | 92 | 10 | 3469 |
| 6 | 46 | 14 | 3775 | 32 | >107 | 2 | 3473 | 60 | 40 | 9 | 3491 |
| 7 | 39 | 25 | 3770 | 33 | >110 | 4 | 3479 | 61 | 53 | 6 | 3480 |
| 8 | 34 | 41 | 3757 | 34 | 59 | 13 | 3488 | 62 | 61 | 15 | 3478 |
| 9 | 37 | 22 | 3750 | 35 | 85 | 13 | 3491 | 63 | 70 | 7 | 3476 |
| 10 | 42 | 19.5 | 3734 | 36 | 60 | 20 | 3502 | 64 | 63 | 14 | 3468 |
| 11 | 23 | 20 | 3732 | 37 | 92 | 13 | 3517 | 65 | 61 | 9 | 3467 |
| 12 | 52 | 0 | 3461 | 38 | 38 | 10 | 3452 | 66 | 87 | 10 | 3458 |
| 13 | 42 | 3 | 3462 | 39 | 41 | 20 | 3461 | 67 | 60 | 5 | 3496 |
| 14 | 35 | 3 | 3463 | 40 | 76 | 30 | 3472 | 68 | 63 | 7 | 3487 |
| 15 | 38 | 4 | 3470 | 41 | 55 | 30 | 3483 | 69 | 68 | 15 | 3474 |
| 16 | 50 | 9 | 3474 | 42 | 32 | 40 | 3501 | 70 | 87 | 18 | 3554 |
| 17 | 40 | 10 | 3482 | 43 | 80 | 35 | 3519 | 71 | 30 | 14 | 3562 |
| 18 | 38 | 10 | 3489 | 44 | 27 | 30 | 3530 | 72 | 85 | 20 | 3572 |
| 19 | 42 | 15 | 3502 | 45 | 49 | 30 | 3522 | 73 | 41 | 17 | 3587 |
| 20 | 37 | 8 | 3494 | 46 | 52 | 30 | 3514 | 74 | 83 | 13 | 3596 |
| 21 | 40 | 10 | 3488 | 47 | 43 | 20 | 3500 | 75 | 67 | 27 | 3612 |
| 22 | 30 | 5 | 3475 | 48 | 44 | 22 | 3484 | 76 | 63 | 20 | 3605 |
| 23 | 30 | 4 | 3472 | 49 | 30 | 25 | 3475 | 77 | >110 | 20 | 3593 |
| 24 | 35 | 4 | 3469 | 50 | 74 | 14 | 3470 | 78 | >110 | 10 | 3574 |
| 25 | 28 | 1 | 3463 | 51 | 37 | 12 | 3464 | 79 | 42 | 15 | 3564 |
| 26 | 29 | 0 | 3459 | 52 | 81 | 6 | 3447 | | | | |

*Elevations were extracted from ALOS PALSAR RT1.

-section 4.4.2: the equations should be described in section 3.

Thanks for this comment. The equations have been moved to section 3.4.2 Aquifer tests part.

**3.4.2 Aquifer tests**

Aquifer tests, including pumping tests and slug tests, were conducted to obtain aquifer hydraulic conductivity (Fig. 3a). The first pumping test was done in 2017, in the borehole ITC_Maqu_1, where core lithology information is available. The pumping rate was constant 55.6 $m^3.d^{-1}$ measured with a flowmeter, and the pumping duration was about 30 minutes. The pumping rate was limited because the borehole ITC_Maqu_1 could easily collapse if the pumping rate were too high. The water level became stable soon after the start of pumping and was recorded every minute using a data logger (TD–Diver manufactured by Van Essen Instruments, with a range of 10 m). Other tests were carried out in 2019, including two pumping tests and eight slug tests (Fig. 3a). For the two pumping tests with the pumping rate of 31.6 $m^3.d^{-1}$ and 101.52 $m^3.d^{-1}$, due to practical reasons, only water level recovery data were analyzed. In the eight slug tests, the groundwater level was abruptly lowered by extracting 11.75 L water from the borehole. The water levels were recorded every second or two seconds in slug tests and every

five seconds or 20 seconds in pumping tests using a data logger (3001 Levelogger Edge manufactured by Solinst, with a range of 10 m).

The pumping test data acquired from the borehole ITC_Maqu_1 were analyzed using the Boulton (1963) method as follows:

$$S = \frac{Q}{4\pi T} W(U_{AB}, r/D),$$   (1)

where $S_D$ is drawdown (m), $Q$ is pumping rate (m³.d⁻¹), $T$ is transmissivity (m².d⁻¹), $W(U_{AB}, r/D)$ is Boulton's well-function (dimensionless).

Slug tests data were analyzed using the Bouwer and Rice (1976) method for hydraulic conductivity as follows:

$$K = \frac{r^2 \ln\left(\frac{R_e}{R}\right)}{2L} \cdot \frac{1}{t} \cdot \ln\left(\frac{h_0}{h_t}\right),$$   (2)

where $K$ is hydraulic conductivity (m.d⁻¹), $r$ is the radius of the borehole casing (m), $R_e$ is the effective radial distance over which the head difference is dissipated (m), $R$ is radius measured from borehole center to undisturbed aquifer (m), $L$ is the length of the screen (m), $t$ is time (d), $h_0$ is the water level at time 0 (m), and $h_t$ is the water level at time t (m).

Another two pumping test data were analyzed using the Boulton and Agarwal method. Agarwal (1980) defines the recovery drawdown $S_r$ (m) as the difference between the head $h_p$ (m) at the end of the pumping period and the head $h$ (m) during the recovery period.

$$S_r = h - h_p,$$   (3)

The recovery time $t_r$ (d) is the time since the recovery started calculated as the difference between the duration of pumping $t_p$ (d) and the time $t$ (d) since pumping started.

$$t_r = t - t_p,$$   (4)

Technical corrections-Line 148: change "the data was" to "the data were"

Thanks a lot. Corrected.

The data were intended to be used to evaluate seven DEM datasets (Table 2).

-Figure 9: should "2019 water table depth (m)" be "hydraulic conductivity (m/d)"?

It is 2019 water table depth (m). I put it in Figure 9 because those hydraulic conductivity values were obtained in 2019. Now it is removed to avoid confusion.

[Figure]

**Figure 9. Hydraulic conductivity (m.d⁻¹) obtained from aquifer tests, east of Maqu catchment.**

---

## Author Response (AR2)

We would like to thank you for the positive and constructive feedback. Below we addressed each specific issue and the manuscript has been updated accordingly.

(1) Line 132 of your revised manuscript, please provide some references for the sentence "Based on the filed survey of geomorphology and geology, the catchment can be divided into two parts".

Thanks for this comment. A references was added.

Based on the field survey of geomorphology and geology (Ministry of Land and Resources of the People's Republic of China, 2015), the catchment can be divided into two parts, the flat eastern area, and the western mountainous area. The western

(2) Since your article focuses on the Maqu catchment, Tibetan Plateau, I strongly recommend that you submit a copy of the data to the National Tibetan Plateau/Third Pole Environment Data Center (https://data.tpdc.ac.cn/en/), which specializes in collecting, integrating, and publishing geoscientific data on and surrounding the Tibetan Plateau. This will help expand the impact of your valuable scientific dataset and the results of this paper.

Thanks a lot for this suggestion. The data is now available at the National Tibetan Plateau/Third Pole Environment Data Center.

**Abstract.** The Tibetan Plateau is the source of most of Asia's major rivers and has been called the Asian Water Tower. Detailed knowledge of its hydrogeology is paramount to enable the understanding of groundwater dynamics, which plays a vital role in headwater areas like the Tibetan Plateau. Nevertheless, due to its remoteness and the harsh environment, there is a lack of field survey data to investigate its hydrogeology. In this study, borehole core lithology analysis, altitude survey, soil thickness measurement, hydrogeological survey, and hydrogeophysical surveys (e.g., Magnetic Resonance Sounding – MRS, Electrical Resistivity Tomography – ERT, and Transient Electromagnetic – TEM) were conducted in the Maqu catchment within the Yellow River Source Region (YRSR). The soil thickness measurements were done in the western mountainous area of the catchment, where hydrogeophysical surveys were difficult to be carried out. The results indicate soil thicknesses are within 1.2 m in most cases, and the soil thickness decreases as the slope increases. The hydrogeological survey reveals that groundwater flows from the west to the east, recharging the Yellow River. The hydraulic conductivity ranges from 0.2 $m.d^{-1}$ to 12.4 $m.d^{-1}$. The MRS soundings results, i.e., water content and hydraulic conductivity, confirmed the presence of an unconfined aquifer in the flat eastern area. The depth of the Yellow River deposits was derived at several places in the flat eastern area based on TEM results. These survey data and results can contribute to integrated hydrological modeling and water cycle analysis to improve a full–picture understanding of the water cycle at the Maqu catchment in the YRSR. The raw data set is freely available at https://doi.org/10.17026/dans-z6t-zpn7 (Li et al., 2020a), and the data set containing the processed ERT, MRS, and TEM data is available at the National Tibetan Plateau Data Center with the link https://doi.org/10.11888/Hydro.tpdc.271221 (Li et al., 2020b).

**5 Data availability**

The raw dataset is archived and freely available in the DANS repository under the link https://doi.org/10.17026/dans-z6t-zpn7 (Li et al., 2020a), and the data set containing the processed ERT, MRS, and TEM data is available at the National Tibetan Plateau Data Center with the link https://doi.org/10.11888/Hydro.tpdc.271221 (Li et al., 2020b).

---

## Author Response (AR3)

We would like to thank you for the very constructive feedback. Below we addressed each specific comment and the manuscript has been updated accordingly. Different font colors represent different things:

Black – comments

Blue – reply

Red – modifications in the paper

1. **Overall**: handled a lot of information from a difficult region, reasonably well-organized and well-written. This reader got the sense of the suite of usual geophysical and hydrologic measurements applied with great effort to this catchment, with better outcomes east vs. west, but overall with mostly tentative initial steps toward a working hydrogeological model. Heroic efforts but mostly to identify what's missing?

Thank you for the comments! What's missing is detailed geometrical mapping of the site's subsurface. The aim of our paper is to present the collected unique data of an hydrogeophysical inveatitagion. To our knowledge, this is the first detailed hydrogeophysical investigation of a cathment on the Tibetan plateau.

To further reduce uncertainties from indirect techniques, ERT, MRS, and TEM, it is important to determine subsurface geometry and its fabric.

Provide an overall status / uncertainty of budget components

Thank you for the comments! The overall uncertainty has been added in the paper in 4.5 Uncertainties section

4.5 Uncertainties

As shown in Fig. 2, direct techniques, i.e. particle size analysis, altitude survey, soil thickness measurement, water table depth measurement, aquifer test, and magnetic susceptibility measurements have low uncertainties. There are random errors for particle size analysis (Wang, 2011), but they are small and not expected to affect the final lithology result (ASTM, 2017), and thus can be neglected. For measured ground surface elevations, soil thicknesses, and water table depths, the uncertainty is supposed to be within a few centimeters considering the accuracy of equipment and errors during the measurement process (Burt, 2014; Cunningham and Schalk, 2011; Rydlund Jr and Densmore, 2012). In terms of hydraulic heads derived from ALOS RT1 in boreholes, the uncertainty not only comes from water table depths measurement, but also from ALOS RT1 which contains the mean absolute error of 4.4 m in the study area based on our results (Table A1). For hydraulic conductivities obtained from aquifer tests, the uncertainty mainly comes from data collection and processing. Though the duration of pumping in the borehole ITC_Maqu_1 did not reach 48 hours, the water levels became steady very soon after the pumping started, so the uncertainty is estimated to be within 25% according to studies from Brown et al. (1995) and Delnaz et al. (2019). For magnetic susceptibility, although the resolution of SM–20 is 1e–6 SI units, the actual reading accuracy is dependant on appropriate corrections, e.g. temperature, shape, volume, effective distance to sensor, etc. The corrections may reach a few orders of magnitudes for volume and up to ± ~50% for shape (Hoffman, 2006). In the case of Maqu catchment, these are so far from the levels at which MRS problem may occur that, corrected or not, the final results will still be below the threshold for concerns.

In terms of indirect techniques, ERT, MRS, and TEM, performances of the raw data could be evaluated with parameters such as S/N for MRS and amount of bad data for ERT and TEM. Knowledge of the subsurface geometry and fabric would lead to the resolution of the main uncertainty issues for inverted data. Because there are implicit modeling assumptions for each method. For example, the assumption for MRS is that the subsurface is made of 1D planar layers parallel to the MRS loop with depth-increasing thickness. We cannot quantify to what extent these assumptions are met, and therefore also to what extent the inversion data are accurate measurements of the site's hydrogeological parameters, thus appropriate uncertainty figures cannot be reliably generated for inverted data. The inverted data, as an illustration of what can be extracted from the raw data, are preliminary results with only inversion RMS errors quantified (ERT and TEM). Lake deposits, being far from the source, should not suffer from the near-source river deposits heterogeneity, but its lithology makes it insensitive to MRS.

To further reduce uncertainties from indirect techniques, ERT, MRS, and TEM, it is important to determine subsurface geometry and its fabric. State-of-the-art airborne electromagnetic technology allows high spatial resolution mapping down to 500 m depth and is probably the most appropriate tool for now (Legault, 2015). After the site geometry is properly mapped and the subsurface fabric is properly understood, optimum borehole drilling locations can be selected. When the detailed geometrical mapping of the subsurface and systematic borehole information are available, the inversion process can be better constrained and improved (Galazoulas et al., 2015; Vouillamoz et al., 2005; Wang et al., 2021).

what works,

Thank you for the comments!
Generally speaking, all the methods work well in the study area, and have confirmed the presence of an unconfined fluvial aquifer within the 250 m below surface and the presence of lake deposits with much finer pores lithology what works elsewhere but not so well in this particular catchment,

Thank you for the comments!
In Maqu catchment, a near-source river environment, without adequate geometrical mapping, the representativity of the various sampled volumes is unknown as well as whether the sampled volume fits the models used for data inversion. This is much less the case further away from the source where homogeneity and fitting of the model to the actual hydrogeological setting is achieved. In such away-from-source case, pumping tests data may be assumed to be representative of the tested formation while in techniques such as MRS, depth and thickness information may be extracted from the datasets as well as hydraulic estimates. This is an ongoing project and it may become available later if such above-mentioned mapping is completed. Any further similar surveys and borehole drillings would benefit from such geometrical mapping since their precise localization may then be optimized in view of proper data inversion and information gaps filling.

what key uncertainties remain, how one might address those,

Thank you for the comments! This has been explained in the 4.5 Uncertainties section
which is shown before.

how users should regard this preliminary data product.

Thank you for the comments!
The data from direct techniques with low uncertainties, as shown in Fig. 2: lithology, ground
surface elevation, soil thickness, water table depth measurement, hydraulic conductivity from
aquifer test, magnetic susceptibility, can contribute to related global or regional databases
where the in-situ data over TP is scarce, or be regarded as verification and validation data for
groundwater modeling over Maqu catchment. The data from indirect techniques, ERT, MRS,
and TEM, is a rare unique and particularly rich training data source for geoscientists interested
in the data processing and interpretation of the particular hydrogeological and
hydrogeophysical techniques used here. It is a dynamic set where additional complementary
data will gradually add constraints to the inversion processes. For example, a researcher
developing new techniques for S/N improvements of some of these techniques will get free
and highly relevant data to work with.

2.1 The 'separation' / assignment of data seems confusing at best, counter-productive at
worst. Most of the data presented in tables (text and appendices) here should in fact
reside in the data set itself.

Thank you for the comments! Table 3 showing borehole core lithology, Table 5 showing
measured soil thicknesses, Table 6 showing water table measurements, and Fig. A1
showing aquifer tests data and derived hydraulic conductivities, have been deleted

2.2 A large section, on GPS-RTK 'validation' of various DEMs detracts from their overall
focus on hydrogeology and should move to an appendix (for list of DEMs and validation
strategy) with actual data in a DEM folder in the repositories. Keep focus on the model
and its data needs, put all necessary data in the TP or DANS repositories, put a description
of GPS-RTK validation of various DEMs, of interest to some users but not directly related
to hydrological parameters, in an appendix with data itself in the repository.

Thank you for the very helpful comments! The content of section 4.2 Altitude survey has
been replaced by a summary sentence, and merged with section Water table depth
measurement. The original content has been moved to the appendix as you suggested
later.

4.3.1 Water table depth measurement
For the altitude survey, ALOS RT1, with a spatial resolution of 12.5 m, performed better than
other DEM products across the whole study area and had a higher resolution than the others.

It was the most suitable DEM to be used in this study area for determining water table (WT)
depths. For details, see Appendix A1.
There were 22 WT depths measured in 2018, and 18 in 2019 (Fig. 3)……
2.3 Many tables (e.g. Table 6, others) in text report data already included in the repository;
no need to duplicate here! No need to include Excel tables here of data already in the
repository.
Thank you for the comments! The data of Fig. 6 (showing the measured altitudes v.s.
altitudes from 7 DEMs) have been put in a DEM.xlsx, which has been uploaded to the
National Tibetan Plateau Data Center.
2.4 If you have data in the repository sorted by folder, refer directly to those folders?
Thank you for the comments! Since the folder is available in the DANS but not available
at the National Tibetan Plateau Data Center. I refer directly to the file DEM.xlsx.
Appendix A
A1 Altitude survey
46 ground surface elevations were measured (33 in the flat east, 13 in the mountainous
west), and were used to evaluate the accuracies of seven DEM datasets (data available in
DEM.xlsx in the National Tibetan Plateau Data Center) and the most accurate one was
applied in this study ……
2.5 Very strong reliance on standard geophysical and hydrological proprietary
commercial software not helpful, perhaps even unacceptable. Replace one of the data
tables (now included in repository data set) with a list of software: free open‑access,
proprietary, etc. Show open‑access options or substitutes for commercial products where
those exist. Provide unfamiliar users with a guide to what they could find easily or develop
themselves, what licenses they may already have accompanying which instruments, and
what they would need to purchase.
Thanks for the helpful comments! A sheet named "Softwares" has been added in the
National Tibetan Plateau Data Center.

[Figure]

Considering the answer to the question before: how users should regard this preliminary
data product, at this step, it is assumed that such interested geoscientists are already 'up‑
and‑running' with respect to the appropriate data processing and interpretation tools. It
may be noted that free and in some cases, open‑source software exists for several of the techniques used here but their use would often be a full-time job on account of the
needed adaptation and improper documentation.
Section 3, Material and methods
Figure 2 - give reader, via text changes (bold, italic, font, etc.), an indication of strengths
(low uncertainties) and weaknesses (high or unknown uncertainties) of the various inputs.
E.g. from text that follows this reader gets the sense that 'aquifer geometry' remains
highly uncertain, almost unknown, due to weaknesses of ERT, MRS, etc.
Thanks for your comments. Figure 2 has been modified according to your comments.

[Figure]

**Figure 2. Fieldwork workflow for setting up a hydrogeological conceptual model at Maqu catchment, where**
**italics represent indirect technique (e.g., inversion type of retrieval) with unknown uncertainty, regular bold**
**letters represent direct technique with low uncertainty, and regular letters do not convey uncertainty**
**information.**

Highlighted uncertainties or places for needed improvements denoted in this figure will
set up discussions (now scattered among various results sections) about impact of future
instrument or measurement improvements.
Thank you for the comments! A new section Uncertainties has been added (shown at the
beginning of this document in red). The uncertainties were discussed and the way to
improve the data reliability was pointed out.
Come back to this figure in conclusion? How close are authors to having a reasonably
well-constrained hydrogeological model and with what reliability should readers regard
these measurements? Elevations and lithology strong but conductivities and aquifer
geometries weak? Or some different combination of relative strengths and weaknesses
that the authors should convey?
Thanks for your comments. The conclusion has been modified. This is part of the
conclusion:

By combining our dataset with available depth to bedrock dataset, a preliminary
hydrogeological conceptual model can be established. If combining our dataset with detailed
geometrical mapping of the subsurface and deep borehole information, a more complete
and accurate conceptual model can be obtained.
The reliability or uncertainty of each component has been discussed in the new section
Uncertainties shown before.
Section 4.2, Altitude survey
Necessary, perhaps skillful, but overall a substantial diversion / distraction from the
hydrogeological focus. Authors made the case for accurate elevation data, but entire
section could be replaced in this text by this (slightly modified) short summary "ALOS RT1,
which performed slightly better than other available DEM product across the whole study
area and had a higher resolution than ALOS RT2, was the most suitable DEM to use in
this study area. For details, see Appendix XX".
Thank you for the comments!
The entire section has been replaced by the summary sentence and merged with the
section Water table depth measurement.
4.3.1 Water table depth measurement
For the altitude survey, ALOS RT1, with a spatial resolution of 12.5 m, performed better than
other DEM products across the whole study area and had a higher resolution than the others.
It was the most suitable DEM to be used in this study area for determining water table (WT)
depths. For details, see Appendix A1.
There were 22 WT depths measured in 2018, and 18 in 2019 (Fig. 3)……
Please define all acronyms (e.g. satellite names).
Thank you for the comments!
The satellite names have been defined in Table 2. Acronyms have been defined in the
table in the appendix:
**Table 2. Seven DEM datasets.**

| Number | Name | DEM | Resolution | Source |
| --- | --- | --- | --- | --- |
| 1 | SRTM | Shuttle Radar Topography Mission | 1 Arc–Second | USGS |
| 2 | ASTER V1 | The Terra Advanced Spaceborne Thermal Emission and Reflection Radiometer (ASTER) Global Digital Elevation Model (GDEM) Version 1 | 1 Arc–Second | USGS |
| 3 | ASTER V2 | ASTER GDEM Version 2 | 1 Arc–Second | USGS |
| 4 | ASTER V3 | ASTER GDEM Version 3 | 1 Arc–Second | USGS |
| 5 | AW3D30 | Advanced Land Observing Satellite (ALOS) World 3D – 30 m Version 2.2 | 30 m | JAXA |
| 6 | ALOS RT2 | ALOS Phase Array type L band Synthetic Aperture Radar (PALSAR) low terrain correction resolution (RT2) | 30 m | ASF |

| 7 | ALOS RT1 | ALOS PALSAR high terrain correction resolution (RT1) | 12.5 m | ASF |
|---|---|---|---|---|

A7 Acronyms

**Table A10. Acronyms.**

| | |
|---|---|
| **ALOS PALSAR RT1** | Advanced Land Observing Satellite - Phase Array type L band Synthetic Aperture Radar - high terrain correction resolution |
| **ALOS PALSAR RT2** | Advanced Land Observing Satellite - Phase Array type L band Synthetic Aperture Radar - low terrain correction resolution |
| **AMSR-E** | Advanced Microwave Scanning Radiometer for Earth Observing System |
| **ASCAT** | Advanced Scatterometer |
| **ASF** | Alaska Satellite Facility |
| **ASTER** | The Terra Advanced Spaceborne Thermal Emission and Reflection Radiometer |
| **CAS** | Chinese Academy of Science |
| **CLM** | Community Land Model |
| **CPC** | Climate Prediction Center |
| **DEM** | Digital Elevation Model |
| **ERT** | Electrical Resistivity Tomography |
| **GDEM** | Global Digital Elevation Model |
| **GLDAS** | Global Land Data Assimilation System |
| **GPS** | Global Positioning System |
| **GPS-RTK** | Real-time Kinematic-Global Positioning System |
| **GRACE** | Gravity Recovery and Climate Experiment |
| **JAXA** | Japan Aerospace Exploration Agency |
| **LAPSUS** | Landscape process modeling at multi-dimensions and scales |
| **ME** | Mean Error |
| **MAE** | Mean Absolute Error |
| **MRI** | Magnetic Resonance Imaging |
| **MRS** | Magnetic Resonance Sounding |
| **NMR** | Nuclear Magnetic Resonance |
| **RMSE** | root mean squared error |
| **SRTM** | Shuttle Radar Topography Mission |
| **TDEM** | Time-Domain Electromagnetic |
| **TEM** | Transient Electromagnetic |
| **TP** | the Tibetan Plateau |
| **USGS** | United States Geological Survey |
| **VIC** | Variable Infiltration Capacity model |
| **WT** | Water Table |
| **YRSR** | Yellow River Source Region |

Text here refers to Table 4 but relevant information also included earlier in Table 2?

Thank you for the comments! The second column resolution in Table 4 is also included
in Table2, The column has been deleted in Table 4:
**Table A1. Statistical analysis of seven DEMs in the study area.**

| DEM | Min Error * (m) | Max Error (m) | Max Error −Min Error (m) | MAE (Mean Absolute Error) (m) | ME (Mean Error) (m) | Correlation coefficient | RMSE (m) |
|---|---|---|---|---|---|---|---|
| SRTM | 22 | 44 | 22 | 35.488 | 35.488 | 0.985 | 35.936 |
| ASTER V1 | −17 | 43 | 60 | 24.761 | 24.010 | 0.950 | 26.565 |
| ASTER V2 | −8 | 55 | 63 | 27.483 | 27.140 | 0.941 | 30.171 |
| ASTER V3 | 4 | 45 | 41 | 28.988 | 28.988 | 0.962 | 30.438 |
| AW3D30 | 25 | 44 | 19 | 36.249 | 36.249 | 0.985 | 36.707 |
| ALOS RT2 | −13 | 8 | 21 | 4.592 | −0.338 | 0.985 | 5.695 |
| ALOS RT1 | −12 | 8 | 20 | 4.404 | −0.360 | 0.986 | 5.477 |

* Error = DEM value – GPS-RTK value

All of this, including tables and figures, should move to appendix. Convey relatively high
reliability factor as a feature of Figure 2?

Thank you for the comments! Texts, tables, and figures, have been moved to the appendix.
Yes, as shown in Figure 2 and discussed in the uncertainty section, altitude surveys convey
relatively high reliability.

One could retain the locations of GPS-RTK validation points as shown in Figure 3 at the
same time as removing text from main narrative to an appendix and data to a DEM folder
at the repository.

The locations of GPS-RTK validation points as shown in Figure 3 are retained. The data
of Fig. 6 (showing the measured altitudes v.s. altitudes from 7 DEMs) have been put in a
DEM.xlsx, which has been uploaded to the National Tibetan Plateau Data Center.

Section 4.3, Soil Thickness
Thickness of weathered layers. e.g depth to bedrock from other studies minus surface soil
depths from this study will give a difference equal to the second lower weathered layer?
But these calculations will happen later, subsequent to data gathered and described here?
With what uncertainty? Plus/minus 1m? 10m?

Yes, depth to bedrock minus soil depth will result in the estimated thickness of the less
weathered layer. The section has been rewritten. The calculation and uncertainty has been
added.
Based on the measurements, the relationship between the soil thickness and slope can
be expressed using the equation:
$y=-1.1739x+82$    $(0 \leqslant x \leqslant 46)$          (9)
Where $x$ is the slope (°), $y$ is soil thickness (cm). Equation 9 is a regression line from data
obtained over residual soils in the west. The measured thickness is a result of in-situ soil forming processes. While in the east, a transported soil is observed, the thickness of which is
controlled by different processes from those acting on residual soils. In general, assuming
similar geology and except for the valley bottom, equation 9 would apply to the western study
area (Fig. 7b).
In the west, under the soil layer, a less weathered layer exists where water can also flow and
needs to be taken into account in the conceptual model. In the field, the difference between
the less weathered layer and the soil layer is that the less weathered layer contains partially
weathered stones. According to the owners of three boreholes located in or near the valley
(numbered 32-34 in Fig. 8), their depths are larger than 10 m and do not reach bedrock. By
subtracting the estimated soil thickness (Fig. 7b) from available depth to bedrock estimates,
for example from Yan et al. (2020) and Shangguan et al. (2017), the thickness of the less
weathered layer can be estimated (Fig. 8). In the mountainous west, because the estimated
depth to bedrock is often at least an order of magnitude larger than the soil thickness, the
uncertainty of the less weathered layer thickness mainly depends on the uncertainty of the
estimated depth to bedrock, which is high due to the lack of boreholes for appropriate
training (Shangguan et al., 2017; Yan et al., 2020).

[Figure]

(a)                                                                                    (b)

**Figure 7. Soil thickness: (a) soil thickness (cm) vs. slope (°); (b) estimated soil thickness using eq. 9.**

[Figure]

(a)                                                                                    (b)

**Figure 8. The estimated thickness of the less weathered layer in the west: (a) based on the ensemble model**
**estimated depth to bedrock from Yan et al. (2020); (b) based on the depth to bedrock from Shangguan et al.**
**(2017).**

Section 4.4.1, Water table depth measurement
Needs revisions to improve content and references. If "Surfer" and "Ordinary Kriging"
represent formal tools, we need to know source, citation, and open availability. This refers
to the commercial software 'Surfer'? Not available to most users.
Thanks for the comments! The content and references have been improved. 'Surfer' is a
commercial software 'Surfer':
There were 22 WT depths measured in 2018, and 18 in 2019 (Fig. 3). In the flat eastern
area, the WT depths were interpolated with the software Surfer
(https://www.goldensoftware.com/products/surfer) using the default Ordinary Kriging
method with the linear variogram model (slope=1, anisotropy ratio=1, anisotropy
angle=0)(Cressie, 1990, 1991), which provides reasonable grids in most circumstances.
Text about linear variogram seems to come straight from GoldenSoftware website?
Thanks for the comments! Ordinary Kriging with linear variogram model is a default
interpolation method from the Surfer. The linear variogram model describes spatial
relationships for the Kriging method. More detailed information on ordinary Kriging and
linear variogram model can be found in newly added references shown in the above red
lines.
Very standard tool, open access substitutes must exist?
One option is to use the trial license. It's free and allows unlimited access to all Surfer
features for two weeks. Software information and substitutes have been included in a new
sheet named "Softwares" as you suggested.

| | A | B | C | D | E |
|---|---|---|---|---|---|
| 1 | Software name | Source | Type | Purchase information | Open-access options or substitutes |
| 2 | AquiferTest | https://www.waterloc | proprietary | around USD$1000, see | the trial license allows a 15-day trial, it |
| 3 | RES2DINV | https://www.aarhusge | proprietary | prices available at: http | the trial license allows a 2-week trial, i |
| 4 | Samovar V6.6 | http://www.iris-instru | free open-access | | MRSMatlab, see published paper: MRS |
| 5 | TEM–Researcher | http://www.aemr.net | proprietary | contact: Tem-Fast@AEMR.net | |
| 6 | Surfer | https://www.goldenso | proprietary | $999, see https://www | the trial license allows a 2-week trial, i |
| 7 | | | | | |
| 8 | | | | | |

... | Magnetic susceptibility | ERT | MRS | TEM | **Softwares** | ⊕

Next paragraph induces confusion. Because people in the west use surface water,
need/interest in ground water remains low and few wells drilled? As a consequence, few
boreholes exist? These are boreholes numbered 32-34 in Figure 8?
Thanks for the comments and sorry for the confusion. The sentence has been modified:
Owing to the fact that most people living in the mountainous west use water from streams
(via field survey), the need for groundwater is low, and only few boreholes exist. As such,
only three boreholes numbered 32-34 were found in that western area (Fig. 9) and WT
depths were measured.

Because of rarity, authors decided to exclude these from interpolation. What interpolation?
The 'Surfer' interpolation already mentioned? No details given. First mention of
interpolation in this document.
Thanks for the comments. "interpolation" means interpolation of water table depths or
piezometric heads in Surfer as mentioned at the beginning of the paragraph.
A good interpolation over a large area needs / uses every point, regardless of isolation?
In this catchment, these points deleted for reasons of quality or for reasons of geographic
isolation?
Thanks for the comments. The paragraph has been modified:
Normally, a good interpolation of WT depths or piezometric head over a large area needs
and uses every measurement. But in this case, a reasonable WT depth map or piezometric
head map in the mountainous west will need more than 100 borehole measurements
(Hopkins and Anderson, 2016), because the ground surface elevation changes
dramatically in the west and so does the groundwater level. The three boreholes are far
from enough to provide a reasonable WT depth map or piezometric head map, and,
therefore, were excluded from the interpolation. In contrast, the measured groundwater
depths (and the interpolation) in the eastern study area can give a reasonable WT depth
map or piezometric head map (Fig. 9a and Fig. 9b).
First and only reference to a dam? Here authors assign lower water tables in 2019 vs 2018
to differences in dam storage, but in concluding sentence of the paragraph the authors
mention different "control points" as well as different dam storage conditions. Need
revision and clarity here!
Thanks for the comments! An introduction about the reservoir has been added in the
section Study area.
There is a reservoir in the catchment (Fig. 1c), with functions of grassland irrigation and
flood control.
"dam" has been replaced by "reservoir (Fig. 9e)". Sorry about the confusion, the confusing
sentences have been removed and the paragraph has been rewritten:
In general, the range of WT depth was between 0.0 m to 19.1 m in 2018 and between 0.7
m to 18.0 m in 2019. In both 2018 and 2019, the interpolated WT depths (Fig. 9a and Fig.
9b) show a similar trend, i.e. the depth increases from the middle of the study area to the
eastern boundary. The difference in WT depth in 2018 and 2019 (Fig. 9e) is probably
caused by: 1) different positions and amount of control points; 2) the gates were open to
reduce water storage in the reservoir (Fig. 9e) in 2019 to facilitate nearby constructions;
3) the interannual variation of precipitation and evapotranspiration. Nevertheless, in both
2018 and 2019, hydraulic heads (Fig. 9c and Fig. 9d) decrease from the middle of the
study area to the eastern boundary, meaning that the groundwater flow is from the west
to the east with the hydraulic gradient of about 0.002 (dimensionless), recharging the

Yellow River (Fig. 9f). This is consistent with the conclusion from Chang (2009). Ground
surface elevations in Fig. 9f were extracted from ALOS RT1, and hydraulic heads were
extracted from Fig. 9c and Fig. 9d. Some hydraulic heads are higher than the ground
surface elevations as shown in Fig. 9f, which is due to: 1) the accuracy of ALOS RT1; 2) the
lack of control points of hydraulic heads.

Final short paragraph of this Section highly redundant. Remove it, or move it to Abstract?

Thanks for the comments!
The final short paragraph has been removed as you suggested.

Section 4.4.2, Aquifer tests
Here the authors accept / use data from isolated rare western stations. Because they do
not apply a software interpolation?

Thanks for the comments! There's no interpolation done in this section, it's just presenting
the hydraulic conductivities in different boreholes located in the whole study area. An
explaining sentence has been added:
Considering the spatial heterogeneity of hydraulic conductivities, they were not
interpolated in the study area.

Authors provide and justify a range of hydraulic conductivities (e.g. ranged from 0.1. to
15.6 m per day) but data provided includes only geographic coordinates and raw data,
not these derived conductivities. Users will need to make their own conversions?

Thanks for the comments! Aquifer tests data in the repository includes only geographic
coordinates and raw data. This is because the detailed calculation processes and derived
conductivities are described and available in the paper in detail: the method is introduced
in Section 3.4.2. The processing software, assumptions based on field observation, and
finally the resulted hydraulic conductivities are described in Section 4.3.2: Users do not
need to make their own conversions, they can directly use the derived hydraulic
conductivities shown in the paper whenever they want.

Better that authors describe their calculations and provide derived conductivities in
addition to raw data, directly in the repository product?

Thanks for the comments! The derived conductivities have been included in the National
Tibetan Plateau Data Center. The details of calculation: method, software, assumptions,
are described in the paper. So maybe it's not important to include them in the repository.

Section 4.5.1 Magnetic susceptibility
Low values of magnetic susceptibility needed only to assure validity of subsequent ERT
or MRS measurements. Provide only a brief sentence of assurance here and refer to text
/ figures in an appendix as well as data in NTPDC for those who want?

Thanks for the comments! The section has been replaced by a summary sentence, and merged with the MRS section. The original content has been moved to the appendix as you suggested. The figure showing the data has been retained because it's easier to see the locations of each value.

4.4.1 MRS

ERT results were used to establish geoelectrical models for MRS inversion (see Appendix A3). The magnetic susceptibility measurements reveal very low susceptibility in the catchment, ensuring the suitability of applying MRS in the study area (see Appendix A2). ⋯⋯

Section 4.5.2 ERT

RES2INDV software mentioned here (first mentioned in Section 3.5.2, ERT) represents another proprietary commercial software not available to most readers / users. Perhaps common in geophysical methods but authors need to describe open-access alternatives. Or, we need a list of proprietary software dependencies that covers the entire measurement suite?

Thanks for the comments! For open-access alternatives, one option is to use the 2-week trial license. It's free and easily assessable. Softwares information and substitutes have been included in a new sheet named "Softwares" as you suggested.

[Figure]

| | A | B | C | D | E |
|---|---|---|---|---|---|
| 1 | Software name | Source | Type | Purchase information | Open-access options or substitutes |
| 2 | AquiferTest | https://www.waterloc | proprietary | around USD$1000, see | the trial license allows a 15-day trial, it |
| 3 | RES2DINV | https://www.aarhusge | proprietary | prices available at: http | the trial license allows a 2-week trial, i |
| 4 | Samovar V6.6 | http://www.iris-instru | free open-access | | MRSMatlab, see published paper: MRS |
| 5 | TEM–Researcher | http://www.aemr.net | proprietary | contact: Tem-Fast@AEMR.net | |
| 6 | Surfer | https://www.goldensc | proprietary | $999, see https://www | the trial license allows a 2-week trial, i |
| 7 | | | | | |
| 8 | | | | | |

◄ ► … | Magnetic susceptibility | ERT | MRS | TEM | **Softwares** | ⊕

"Half of the data missing in the filtering process"? What filtering process? Part of the proprietary RES2INDV processing? Are these data flagged? We get no information on data needed to meet various quality control criteria;

Thanks for the comments!
The details about filtering have been added at the beginning of the paragraph now:
For a specific pseudo depth, the values between adjacent points generally vary smoothly. Bad data points can be easily identified as they appeared as outlier points in the pseudosection plot in RES2DINV due to their too high/low apparent resistivity values. The bad data points were filtered out based on the following criteria: (i) having negative apparent resistivity or small apparent resistivity close to 0 Ω m; (ii) having negative/positive pulse amplitude ratios < 0.75 or > 1.33 (a measure of waveform symmetry) (Slater et al., 2010; Wilkinson et al., 2016).

did so much data at so many sites fail? How do these failures affect overall conclusions?

Affects only ERT1? But periodic rainfall occurred at other stations as well?

For ERT1, rainfall occurred during the field measurement, resulted in many bad data points. These failures do not affect overall conclusions, they affect only ERT1. Because ERT1-ERT7 are located at different places, they are independent of each other, and there's no rainfall when conducting ERT2-ERT7.

Overall, ERT measurements seem useful to or necessary for MRS measurements but not useful or reliable in the absence of other depth-resolved lithology information, for example. Authors say "ERT has equivalence problems, i.e., non–uniqueness of inversion results." Provide instead a short sentence assuring that ERT supports and allows valid MRS measurements but put ERT test in a separate Appendix? You already have ERT data in a repository?

Thank you for the comments! The section has been replaced by a summary sentence as you suggested, and merged with MRS section.

4.4.1 MRS
ERT results were used to establish geoelectrical models for MRS inversion (see Appendix A3). The magnetic susceptibility measurements reveal very low susceptibility in the catchment, ensuring the suitability of applying MRS in the study area (see Appendix A2). ⋯⋯

ERT section also has been modified:
⋯⋯Nevertheless, like other geophysical methods, ERT has equivalence problems, i.e., non–uniqueness of inversion results. Despite equivalence problems, the ERT method still provides important subsurface information in Maqu catchment where we have little fundamental information. This is a very first investigation in this area, when more lithology information becomes available later, ERT can be better constrained to reflect the subsurface lithology.

ERT data are already in the repositories.

Section 4.5.3 MRS
"at two near MRS sounding sites" - Authors mean at two 'adjacent' sites? Sufficient mention of ERT here, don't need a separate section?

Thank you for the comments! Yes, 'adjacent' is more accurate, thanks. The sentence mentioning ERT has been removed.

Samovar V6.6 mentioned here represents open access software from IRIS instruments (e.g. described in Section 3.5.3) but a few sentences later in this section reader encounters Samovar V11.4? Different software version? Different instrument type? Because authors clearly assign interpretation differences to V6.6 vs V11.4, readers need to know source of those differences?

Thank you for the comments! Samovar V11.4 is the updated version of Samovar V6.6,
and has been explained in the paper:
Besides, the invalid values for T2* and T1 may be attributed to the hydrogeological
conditions, such as highly heterogeneous lithology or too low signal/noise ratio, and may
be eased using an updated version of Samovar V6.6, such as Samovar V11.4 which not
only improves the capability of signal analysis, for example, allows optimizing the number
of inverted layers, but also adds uncertainty estimation function by incorporating singular
value decomposition.

How much "in situ" water is "missing". 10%? 50%? Not surprising, but how does a reader
find this information?

The information about missing water has been added:
MRS has its own limitations in that some of the in-situ water information is missing, and that
the current 'window of the technique' is only sensitive to the larger pore fraction of water
content. Near-source river environment leads to the unknown mixture of varied lithology.
Missing water is unknown, but accounting for a variety of lithology, including fine pore ones,
from water table depth (Fig. 9) to the base of the aquifer (50 to 208 m range, see the following
TEM section) may lead to well over 50% missed water (Boucher et al., 2011).

"Un-determination": what does this mean? Not resolved? Under-determined? Other
instrumentation or lithological factors?

"Un-determination" has been replaced by indetermination. It means not determined.

We need a much different, much better discussion of sources and levels of uncertainty
here; this reader found very little basis to accept any MRS data. Did MRS function
effectively or not given these (supposed?, estimated?, measured?) aquifer depths. Not
clear that MRS contributes valid information to hydrogeological model, e.g. more/better
than borehole estimates. Authors do not provide information necessary to make that
determination? By authors own admission, the best they / we can get from MRS remains
amount of free water?

Thank you for the comments! They are very helpful! MRS results have been rewritten: Part
1. presenting information about data processing; Part 2. Explain the results; Part 3.
Problems. In Uncertainty section (shown at the beginning of this document), uncertainties
of all the results were discussed, including MRS. Part 2 and 3 are as follows:
The water content distribution of MRS9–2, MRS7-2, MRS7-1, and MRS4-1 (Fig. A3) extends
down to 150 m deep. Except for MRS4-1, soundings MRS9–2, MRS7-2, and MRS7-1 are
adjacent, indicating that in the southeast, near the Yellow River, the groundwater extends to
more than 150 m depth. So it is concluded that the flat east plays the main role in storing
groundwater and the groundwater can extend to more than 150 m depth.
Limiting values of 0.00 ms and 1000.00 ms for T2* and 0.00 ms and 3000.00 ms for T1 are indicators that a valid numerical solution to the measured records (i.e., the inversion) was not
reached and no valid outcome is available. Except for invalid values, T1 derived hydraulic
conductivity (KT1) ranges from 0.00 m d-1 to 210.98 m d-1, T2* derived hydraulic conductivity
(KT2*) ranges from 0.00 m d-1 to 19.64 m d-1. The value of 0.00 m d-1 comes from the
estimation of very low water content. Here, an order of magnitude difference is observed
between the range of KT1 and KT2*, which is due to the big difference between T1 and T2*.
In theory, T1 is less affected by magnetic heterogeneities, thus permits a better estimation of
the hydraulic conductivity compared to T2*. However, it is to note that no magnetic
disturbance is expected in Maqu catchment (Fig. A1). Furthermore in the case of T1, because
two timed delayed responses are compounded, any model mismatch, e.g. the MRS loop
sampled volume being significantly different from a layered model parallel to the loop due to
near-source river deposit media heterogeneity, can make the measured responses 'doubly'
distorted and may not fit a T1 expected response. In both cases, T1 and T2*, a distortion is
occurring. Nevertheless, according to specific circumstances, T2*, which is evaluated from rest
with a single pulse, may undergo less severe overall distortion. So KT2* and TT2* tend to be
more reliable than KT1 and TT1, and should be used for future study. By checking the values
of KT2*, it is concluded that there is an unconfined aquifer in the eastern study area. Based
on KT1 (and water content results), with a proper threshold to define aquifer and non-aquifer,
the aquifer geometry can be defined.
MRS has its own limitations in that some of the in-situ water information is missing, and that
the current 'window of the technique' is only sensitive to the larger pore fraction of water
content. Near-source river environment leads to the unknown mixture of varied lithology.
Missing water is unknown, but accounting for a variety of lithology, including fine pore ones,
from water table depth (Fig. 9) to the base of the aquifer (50 to 208 m range, see the following
TEM section) may lead to well over 50% missed water (Boucher et al., 2011). Besides, the invalid
values for T2* and T1 may be attributed to the hydrogeological conditions, such as highly
heterogeneous lithology or too low signal/noise ratio, and may be eased using an updated
version of Samovar V6.6, such as Samovar V11.4, which not only improves the capability of
signal analysis, for example, allows optimizing the number of inverted layers, but also adds
uncertainty estimation function by incorporating singular value decomposition. Nevertheless,
in highly heterogeneous environments, the indetermination of some parameters may remain
with current technology. In terms of using default inversion parameters, part of the difficulty
is in fitting the observed data to a too large number of layers: i.e. partly fitting to the noise
component of the records. The heterogeneity of the near-source river environment is also
contributing to this difficulty. With more recent tools, like Samovar V11.4, the difficulty can be
better handled (Legchenko et al., 2017).
In general, MRS provides preliminary and valuable information on water content, hydraulic
conductivity, and transmissivity. Once the geometrical mapping and its fabric have been
mapped, groundwater flow parameters, and groundwater storage or volume can be better
determined.
Section 4.5.4 TEM
Based again on a commercial proprietary software TEM-Researcher (e.g. mentioned in
Section 3.5.4); can the authors explain or list open-access alternatives?

*Thanks for the comments! Unfortunately, this is the only software that we can not find*
*open-access alternatives.*

In Figure 13 the linear red lines indicate the initial model with the connected red dots
represent the interpolated values?

*Thanks for the comments! The explanation of line and triangle in the figure has been*
*added.*

[Figure]

TEM8 (RMS error = 9.89%)                    TEM9 (RMS error = 0.95%)

Figure 12. Apparent resistivity with depth. The red triangles connected by the red line
represent the measured apparent resistivity values, and the red line without triangles
represents the inverted 1-D geoelectric model.

At best, these represent preliminary data, e.g as the authors say "several additional
measurements will be needed in the future". Need explicit uncertainties here!

*Thanks for the comments! About additional measurements, the explanation has been*
*given:*
To determine exactly what structure it is, and the scope of the structure, further
investigation is needed. For example, a systematic high spatial resolution geophysical
survey with appropriate depth capability, such as the airborne electromagnetic survey,
followed by systematic borehole drilling.

*For uncertainties, in the TEM section, it's said that:*
The RMS error of the inversion results shown in Fig. 12 is below 2% in the flat area and
below 10% in the mountainous area.
*And in the uncertainty section, the uncertainties were also discussed.*

Section 6 Conclusion
Authors write "data in this paper can be used for future set up of a hydrogeological
conceptual model and groundwater modeling which will be presented in follow up
papers." Good effort, no doubt, and thanks for an admirable effort to share, but reader
never learns how close they got to a useful reliable groundwater model. What are their priorities for future efforts? Improve instruments / measurements in this catchment?
Duplicate work in a second catchment? Focus on modeling rather than observations?
Thanks for the comments! The conclusion has been modified:
Generally speaking, all the methods work well in the study area, and have confirmed the
presence of an unconfined fluvial aquifer within the 250 m below surface and the
presence of lake deposits with much finer pores lithology. By combining our dataset with
available depth to bedrock dataset, a preliminary hydrogeological conceptual model can
be established. If combining our dataset with detailed geometrical mapping of the
subsurface and deep borehole information, a more complete and accurate conceptual
model can be obtained. Furthermore, we will be monitoring the groundwater and surface
water in the study area and aim for establishing a long-term monitoring network, which
will eventually contribute to the verification and validation of future studies on
groundwater modeling over the Maqu catchment.
How do they recommend that potential users consider or use these data? What do they
consider strong or adequate? Where (everywhere?) do they recommend future
improvements? Can we as users rely on their soil depths, their borehole pumping data,
their "unconfined" aquifer conclusions? Authors give users very little basis for confidence
in their efforts and their data.
Thanks for the comments! The conclusion has been modified:
The data from direct techniques with low uncertainties, as shown in Fig. 2: lithology,
ground surface elevation, soil thickness, water table depth measurement, hydraulic
conductivity from aquifer test, magnetic susceptibility, can contribute to related global or
regional databases where the in-situ data over TP is scarce, or be regarded as verification
and validation data for groundwater modeling over Maqu catchment. The data from
indirect techniques, ERT, MRS, and TEM, is a rare unique and particularly rich training data
source for geoscientists interested in the data processing and interpretation of the
particular hydrogeological and hydrogeophysical techniques used here. It is a dynamic
set where additional complementary data will gradually add constraints to the inversion
processes. For example, a researcher developing new techniques for S/N improvements
of some of these techniques will get free and highly relevant data to work with.